# Beyond Noether: A Covariant Study of Poisson-Lie Symmetries in Low Dimensional Field Theory

**Florian Girelli[1]★, Christopher Pollack[1,2]† and Aldo Riello[1,2]‡**

**1** Department of Applied Mathematics, University of Waterloo, 200 University Avenue West, Waterloo, Ontario, Canada, N2L 3G1
**2** Perimeter Institute for Theoretical Physics, 31 Caroline St. N., Waterloo, Ontario, Canada, N2L 2Y5

★ florian.girelli@uwaterloo.ca , † cajpolla@uwaterloo.ca , ‡ ariello@perimeterinstitute.ca

## Abstract

We explore global Poisson-Lie (PL) symmetries using a Lagrangian, or "covariant phase space" approach, that manifestly preserves spacetime covariance. PL symmetries are the classical analog of quantum-group symmetries. In the Noetherian framework symmetries leave the Lagrangian invariant up to boundary terms and necessarily yield (on closed manifolds) $\mathfrak{g}^*$-valued conserved charges which serve as Hamiltonian generators of the symmetry itself. Non-trivial PL symmetries transcend this framework by failing to be symplectomorphisms and by admitting (conserved) non-Abelian group-valued momentum maps. In this paper we discuss various structural and conceptual challenges associated with the implementation of PL symmetries in field theory, focusing in particular on non-locality. We examine these issues through explicit examples of low-dimensional field theories with non-trivial PL symmetries: the deformed spinning top (or, the particle with curved momentum and configuration space) in 0+1D; the non-linear $\sigma$-model by Klimčík and Ševera (KS) in 1+1D; and gravity with a cosmological constant in 2+1D. Although these examples touch on systems of different dimensionality, they are all ultimately underpinned by 2D $\sigma$-models, specifically the A-model and KS model.

# 1 Introduction

Symmetries are a fundamental aspect of analyzing and solving partial differential equations (PDEs). When the studied PDEs originate in a variational problem, through Noether's theorem, symmetries provide conserved or invariant quantities called *charges*, which help find and classify solutions [1, 2]. When a system has a maximal number of charges, in involution, it is said to be *integrable* [3].

In physics, equations of motion are (usually) PDEs derived from Hamilton's variational action principle involving a local field functional called the *Lagrangian* which features at most first order derivatives of the fields.

Given a Lagrangian, the space of solutions $\mathcal{F}_{\mathrm{EL}}$ to its equations of motion, called the *shell*, comes equipped with a (weakly) symplectic structure $\Omega \in \Omega^2(\mathcal{F}_{\mathrm{EL}})$, yielding the so-called *covariant phase space* $(\mathcal{F}_{\mathrm{EL}}, \Omega)$ [4–14].

Actions of a Lie group $\mathcal{G}$, or Lie algebra $\mathfrak{g}$, on the space of (field) configurations that leave the Lagrangian invariant, possibly up to a boundary term, are called *Noether symmetries*. Noether symmetries necessarily preserve the equations of motion, i.e. they map solutions of the equations of motion into other solutions thence defining actions on the covariant phase space. This action can be shown to be not only a symplectomorphism but also Hamiltonian.

The converse is in general *not* true: there can be Lie group or Lie algebra symmetries of the equations of motion that are *not* symmetries of the corresponding Lagrangian [15, Section 4.2]. These *non*-Noether symmetries, as we will discuss at length later, can still lead to physically meaningful conserved charges.

Noether's first theorem assigns to each Noether symmetry $\mathfrak{g}$ and to each solution of the equations of motion $\varphi \in \mathcal{F}_{\mathrm{EL}}$ a $\mathfrak{g}^*$-valued *Noether current* $j^\mu$ that is conserved on-shell, $\nabla_\mu j^\mu|_\varphi = 0$. By integration over any Cauchy surface $C$, the Noether current yields a *Noether charge* $q|_\varphi \in \mathfrak{g}^*$, which is *independent* of the choice of $C$. Understood as a map $q \in \Omega^0(\mathcal{F}_{\mathrm{EL}}) \otimes \mathfrak{g}^*$, the Noether charge is the Hamiltonian generator, or "momentum map", of the flow of the Noether symmetry over the covariant phase space.

This analysis can also be performed in a non-spacetime covariant setting. Choosing a foliation of spacetime, one can recast the second order Lagrangian equations of motion into a Hamiltonian form, where configurations over a canonical phase space $\mathcal{P}_{\mathrm{can}}$ evolve in time through a set of first order differential equations encoded in the Hamiltonian functional. The canonical and covariant phase spaces are in fact symplectomorphic, whence Noetherian symmetries can be expressed in terms of Hamiltonian actions over the canonical phase space. In the canonical—or Hamiltonian—framework, a Noether symmetry compatibility with the Lagrangian function is replaced with its compatibility with the Hamiltonian functional.

The study of integrable systems in the Hamiltonian framework led to the discovery of a new class of so-called *Poisson-Lie symmetries*, which are a "classical limit" of quantum group (Hopf algebra) symmetries [16–25]. These symmetries feature conserved charges valued in a non-linear space, in fact a Lie group, denoted $\mathcal{G}^*$, rather than in a vector space such as the dual of the Lie algebra $\mathfrak{g}^*$. Furthermore, Poisson-Lie symmetries fail to be symplectomorphisms, although this failure is tightly controlled by the group structure on $\mathcal{G}^*$. In fact, for this class of symmetries it is possible to restore a generalized notion of symplectomorphism by equipping the symmetry group itself $\mathcal{G}$ with a non-trivial (necessarily degenerate) Poisson structure so that its action on the (symplectic) phase space, seen as a map $\mathcal{G} \times \mathcal{P}_{\mathrm{can}} \to \mathcal{P}_{\mathrm{can}}$ is a Poisson map. This generalization of the notion of symplectomorphism is thus possible if the Poisson structure on $\mathcal{G}$ is related to the Lie group structure of $\mathcal{G}^*$. The standard, Hamiltonian, notion is then recovered in the limit where the Poisson structure on $\mathcal{G}$ is trivial (vanishes) and thus $\mathcal{G}^*$ becomes Abelian and identifies with its Lie algebra, the vector space $(\mathfrak{g}^*, +)$.

From this overview one thing is clear: Poisson-Lie symmetries *cannot* be Noetherian symmetries, since the latter necessarily yield conserved charges valued in a linear space generating Hamiltonian actions on phase space, whereas the former yield conserved charges valued in a non-linear space generating actions that fail to be symplectomorphisms. This is probably why, historically, these symmetries were often characterized as "hidden symmetries" [26–28] made apparent deep into the analysis of the model (e.g., from the existence of braiding relations amongst scattering amplitudes [3, 26, 29, 30]) far beyond the typical Lagrangian or Hamiltonian starting point of any field theoretic analysis.

In an effort to uncover a unifying covariant framework that encompasses both Noetherian and Poisson-Lie symmetries, in this manuscript we investigate in a number of examples of increasing complexity, the properties of Poisson-Lie symmetries from a *spacetime covariant* perspective, focusing in particular on those aspects in which Poisson-Lie symmetries depart from the standard Noetherian framework and genuinely go beyond it. For example, from very early on, we will stress how a tension exists between Poisson-Lie symmetries and *locality.*

Our article is organized as follows. Section §2.1 is dedicated to reviewing notions of locality for different structures on field space. Special emphasis is put on this as we will see that one can expect Poisson-Lie symmetries and their generators to be *non-local*. In section §2.2, we revisit the covariant phase space method. In section §2.3, we recall and emphasize the strength of Noether's *construction in the covariant phase space* to build conserved charges in terms of *local* currents when we have Lagrangian/Noetherian symmetries. We propose then a framework for Poisson-Lie symmetries and charges in the covariant phase space context which goes beyond Noether's framework for Lagrangian symmetries. In §2.4, we recall for the unfamiliar reader the notions of Poisson-Lie groups, Drinfel'd and Heisenberg doubles, as well as that of Poisson-Lie symmetries of a symplectic (or Poisson) manifold. The subsequent sections study various examples of Poisson-Lie symmetric field theories in different spacetime dimensions:

- *0+1D Field Theory: The Generalized Spinning Top & Deformed Spinning Top* (§3)
  These models are about mechanical systems which have either configurations or momenta valued in a (non-Abelian) group, the regular spinning top having a phase space of the form $T^*SO(3)$. For cases involving both non-trivial configuration *and* momentum space geometries, the particle's action must be extended from a worldline to a "topological" string worldsheet, as discussed in [31].

- *1+1D Field Theory: The Klimčík & Ševera (KS) Model* (§4)
  This model describes a string with non-Abelian Lie group target space via a non-linear $\sigma$-model which dynamically satisfies a non-Abelian T-duality, as was introduced and extensively studied by Klimčík and Ševera [32–34]. We will detail the Poisson-Lie symmetries of a manifestly covariant version of the original second order KS model [32], for both the open and closed string.

- *2+1D Field Theory: 3D Gravity as a BF Theory* (§5)
  This model is a first order description of 3D gravity with a cosmological constant. We will recall how Poisson-Lie groups are relevant in organizing and solving for the physical data of the theory [35], with non-trivial Poisson-Lie symmetries arising upon discretization. As a topological theory, its holographic nature reduces it to a 1+1D field theory at the boundary, showing parallels in its symplectic structure with the KS first order open string model [36].

These models and their associated Poisson-Lie symmetries will be shown to share a key feature: the codimension-$d$ localization of Poisson-Lie charges. Obtaining a non-trivial group-valued conserved scalar charge (Poisson-Lie charge) associated to the Poisson-Lie symmetry is shown to require codimension-$d$ substructures (marked points) in spacetime. For the 0+1D particle case, this condition is clearly met by picking a 0D "Cauchy surface". In the 1+1D KS model, the string must be open to satisfy this condition. In 2+1D gravity we show that non-trivial Poisson-Lie charges arise upon discretization of the 2D Cauchy surface into cells, from which it follows that the Poisson-Lie charges localize to the boundary of the edges (i.e., to vertices) defining cell intersections.

We conclude in §6 with a summary and outlook.

## 2 From Noether to Poisson-Lie Symmetries: A Spacetime Covariant Approach

In this section we begin by reviewing aspects of the differential geometry of fiber bundles, their jet spaces, and the infinite dimensional spaces of their sections as well as other "field spaces", with a focus on the notion of locality. We then introduce the covariant phase space formalism as a tool for treating symmetries geometrically and manifestly covariantly in field theory which we will make much use of throughout this manuscript. Then, we outline how one treats Noether symmetries in field theory using this formalism. We will in particular provide a change of perspective on the usual Noetherian construction that will help us motivate the extension to *Poisson-Lie symmetries* and their group valued momentum maps—which, unless trivial, do *not* fit the standard Noetherian framework. This will lead us to our proposal for a general, manifestly covariant, formalism for Poisson-Lie symmetries in field theory. Finally, we will review background material on Poisson-Lie groups, doubles, and dressing transformations which the reader familiar with the topic can skip.

### 2.1 Spaces of Fields & Locality

While mechanical systems can be described in terms of finite dimensional symplectic manifolds (phase spaces), a symplectic treatment of field theories requires infinite dimensional manifolds. A fundamental organizing principle when studying field theories and their symmetries is *locality*. We recall here some formal definition of this concept [14].

**Local Vector Fields & Local Actions.** Let $\pi : F \to M$ be a (finite dimensional) fiber bundle, and denote $\mathcal{F} := \Gamma(F)$ the space of its smooth sections. This can be given the structure of an infinite dimensional manifold[1] which we will interpret as the space of unconstrained field configurations over spacetime $M$, the "off-shell histories".

The $k$-th jet bundle $\pi_k : J^k F \to M$ is the bundle over $M$ whose fiber $\pi_k^{-1}(x)$ is the set of equivalence classes of sections $\varphi \in \mathcal{F}$ with the same $k$-th jet [39, 40]

$$j_x^k \varphi = (x, \varphi^I(x), \partial_\mu \varphi^I(x), \dots, \partial^k_{\mu_1, \dots, \mu_k} \varphi(x)). \tag{1}$$

Given two fiber bundles over $M$, $F_{1,2} \to M$, a map $f : \mathcal{F}_1 \to \mathcal{F}_2$ is said to be *local* if there exist a $k < \infty$ and a smooth map $f_k$ such that the following diagram commutes:

$$
\begin{array}{ccc}
M \times \mathcal{F}_1 & \xrightarrow{\mathrm{id}_M \times f} & M \times \mathcal{F}_2 \\
\downarrow{\scriptstyle j^k} & & \downarrow{\scriptstyle j^0} \\
J^k F_1 & \xrightarrow{\quad f_k \quad} & F_2
\end{array}
\tag{2}
$$

In other words, a map between spaces of sections of vector bundles over $M$ is local if the target section at $x \in M$ is a smooth function of a finite number of derivatives of the source section at the same point.

A field variation around $\varphi \in \mathcal{F}$ is an element of the (kinematical) tangent space $T_\varphi \mathcal{F}$ [37, Section 28 and 32]. A vector field on $\mathcal{F}$, denoted $\mathbb{X} \in \Gamma(T\mathcal{F})$, can be seen as a map $\mathcal{F} \to T\mathcal{F}$. This is itself a map between spaces of sections of bundles over $M$, since

---

[1]Often, of an infinite dimensional nuclear Fréchet manifold [37]. See also [38, Appendices A and B] for a brief explanation and detailed references to the functional analytic literature.

$TF = \Gamma(VF)$ with $VF \to M$ the vertical subbundle of $TF$.[2] Whence, $\mathbb{X}$ is said to be *local* if it is local as a map between $\Gamma(F) \to \Gamma(VF)$ [14].

The action of a (finite dimensional, real) Lie algebra $\mathfrak{g}$ on $\mathcal{F}$ is a smooth Lie algebra homomorphism

$$\mathbb{Q} : \mathfrak{g} \to \mathfrak{X}^1(\mathcal{F}) \tag{3}$$

where $\mathfrak{X}^1(\mathcal{F}) = (\Gamma(T\mathcal{F}), [\cdot, \cdot])$ is the Lie algebra of smooth vector fields on $\mathcal{F}$. An action $\mathbb{Q}$ is said to be *local* if its image is in the space of local vector fields.

In other words, if $\mathbb{Q}$ is local then the infinitesimal (symmetry) transformation of $\varphi(x)$ given by $\delta_\alpha \varphi(x) := (\mathbb{Q}(\alpha)\varphi)(x)$, $\alpha \in \mathfrak{g}$, depends on the value of $\varphi$ at $x \in M$ together with a finite number of its derivatives, also evaluated at $x$.

In this article we will deal both with local and *non-local* actions.

**Mixed & Local Forms.** On the space $M \times \mathcal{F}$, one can defined the *bi-complex* of mixed differential forms[3] $\Omega^\bullet(M \times \mathcal{F})$ equipped with the total differential $\mathbf{d} := d + \mathrm{d\!\!|}$, where $d$ is the exterior differential on $M$ and $\mathrm{d\!\!|}$ that on $\mathcal{F}$. The nilpotency of $\mathbf{d}$ implies $d\mathrm{d\!\!|} + \mathrm{d\!\!|}d = 0$.

A mixed form $\alpha \in \Omega^\bullet(M \times \mathcal{F})$ is said to be *local* if there exists a $k < \infty$ and a $\check{\alpha} \in \Omega^\bullet(J^k F)$ such that $\alpha = j^k \check{\alpha}$. By means of a direct limit one can define forms on the infinite jet-bundle, and through the infinite jet-evaluation

$$j^\infty : M \times \mathcal{F} \to J^\infty(F), \quad (x, \varphi) \mapsto (x, \varphi(x), \partial_\mu \varphi(x), \partial^2_{\mu\nu} \varphi(x), \dots) \tag{4}$$

define the space of *local forms*:

$$\Omega^\bullet_{(\mathrm{loc})}(M \times \mathcal{F}) \doteq (j^\infty)^* \Omega^\bullet(J^\infty(F)) \subset \Omega^\bullet(M \times \mathcal{F}). \tag{5}$$

On the space of local forms, $d$ and $\mathrm{d\!\!|}$ agree with the pullback along $j^\infty$ of the horizontal and vertical differentials on $J^\infty(F)$ [14, 39, 40].

We denote the space of mixed (local) forms of bi-degree $(k, \ell)$ in the $(d, \mathrm{d\!\!|})$-bicomplex as $\Omega^{k,\ell}_{(\mathrm{loc})}(M \times \mathcal{F})$, so that

$$\Omega^\bullet_{(\mathrm{loc})}(M \times \mathcal{F}) = \bigoplus_{k,\ell \geq 0} \Omega^{k,\ell}_{(\mathrm{loc})}(M \times \mathcal{F}). \tag{6}$$

In the following, we will refer to the differentiation of a local form $\mu$ with respect to $\varphi \in \mathcal{F}$, i.e. $\mathrm{d\!\!|} : \Omega^{k,\ell}_{(\mathrm{loc})}(M \times \mathcal{F}) \to \Omega^{k,\ell+1}_{(\mathrm{loc})}(M \times \mathcal{F})$, $\mu \mapsto \mathrm{d\!\!|}\mu$, as the *variation* of $\mu$.

Intuitively, a local form $\mu \in \Omega^{k,\ell}_{(\mathrm{loc})}(M \times \mathcal{F})$ is a differential form of degree $k$ on spacetime *and* of degree $\ell$ on field space, that is furthermore characterized by a local spacetime dependence, meaning that $\mu(x, \varphi)$ depends on the value at $x \in M$ of: $(i)$ the field $\varphi$, $(ii)$ a finite number of its derivatives $\partial^n \varphi$, $(iii)$ a finite number of its variations $\mathrm{d\!\!|}\varphi$, and $(iv)$ a finite number of derivatives of its variations $\partial^m \mathrm{d\!\!|}\varphi \equiv \mathrm{d\!\!|}\partial^m \varphi$. For example, if $F \to M$ is a line bundle, $\mathcal{F}$ is the space of scalar field configurations $\varphi$ and e.g.

$$\mu(x, \varphi) = \mathrm{d\!\!|}\varphi(x) \wedge d\mathrm{d\!\!|}\varphi(x) = -\mathrm{d\!\!|}\varphi(x) \wedge \partial_\mu \mathrm{d\!\!|}\varphi(x) \wedge dx^\mu \in \Omega^{1,2}_{(\mathrm{loc})}(M \times \mathcal{F}) \tag{7}$$

is a local $(1, 2)$-form.

---

[2]Let $(x^\mu, u^I)$ be local coordinates on $F \to M$ and denote $(x^\mu, u^I, \delta x^\mu, \delta u^I)$ a set of local coordinates on $TF$. Then $VF \to M$ is the subbundle defined by $(x^\mu, u^I, \delta u^I) \mapsto x^\mu$.

[3]$\Omega^k(\mathcal{F}) := \Gamma(\mathrm{Lin}(\wedge^k T\mathcal{F}, \mathcal{F} \times \mathbb{R}) \to \mathcal{F})$ [37, Section 33].

205 For simplicity in tracking minus signs due to $\mathbb{d}d = -d\mathbb{d}$, it can be convenient to present
206 results of physical interests using tensor (densities) rather than differential forms.

207 On the space of forms and vector fields on $\mathcal{F}$ one can define a Cartan calculus [37, Sec-
208 tion 33] that we denote[4] $(\mathbb{d}, \mathbb{i}_\bullet, \mathbb{L}_\bullet)$ in analogy with the Cartan calculus on spacetime $M$,
209 denoted instead $(d, i_\bullet, L_\bullet)$. These operations naturally extend to mixed forms, where they
210 appropriately restrict to local forms. Although slightly redundant, this notation is apt
211 because we never consider mixed vector fields and thus it helps the reader to recognize
212 whether we are acting on mixed forms with a spacetime or a field-space vector field.

213 **A Remark on Locality.** The notion of locality make sense only on field spaces $\mathcal{F}$ which
214 are (unconstrained) sections of vector bundles. A subspace $\tilde{\mathcal{F}}$ of such an $\mathcal{F}$ might not be
215 a local space of fields, i.e. it might not be itself a space of unconstrained sections of a
216 subbundle $\tilde{F} \to M$ of $F \to M$. Although pullbacks of local forms from $\mathcal{F}$ to $\tilde{\mathcal{F}}$ and, when
217 defined, restrictions of local vector fields from $\mathcal{F}$ to $\tilde{\mathcal{F}}$, keep being meaningful as forms
218 and vector fields on $\tilde{\mathcal{F}}$; on such non-local $\tilde{\mathcal{F}}$ one cannot define spaces of *local* forms and
219 *local* vector fields *intrinsically.*

220 In physical applications, the "shell" $\mathcal{F}_{\mathrm{EL}} \subset \mathcal{F}$—the space of solutions to the equation of
221 motion defined shortly in (11)—can be isomorphic to a space of sections of a bundle over
222 a Cauchy surface $C \hookrightarrow M$ (corresponding to the specification of "initial data"). However,
223 this generally yields a notion of locality that depends on the specific choice of embedding
224 $C \hookrightarrow M$ and is therefore *un*satisfactory for our purposes since we want to maintain
225 manifest spacetime covariance. In fact, the shell $\mathcal{F}_{\mathrm{EL}}$ generally fails to be a space of
226 unconstrained sections of some fiber bundle over *spacetime $M$*. Intuitively this is because
227 we cannot *independently* choose the value of an on-shell field at causally connected points:
228 any changes in the field values in one region will propagate by the equations of motion to
229 all causally connected regions, and from one Cauchy surface to another. Therefore, if all
230 one has access to is the shell, and thus defines e.g. the action of a symmetry on on-shell
231 configurations only, one cannot make sense of what "spacetime locality" of said action
232 should even mean.

## 2.2 The Covariant Phase Space

234 The covariant phase space method [5–13] is a framework used to study the Hamiltonian
235 (phase space) dynamics of Lagrangian field theories without having to relinquish manifest
236 spacetime covariance. It allows one to construct a symplectic structure on the space of
237 on-shell fields $\mathcal{F}_{\mathrm{EL}}$ admitted by the governing Lagrangian without committing to any
238 foliation of spacetime. The method is a powerful tool for phrasing Noether's theorems
239 and thus studying the symmetry content of Lagrangian field theories. Applications of the
240 method throughout physics have been abundant, especially in recent years in the context
241 of gauge theories. Here we review the application of the covariant phase space method to
242 *global* symmetries, and highlight its limitations when it comes to the study of non-trivial
243 Poisson-Lie symmetries in field theory.

244 Let $M$ be a topological manifold of the form

$$M \simeq C \times \mathbb{R}. \tag{8}$$

245 Here $M$ has the interpretation of spacetime, and $\mathbb{R}$ of time. However, for us it is important

---

[4]Respectively: exterior differentiation of forms, interior differentiation of forms by vectors, and Lie
differentiation of forms along vectors.

246 not to commit to any specific foliation of $M$. This is what we refer to as "manifest
247 spacetime covariance" (the only exception is §4.5).

248 In the following, unless otherwise stated, the reader can assume that $C$ is a closed, i.e.
249 compact and boundary-less, (spacelike) Cauchy hypersurface. This said, throughout the
250 paper, we *will* often consider cases in which $C$ has boundaries, a circumstance that will
251 always be declared explicitly. As we are going to review below, if $\partial C \neq \emptyset$ the space of
252 (local) fields $\mathcal{F}$ is assumed to be restricted by a suitable set of boundary conditions along
253 the timelike boundary $\partial C \times \mathbb{R}$. In this case the breaking of locality is very mild, since
254 locality is maintained in the interior of $M$.

255 For us it is convenient to fix on $M$ a spacetime "background" Lorentzian metric $\gamma_{\mu\nu}$
256 once and for all. Fixing a metric allows us to identify $k$-forms over spacetime with skew
257 contravariant $(\dim(M) - k)$-tensors (multivector fields), and exterior derivatives of the
258 former with divergences of the latter. Denoting $\boldsymbol{\epsilon} = \star_\gamma 1$ the volume form on $M$ associated
259 to $\gamma$, for any $(\dim(M) - k)$-form $\boldsymbol{\alpha}$ we set[5]

$$\boldsymbol{\alpha} = \frac{1}{k!}\alpha^{\mu_1\ldots\mu_k}\boldsymbol{\epsilon}_{\mu_1\ldots\mu_k} \quad \text{and} \quad d\boldsymbol{\alpha} = \nabla_\nu \alpha^{\mu_1\ldots\mu_{k-1}\nu}\boldsymbol{\epsilon}_{\mu_2\ldots\mu_{k-1}}. \tag{9}$$

260 where $\boldsymbol{\epsilon}_{\mu_1\ldots\mu_k} := i_{\partial_{\mu_k}}\ldots i_{\partial_{\mu_1}}\boldsymbol{\epsilon}$ and $\nabla_\mu$ is the Levi-Civita connection associated to $\gamma$.

261 Mixed forms of spacetime rank $(\dim(M) - 1)$ will be called *currents*, $j^\mu$. When closed,
262 they are said to be *conserved*, $\nabla_\mu j^\mu = 0$.

263 Over the spacetime $M$ and the space of fields $\mathcal{F} = \Gamma(F) \ni \varphi$, consider a local *Lagrangian*
264 $\mathcal{L} \in \Omega^{\dim(M),0}_{(\text{loc})}(M \times \mathcal{F})$, involving at most first derivatives of the fields.

265 Varying $\mathcal{L}$ with respect to $\varphi \in \mathcal{F}$, and applying Takens' theorem for the decomposition
266 of $(\dim(M), 1)$-forms into their *source* and *boundary* components[6] [8, 42], one obtains re-
267 spectively the Euler-Lagrange equations of motion as the source component $\mathcal{E} = \mathcal{E}_I \mathbb{d}\varphi^I$
268 $\in \Omega^{\dim(M),1}_{(\text{loc})}(M \times \mathcal{F})$ and the divergence of a pre-symplectic *potential* (pSP) current
269 $\theta^\mu \in \Omega^{\dim(M)-1,1}_{(\text{loc})}(M \times \mathcal{F})$ as the boundary component:

$$\mathbb{d}\mathcal{L} = \mathcal{E} + \nabla_\mu\theta^\mu \equiv \mathcal{E}_I\mathbb{d}\varphi^I + \nabla_\mu\theta^\mu, \tag{10}$$

270 where we used $\bullet^I$ as a multi-index for the various field "species" and components.

271 Denote the subset of fields solving the equations of motion, called the *shell*, by

$$\mathcal{F}_{\text{EL}} := \{\varphi \in \mathcal{F} \ : \ \mathcal{E}_I(\varphi) = 0\}. \tag{11}$$

272 We will work formally as if $\mathcal{F}_{\text{EL}} \subset \mathcal{F}$ were both smooth (generally infinite dimensional)
273 manifolds. An equality that holds "on-shell" is denoted "$\approx$", which means "upon pullback
274 from $\mathcal{F}$ to $\mathcal{F}_{\text{EL}}$" along the embedding $\iota_{EL} : \mathcal{F}_{\text{EL}} \hookrightarrow \mathcal{F}$.

---

[5]Mind that if $k = \dim(M)/2$ this notation can create confusion, since the contravariant tensor $\alpha^{\mu_1\mu_2\cdots}$ is *not* the same as the covariant tensor $\boldsymbol{\alpha}$ with the indices raised by $\gamma^{\mu\nu}$. When this issue arises, we will introduce appropriate notation to disambiguate.

[6]Owing to Takens' theorem the space of local $(\dim(M), 1)$-forms decomposes into the direct sum of a space of *source* and *boundary* forms. Heuristically, *source forms* are such that they depend on the variation $\mathbb{d}\varphi$ but none of its derivatives, while *boundary forms* are $d$-exact. Takens' decomposition can be understood as a procedure akin to "integration by parts" at the level of $(\dim(M), 1)$-forms. This decomposition generalizes to other $(k, \ell)$-forms through Anderson's acyclicity theorem and "homotopy operators" [14, 39]; see [41] for a physical application.

The pre-symplectic *form* (pSF) current $\omega^\mu \in \Omega^{\dim(M)-1,2}_{(\text{loc})}(M \times \mathcal{F})$ of $\theta^\mu$ is defined as the variation

$$\omega^\mu := \mathrm{d}\theta^\mu. \tag{12}$$

It is manifestly $\mathrm{d}$-closed and, *on-shell*, also conserved i.e.

$$\nabla_\mu \omega^\mu \approx 0. \tag{13}$$

Indeed, $0 \equiv \mathrm{d}^2 \mathcal{L} = \mathrm{d}\mathcal{E} + \nabla_\mu \omega^\mu \approx \nabla_\mu \omega^\mu$.

If $M$ is globally hyperbolic with Cauchy surface $C$, $\partial C = \emptyset$, we can define a $\mathrm{d}$-closed 2-form $\Omega$ on $\mathcal{F}_{\text{EL}}$ by integration over $C$:

$$\Omega := \int_C n_\mu \omega^\mu, \tag{14}$$

where we introduced $n_\mu$ the unit conormal to $C \hookrightarrow M$, and left the induced volume factors implicit. The fact that the pSF current is conserved on-shell guarantees that on-shell $\Omega$ is independent of the choice of Cauchy hypersurface $C$. Since, at the end of the day, we will only be interested in on-shell equalities, we have here abused notation and omitted to label $\Omega$ by the Cauchy hypersurface $C$ on which it is defined off-shell.

In the absence of gauge freedom one expects $\Omega$ to be non-degenerate, hence symplectic.[7] The pair

$$(\mathcal{F}_{\text{EL}}, \Omega) \tag{15}$$

is then called *the covariant phase space* (CPS) of the theory, and it is expected to be isomorphic to the canonical phase space constructed by selecting a Cauchy surface $C \hookrightarrow M$ [5, 7, 13].

Conversely, in the presence of gauge symmetries, $\Omega$ is *not* expected to be non-degenerate; however, in this case, one still expects $\Omega$ to descend to a non-degenerate, and hence symplectic, 2-form on the *reduced* covariant phase space $\mathcal{F}_{\text{red}} \ni [\varphi]$ of on-shell field configurations *modulo* gauge [12, 44] (see also [38, Appendix D]). The focus of this article is on *global* symmetries, with gauge symmetries only appearing in §5.

In the following, we will also consider expressions such as

$$\Omega = \int_C n_\mu \omega^\mu \quad \text{with} \quad \partial C \neq \emptyset. \tag{16}$$

That is, instead of a closed Cauchy surface $C$ we will be considering the phase space of the theory over a spacetime manifold with timelike boundaries, $\partial M = \partial C \times \mathbb{R} \neq \emptyset$. The boundary $\partial C$ is codimension-2 and will be called the *corner*.

Note that in this case $\Omega = \int_C n_\mu \omega^\mu$ is a priori ill-defined, for there is an ambiguity in the definition of $\omega^\mu$ from the Lagrangian: Equation (10) defines $\theta^\mu$ only up to a $d$-exact form.[8] In some cases, e.g. if the Lagrangian is quadratic in the velocities and contains no higher derivatives, this ambiguity can be fixed by demanding that $\theta^\mu$ is linear in $\mathrm{d}\varphi^I$ with no derivatives acting on it, i.e. demanding that it is of the form $\theta^\mu(x) = \pi^\mu_I(x, \varphi, \partial\varphi, ...)\mathrm{d}\varphi^I(x)$. In all cases, given a theory, we will choose a $\theta^\mu$ and work with it, neglecting this type of ambiguities.

---

[7]More precisely, in infinite dimensions, the best one can often hope for is that $\Omega$ is "weakly" symplectic meaning that $\Omega^\flat : T\mathcal{F} \to T^*\mathcal{F}$ is only required to be injective. We take this to be understood throughout this manuscript. See e.g. [6, 38, 43].

[8]All $d$-closed $(k, l)$-forms for $k \neq \dim(M)$ and $l \geq 1$ are $d$-exact [39]. See also [12] for a physicists' argument.

In all cases in which $\partial C \neq \emptyset$, we will implicitly assume that the definition of the shell is augmented by the imposition of boundary conditions restricting the "allowed" field configurations to ensure that $\Omega = \int_C n_\mu \omega^\mu$ is conserved, i.e. independent of the choice of $C \hookrightarrow M$. In other words, denoting $s_\mu$ the unit conormal to the timelike boundary $\partial M = \partial C \times \mathbb{R}$, we demand that on shell $s_\mu \omega^\mu|_{\partial M} \approx 0$.

## 2.3 Symmetries: Noether & Poisson-Lie

We want to define a notion of symmetries on the CPS. We will now put forward a framework that explicitly highlights where one can go beyond the Noetherian notion of Lagrangian symmetries since, as we shall see, non-trivial Poisson-Lie symmetries are an example of such an extension.

**Noether Symmetries Revisited.** The idea is to define a symmetry by its desired properties: a Lie group (or algebra) action on phase space leading to conserved charges that moreover serve as symplectic generators of the symmetry itself. More formally, we introduce the following:

**Definition 2.1** (Noetherian symmetry). Given a local field theory $(F \to M, \mathcal{L})$, a *Noetherian symmetry* over the CPS $(\mathcal{F}_{\mathrm{EL}}, \Omega)$ is

1. a smooth Lie algebra *action*
$$\mathbb{O} : \mathfrak{g} \to \mathfrak{X}^1(\mathcal{F}_{\mathrm{EL}}), \tag{17}$$

2. whose flows are *Hamiltonian*,
$$\mathbb{i}_{\mathbb{O}(\cdot)}\Omega \approx \mathbb{d}q, \tag{18}$$

   for a momentum map denoted $q \in \Omega^0(\mathcal{F}) \otimes \mathfrak{g}^*$,

3. such that there exists a *local* off-shell current $j^\mu \in \Omega_{(\mathrm{loc})}^{\dim(M)-1,0}(M \times \mathcal{F}) \otimes \mathfrak{g}^*$ that satisfies for all $C \hookrightarrow M$ the equation
$$q \approx \int_C n_\mu j^\mu. \tag{19}$$

$\diamondsuit$

Points 1 and 2 simply state that a symmetry should be a Hamiltonian action on the covariant phase space. This is however not enough to select physically relevant symmetries associated to physically interesting conserved charges. This is where point 3 comes into play.

Before we explain its significance, let us come back to the remark at the end of section 2.1. The covariant phase space $\mathcal{F}_{\mathrm{EL}}$ is not a space of unconstrained sections of a fiber bundle over $M$. Therefore, it lacks a meaningful notion of locality and hence of conservation—which implicitly presupposes the fact that a quantity can be locally computed, e.g. "locally in time", and thus compared across different time instances, or Cauchy surfaces. This is why 3 needs to make reference to an off-shell current. We will come back to this point later.

In a $(d+1)$-Hamiltonian/canonical picture, conservation is a form of compatibility with the dynamics. This compatibility can be expressed in terms of the invariance of the Hamiltonian function $H$ under the action $\mathbb{O}$, i.e. $\mathbb{L}_{\mathbb{O}(\alpha)}H = 0$; in the case of a Hamiltonian action, this condition can be written as $\{q, H\} = 0$. However, in the covariant phase space

344 picture that we are pursuing here, it is less obvious what compatibility with the dynamics
345 means. Although a priori any vector field on $\mathcal{F}_{\text{EL}}$ "respects the dynamics", it can still do
346 so in a rather meaningless way: in fact, the Hamiltonian flow of *any*[9] function on $\mathcal{F}_{\text{EL}}$
347 defines a 1-parameter family of solutions to the equations motion that could be seen as
348 the action of a 1-dimensional Lie algebra, even if neither that function nor its action have
349 any distinguished physical meaning at all.[10]

350 Point 3 is meant to replace compatibility with the Hamiltonian function with a more
351 spacetime covariant notion of conservation of the charge. As noticed above, this requires
352 us to refer to the space of off-shell fields in order to borrow a viable notion of locality on
353 which to anchor our notion of conservation.

354 This is essentially how conserved currents appear via Noether's first theorem from a La-
355 grangian symmetry:

356 **Definition 2.2** (Lagrangian symmetry)**.** Given a local field theory $(F \to M, \mathcal{L})$, a *La-*
357 *grangian symmetry* is a *local* Lie algebra action $\mathbb{Q} : \mathfrak{g} \to \mathfrak{X}^1_{(\text{loc})}(\mathcal{F})$ that leaves the La-
358 grangian invariant up to boundary terms, namely such that there exists a *remainder*
359 *current* $R^\mu \in \Omega^{\dim(M)-1,0}_{(\text{loc})}(M \times \mathcal{F}) \otimes \mathfrak{g}^*$ for which

$$\mathbb{L}_{\mathbb{Q}(\alpha)}\mathcal{L} = \nabla_\mu \langle R^\mu, \alpha \rangle \quad \forall \alpha \in \mathfrak{g}. \tag{20}$$

360 $\diamond$

361 The remainder $R^\mu$ is valued in $\mathfrak{g}^*$ simply as a reflection of the linearity of $\mathbb{Q}$ in $\mathfrak{g}$.

362 **Theorem 2.3.** Lagrangian symmetries preserve the equations of motion

$$\mathbb{L}_{\mathbb{Q}(\cdot)}\mathcal{E} \approx 0 \tag{21}$$

363 and thus (assuming smoothness of $\mathcal{F}_{\text{EL}} \hookrightarrow \mathcal{F}$) descends to an action on the covariant
364 phase space. Furthermore, if $\partial C = \emptyset$, this action is Noetherian, i.e. it is Hamiltonian with
365 conserved (i.e. $C$-independent) *local* generators

$$q \approx \int_C n_\mu j^\mu \quad \forall C \hookrightarrow M \tag{22}$$

366 which are given by the integral of the conserved Noether current

$$j^\mu := (\mathring{\mathbb{i}}_{\mathbb{Q}(\cdot)}\theta^\mu - R^\mu) \in \Omega^{\dim(M)-1,0}_{(\text{loc})}(M \times \mathcal{F}) \otimes \mathfrak{g}^*, \quad \nabla_\mu j^\mu \approx 0. \tag{23}$$

367 In sum, if $\partial C = \emptyset$, all Lagrangian symmetries are Noetherian. $\diamond$

368 The (local) quantities $q$ and $j^\mu$ appearing in this theorem are called the *Noether charge*
369 and *Noether current*, respectively. The conservation of $j^\mu$ is independent of the nature of
370 $\partial C$ and is the content of Noether's first theorem [1, 2].

371 See [14, 15] for a proof of the theorem.

372 In the previous theorem we set $\partial C = \emptyset$. If $\partial C \neq \emptyset$, a Lagrangian symmetry need not be
373 Hamiltonian, even if one picks boundary conditions at the timelike boundary $\partial M = \partial C \times \mathbb{R}$
374 such that both $\Omega$ and $\int_C n_\mu j^\mu$ are independent of the choice of hypersurface $C \hookrightarrow M$. The

---

[9]Bar functional-analytic subtleties which are irrelevant for this argument.

[10]Note that any function on $\mathcal{F}_{\text{EL}}$ by construction defines a quantity constant along "time evolution",
simply in virtue of the fact that it associates one number to the *entire* spacetime "history" of an on-shell
field configuration. However, most of these constant quantities do *not* correspond to physically meaningful
*conserved* quantities.

mismatch is due to corner contributions supported on $\partial C$ that might spoil the Hamiltonian flow equation (18); cf. §3.2 and §4.5 for examples of this scenario.

In the case of Lagrangian symmetries, the extra input that selects dynamics-compatible actions is given by the condition that they leave the Lagrangian invariant up to boundary terms (20). This condition presupposes that its action is well defined *off*-shell and that it is local. These assumptions trickle down the construction and are ultimately responsible for the existence of an (off-shell) local expression of the charges, which in turn allow one to talk about their conservation in terms of their (on-shell) independence of their values from the choice of hypersurface $C \hookrightarrow M$.

From this perspective, the Noether current construction for local (in the sense laid out in §2.1) Lagrangian symmetries is a powerful trick to be able to solve the conditions $1, 2, 3$ of Definition 2.1 all at once. If one wishes to consider symmetries such that the Noether current construction is not available, finding a proper notion of current to build the charge becomes a non-trivial problem especially in relation to locality.

As we will now discuss, this is what happens when studying Poisson-Lie symmetries where one loses the Lagrangian symmetry prescription (20) and at times even the locality of the symmetry action and of its charges. We will circumvent these problems by modifying the points 2 and 3 of the Definition 2.1 of what constitutes a viable symmetry and charge is.

**Group-Valued Momentum Maps.** To motivate the deformation of Noetherian symmetries into PL symmetries, we now turn to the diffeological perspective on Hamiltonian actions on a phase space $(\mathcal{F}_{\mathrm{EL}}, \Omega)$ advanced in [45].

For this discussion it is convenient to introduce the following: given an (on-shell) action $\mathcal{Q}$, define the *variational current* $\mathbb{J}^\mu$ as the mixed form

$$\mathbb{J}^\mu :\approx \mathbb{i}_{\mathcal{Q}(\cdot)} \omega^\mu \in \Omega^{\dim(M)-1,1}(M \times \mathcal{F}_{\mathrm{EL}}) \otimes \mathfrak{g}^*, \tag{24}$$

as well as the *variational charge* $\mathbb{Q}$ as the field-space 1-form

$$\mathbb{Q} :\approx \mathbb{i}_{\mathcal{Q}(\cdot)} \Omega \approx \int_C n_\mu \mathbb{J}^\mu \in \Omega^1(\mathcal{F}_{\mathrm{EL}}) \otimes \mathfrak{g}^*. \tag{25}$$

Notice that these definitions are done at the level of mixed forms, not necessarily local ones.

If $\mathcal{Q}$ is a (local) Lagrangian action, then both quantities are conserved on-shell in virtue of $\mathcal{Q}$ preserving the shell and $\omega^\mu$ (or $\Omega$) being conserved. Nonetheless, both are *1-forms* over $\mathcal{F}_{\mathrm{EL}}$, and therefore fail to define a conserved current/charge, in the usual sense.

An action on $(\mathcal{F}_{\mathrm{EL}}, \Omega)$ is Hamiltonian if and only if its variational charge is $\mathbb{d}$-exact, $\mathbb{Q} = \mathbb{d}q$.

In [45], it was shown that momentum maps, i.e. Hamiltonian generators, can be defined in diffeology—namely under extremely mild hypotheses—by considering field-space line integrals of the variational charge $\mathbb{Q}$.

Indeed, integrating the $\mathbb{d}$-exact variational charge $\mathbb{Q}$ along any field-space path, symbolically denoted "$\varphi_0 \to \varphi$", between an arbitrary reference configuration[11] $\varphi_0$ and $\varphi$ in $\mathcal{F}_{\mathrm{EL}}$ one obtains:

$$\psi(\varphi_0, \varphi) :\approx \int_{\varphi_0 \to \varphi} \mathbb{Q} \quad \text{and} \quad \psi(\varphi_0, \varphi) :\approx q(\varphi) - q(\varphi_0). \tag{26}$$

---

[11]If $\mathcal{F}$ is the space of sections of a vector bundle, one can choose $\varphi_0 = 0$ if this is a solution of the equations of motion. More generally, one can might want to choose $\varphi_0$ to be a "vacuum solution", whenever this concept makes sense.

In [45], the map $\psi$ is called the *2-point momentum map* and the existence of its primitive $q$, uniquely defined up to a choice of reference $\varphi_0$ and a constant, follows from the cocycle properties of $\psi$ obtained from the concatenation of paths in its definition:

$$\psi(\varphi_1, \varphi_2) + \psi(\varphi_2, \varphi_3) = \psi(\varphi_1, \varphi_3) \quad \forall \varphi_1, \varphi_2, \varphi_3 \in \mathcal{F}_{\mathrm{EL}}. \tag{27}$$

From this point of view, the generalization to a PL momentum map can then be motivated by modifying the cocycle condition by introducing a non-Abelian group operation for what is now a *group-valued* (Poisson-Lie) 2-point momentum map $\Psi : \mathcal{F}_{\mathrm{EL}} \times \mathcal{F}_{\mathrm{EL}} \to \mathcal{G}^*$:

$$\Psi(\varphi_1, \varphi_2)\Psi(\varphi_2, \varphi_3) = \Psi(\varphi_1, \varphi_3) \quad \forall \varphi_1, \varphi_2, \varphi_3 \in \mathcal{F}_{\mathrm{EL}}. \tag{28}$$

The target (Lie) group denoted $\mathcal{G}^*$ can be thought of as a non-linear, and generally non-Abelian, generalization of the (Abelian) linear Lie *group* $(\mathfrak{g}^*, +)$ in which $\psi$ and $q$ are naturally valued.[12]

This generalization is equivalent to asking what kind of 1-forms $\mathbb{Q}$ generalize the fundamental theorem of calculus available for exact 1-forms $\mathbb{d}q$, meaning that their line integrals depend only on the initial and final points of the paths. Path ordered exponentials (Wilson lines) of a flat Lie-algebra valued 1-forms do exactly this. Then, the relevant *group*-valued 2-point momentum map is:

$$\Psi(\varphi_0, \varphi) :\approx \mathrm{Pexp} \int_{\varphi_0 \to \varphi} \mathbb{Q} \quad \text{and} \quad \Psi(\varphi, \varphi_0) :\approx Q(\varphi_0)^{-1} Q(\varphi). \tag{29}$$

The generalized $\mathbb{d}$-exactness condition then becomes

$$\mathbb{Q} \approx Q^{-1} \mathbb{d}Q \tag{30}$$

where the right hand side is the pullback by $Q : \mathcal{F}_{\mathrm{EL}} \to \mathcal{G}^*$ of the left Maurer-Cartan form on $\mathcal{G}^*$, and $Q$ is called the *left Poisson-Lie momentum map*. (Of course, a definition of a *right* Poisson-Lie momentum map can also be introduced, see below.)

From the definition of the variational charge, we see that the above is equivalent to asking that the following non-linear generalization of the Hamiltonian flow equation, which we call the *Poisson-Lie flow equation*, holds[13] [20, 22, 46]:

$$\mathbb{i}_{\mathbb{Q}(\alpha)}\Omega \approx \langle Q^{-1}\mathbb{d}Q, \alpha \rangle \quad \forall \alpha \in \mathfrak{g}. \tag{31}$$

This equation shows in particular that as a vector space[14]

$$\mathrm{Lie}(\mathcal{G}^*) = \mathfrak{g}^*. \tag{32}$$

We anticipate here that not any Lie algebra structure $[\cdot, \cdot]_*$ on $\mathfrak{g}^*$ will do. Reviewing which such structures are compatible with the Poisson-Lie flow equation (31) is the topic of §2.4.

In the remainder of this section, we will focus instead on what conditions are needed to turn this non-linear version of (kinematical) Hamiltonian actions to a generalization of the (dynamically meaningful) Noetherian symmetries given in Definition 2.1.

---

[12]Note that $\mathfrak{g}^*$ and the space of left-invariant 1-forms on the symmetry group $\mathcal{G}$, $\Omega^1_{\mathrm{left}}(\mathcal{G})$, are naturally isomorphic whenever both are defined, although the $\Omega^1_{\mathrm{left}}(\mathcal{G})$ has larger applicability in diffeology. Whereas the dual of a Lie algebra $\mathfrak{g}^*$ can be defined in diffeological satisfactory terms as $\Omega^1_{\mathrm{left}}(\mathcal{G})$, we don't know at this stage if a diffeologically meaningful definition of $\mathcal{G}^*$ can be given.

[13]Using Poisson brackets: $\mathbb{Q} = Q^{-1}\{Q \overset{\otimes}{,} \cdot\}$ or, more explicitly, $\mathbb{Q}(\xi)f = (\Omega_{IJ})^{-1}\langle Q^{-1}\partial_I Q, \xi \rangle \partial_J f$ [22].

[14]For a mathematically rigorous approach to Poisson-Lie symmetries in the infinite dimensional context see [47], where a series of interesting and non-trivial *Abelian* Poisson-Lie symmetries related to the (horizontal) automorphisms of certain symplectic bundles is also covered in detail. The viewpoint developed there is however detached from an action principle of a Lagrangian field theory.

$_{438}$ **Dynamical Poisson-Lie Symmetry & (Non-)Locality.** The goal of this section is to modify
$_{439}$ Definition 2.1 in a way that can encompass non-trivial Poisson-Lie (PL) symmetries.

$_{440}$ First, we observe that in view of Theorem 2.3 non-trivial PL symmetries are in general
$_{441}$ not Lagrangian. This is because the latter always lead to Hamiltonian, rather than PL
$_{442}$ charges. To highlight the difference between the two cases, we note that PL actions, as
$_{443}$ opposed to Hamiltonian actions, are *not* symplectomorphisms. Indeed, from (31) and the
$_{444}$ $\mathbb{d}$-closure (in fact, exactness) of $\Omega$ it follows that

$$\mathbb{L}_{\mathbb{Q}(\alpha)}\Omega \approx -\frac{1}{2}\langle [Q^{-1}\mathbb{d}Q, Q^{-1}\mathbb{d}Q]_*, \alpha\rangle \tag{33}$$

$_{445}$ which does not vanish unless $\mathcal{G}^*$ is Abelian.

$_{446}$ (This said, if $\partial C \neq \emptyset$, there is a loophole in the above conclusions, namely a Lagrangian
$_{447}$ action might happen to be Hamiltonian. Indeed, it could so happen that the Noether
$_{448}$ charge $\int_C n_\mu j^\mu$ of a certain Lagrangian symmetry vanishes while at the same time the
$_{449}$ corner obstruction to Theorem 2.3 happens to take the form $Q^{-1}\mathbb{d}Q$. This somewhat
$_{450}$ fortunate situation describes precisely what happens in certain examples, cf. §3.2 and
$_{451}$ §4.5.)

$_{452}$ This is why here we shall focus on adapting the Definition 2.1 of Noetherian symmetries,
$_{453}$ rather than that of a Lagrangian symmetry. The obvious modification concerns replacing
$_{454}$ point 2, which can be rephrased as $\mathbb{Q} \approx \mathbb{d}q$, with its PL analogue, namely $\mathbb{Q} \approx Q^{-1}\mathbb{d}Q$.
$_{455}$ Less obvious is that point 3, which is about the locality and thus the conservation of the
$_{456}$ charge, must also be adapted.

$_{457}$ Indeed, there is another way in which PL actions are expected to markedly lie outside of
$_{458}$ the Noetherian framework: namely one also expects PL actions and their generators to
$_{459}$ be *non-local*; a related fact is that some examples also show that dynamically relevant PL
$_{460}$ actions are sometimes best defined on-shell.[15]

$_{461}$ An action is Hamiltonian if and only if its variational charge is $\mathbb{d}$-exact, $\mathbb{Q} = \mathbb{d}q$, and it is
$_{462}$ left PL if and only if its variational charge satisfies the "flatness" equation:[16]

$$\mathbb{d}\mathbb{Q} + \frac{1}{2}[\mathbb{Q}, \mathbb{Q}]_* = 0. \tag{34}$$

$_{463}$ This equation, which is responsible for the right hand side of (33), can be used to argue
$_{464}$ that, in field theory, there is a tension between PL symmetries and locality. Indeed, assume
$_{465}$ that $\mathbb{Q}$ were local. Then, since $\Omega$ is by construction a local functional of the fields, so would
$_{466}$ be $\mathbb{Q}$. Consider now equation (33) (or, equivalently, (34)): on its left-hand side we have
$_{467}$ *one* integral over $C$ of a local 2-form on $\mathcal{F}$ evaluated at an on-shell configuration, while
$_{468}$ on its right-hand side we find a Lie bracket of *two* integrals of local 1-forms over $C$.

$_{469}$ Beside the somewhat trivial cases where $\dim(C) = 0$ or $\mathfrak{g}^*$ is Abelian, we are aware of
$_{470}$ two possible resolutions of this tension. In the first scenario, the action $\mathbb{Q}$ is itself non-
$_{471}$ local; this is what happens in the second-order Klimčík-Ševera model discussed in §4, see
$_{472}$ especially §4.3.2. In the second scenario, the integral defining $\mathbb{Q}$ localizes at a point of $C$;
$_{473}$ this is what happens in the first-order Klimčík-Ševera model discussed in §4.5, *which is*
$_{474}$ *however not manifestly spacetime covariant being tied to a choice of foliation of $M$.* In
$_{475}$ this case, although everything appears local, it is the physical interpretation of the charge

---

[15]These latter expectations are not strictly speaking necessary features, but one of the goals of the paper
is to get a better grasp on them, through examples. See §4 and in particular the discussion around (212).

[16]For a right PL action, the relative sign in the flatness equation is flipped.

$Q$ that is in tension with locality, since then what is naturally interpreted as a (deformed) "*total* momentum" of the Klimčík-Ševera string with respect to a given global symmetry is captured by the value of the worldsheet fields at a *single* point of the string. In this scenario this is possible because, in passing from the second- to the first order formulation, one employs a non-local field redefinition. In §5, a similar point-localization of $\mathbb{Q}$ and $Q$ happens in a 2+1 dimensional model, this time thanks to the non-locality intrinsic to the constrained (and reduced) phase space of a gauge theory (cf. remark on locality from §2.1).

This remark emphasizes that a very delicate balance between local and non-local features is needed to construct field theories with non-trivial PL symmetries. This will be exemplified in the two definitions below

**Definition 2.4.** Let $F \to M$ be a fiber bundle over $M \simeq C \times \mathbb{R}$, and $\iota : C \hookrightarrow M$ any embedded codimension-1 (Cauchy) surface. Denote $F_C \to C$ the pullback bundle of $F \to M$ along $\iota$ and $\mathcal{F}_C := \Gamma(C, F_C)$ the space of its sections. The jet space $J_t^r(\mathbb{R}, \mathcal{F}_C)$ is the space of equivalence classes of smooth maps $\gamma : \mathbb{R} \to \mathcal{F}_C$ with the same Taylor expansion at $t \in \mathbb{R}$ up to order $r$. All embeddings give rise to isomorphic jet spaces $J_0^r(\mathbb{R}, \mathcal{F}_C)$, which we abstractly denote $\mathcal{P}^{(r)}$. Finally, we introduce the evaluation map:

$$j_1^r : \{C\} \times \mathcal{F} \to \{C\} \times \mathcal{P}^{(r)}, \quad (C, \varphi) \mapsto (C, \varphi|_C, \partial_n \varphi|_C, \partial_n^2 \varphi|_C, \dots, \partial_n^r \varphi|_C),$$

where $\{C\}$ is the set of embedded Cauchy surfaces in $(M, \gamma_{\mu\nu})$ and $\partial_n$ is the derivative along $n_C^\mu$ the unit (future pointing) normal to $C$ in $M$. ◇

**Definition 2.5** (Poisson-Lie symmetries)**.** Given a local field theory $(F \to M, \mathcal{L})$, a *left Poisson-Lie symmetry* over the CPS $(\mathcal{F}_{\text{EL}}, \Omega)$ is

1′. a smooth Lie algebra *action*

$$\mathbb{Q} : \mathfrak{g} \to \mathfrak{X}^1(\mathcal{F}_{\text{EL}}),$$ (35)

2′. whose flows are *left Poisson-Lie*,

$$\mathbb{i}_{\mathbb{Q}(\cdot)}\Omega \approx Q^{-1}\mathbb{d}Q,$$ (36)

   for a (group-valued) momentum map denoted $Q \in \Omega^0(\mathcal{F}) \otimes \mathcal{G}^*$,

3′. such that there exists an $r < \infty$ and a map $\check{Q} \in \text{Maps}(\{C\} \times \mathcal{P}^{(r)}, \mathcal{G}^*)$ that satisfies for *all* embedded $C \hookrightarrow M$ the equation

$$Q(\cdot) \approx \left((j_1^r)^* \check{Q}\right)(C, \cdot).$$ (37)

Right Poisson-Lie symmetries are defined analogously, by replacing $Q^{-1}\mathbb{d}Q$ with $\mathbb{d}QQ^{-1}$ in point 2′. ◇

Note that each $\varphi \in \mathcal{F}_{\text{EL}}$ can be seen as an element of $\mathcal{F} = \Gamma(M, F)$. Point 3′ says that the charge $Q$ can always be written as a function of $\varphi$ restricted to $C \subset M$ and a finite number of its normal derivatives:

$$Q(\varphi) = \check{Q}(C, \varphi|_C, \underbrace{\partial_n \varphi|_C, \dots}_{r})$$ (38)

This function is not required to be local over $C$ and it can depend on geometric data of the embedding $C \hookrightarrow M$ (e.g. induced metric, extrinsic curvature, etc.).

The formulation of point $3'$ of this definition generalizes that of point 3 above in terms of local forms over $M$. This is meant to balance a potential non-local expression "over $C$" with the requirement of "time locality", or dependence on the choice of $C$, which is necessary to even talk about a notion of conservation.

**Remark.** This formulation is still somewhat convoluted and therefore not fully satisfactory. We think that a more geometric treatment might be given using Dirac's hypersurface deformation algebra (HDA) [48, 49]. For *non* general-covariant theories like those studied in this manuscript, this formulation would require an extension of the phase space to include "embedding (or, surface) variables" as fields—that is, it would require writing the field theory in a parametrized form. The advantage of this formulation is to avoid talking about the locality of $Q$ and simply checking its conservation, i.e. $C$-independence, by means of its Poisson-algebra with the HDA generators. We postpone the investigation of this formulation to future work.

## 2.4 Poisson-Lie Groups, Their Actions, & Their Doubles

In this section we review and recall standard material on Poisson-Lie groups, Poisson-Lie actions, the Heisenberg and Drinfel'd doubles, and dressing transformations. The discussion is self-contained. The reader familiar with these topics can safely skip this section after familiarizing themselves with our notation.

**Poisson-Lie Groups & Poisson Actions.** We start off by defining Poisson-Lie groups, i.e. Lie groups equipped with a Poisson structure compatible with group multiplication:

**Definition 2.6** (Poisson-Lie Group)**.** A Lie group $\mathcal{G}$ is called a *Poisson-Lie group* if it is also a Poisson manifold such that the group multiplication $m : \mathcal{G} \times \mathcal{G} \to \mathcal{G}$ is a Poisson map, where $\mathcal{G} \times \mathcal{G}$ is equipped with the product Poisson structure. In this case we say that the Poisson structure on $\mathcal{G}$ is *multiplicative*. In terms of a Poisson bivector on $\mathcal{G}$, $\pi_{\mathcal{G}} \in \mathfrak{X}^2(\mathcal{G})$, the Poisson structure on $\mathcal{G}$ is multiplicative if and only if,

$$\pi_{\mathcal{G}}(gh) = (l_g)_* \pi_{\mathcal{G}}(h) + (r_h)_* \pi_{\mathcal{G}}(g), \ \forall g, h \in \mathcal{G} \tag{39}$$

where $l_g$ and $r_h$ denote respectively the left and right translations in $\mathcal{G}$ by $g$ and $h$, and $(l_g)_*$, $(r_h)_*$ their linearizations extended to multivector fields. $\diamond$

It is not hard to check that this multiplicative condition implies all Poisson-Lie groups have degenerate Poisson brackets at the identity $1 \in \mathcal{G}$:

$$\pi_{\mathcal{G}}(1) = 0. \tag{40}$$

In particular, Poisson-Lie groups are *never* themselves symplectic.

**Example 2.7.** The Poisson structure on the Abelian Lie group $\mathcal{G} = (\mathbb{R}^3, +)$ given by

$$\{x_i, x_j\} = \epsilon_{ijk} x_k \iff \pi_{\mathcal{G}}(df_1, df_2)(x) = \{f_1, f_2\}(x) = \vec{x} \cdot (\vec{\nabla} f_1 \times \vec{\nabla} f_2), \tag{41}$$

for all $f_1, f_2 \in C^\infty(\mathbb{R}^3)$, and $x \in \mathbb{R}^3$, is multiplicative and therefore $(\mathbb{R}^3, +, \{\cdot, \cdot\})$ is a Poisson-Lie group. The multiplicative property (39) can be checked explicitly. Let $x, y \in \mathbb{R}^3$; then the left hand side of (39) is given by

$$\pi_{\mathcal{G}}(df_1, df_2)(x+y) = \{f_1, f_2\}(x+y) = \vec{z} \cdot (\vec{\nabla} f_1 \times \vec{\nabla} f_2)\Big|_{z=x+y} \tag{42}$$

543  and the right hand side by

$$(l_x)_* \pi_{\mathcal{G}}(df_1, df_2)(y) = \pi_{\mathcal{G}}(d(f_1 \circ l_x), d(f_2 \circ l_x))(y) = \left\{ f_1^{(x)}, f_2^{(x)} \right\}(y)$$
$$= \vec{z} \cdot (\vec{\nabla} f_1^{(x)} \times \vec{\nabla} f_2^{(x)}) \Big|_{z=y} \qquad (43)$$

$$(r_y)_* \pi_{\mathcal{G}}(df_1, df_2)(x) = \pi_{\mathcal{G}}(d(f_1 \circ r_y), d(f_2 \circ r_y))(x) = \left\{ f_1^{(y)}, f_2^{(y)} \right\}(x)$$
$$= \vec{z} \cdot (\vec{\nabla} f_1^{(y)} \times \vec{\nabla} f_2^{(y)}) \Big|_{z=x} \qquad (44)$$

544  where we denoted $(f \circ l_x)(z) = f(x + z) =: f^{(x)}(z)$ and $(f \circ r_y)(z) = f(z + y) =: f^{(y)}(z)$.
545  The conclusion follows noting that e.g. $(\vec{\nabla} f^{(x)})(y) = (\vec{\nabla} f)(x + y)$. Therefore, (39) is
546  satisfied and $\mathcal{G} = (\mathbb{R}^3, +)$ is a Poisson-Lie group when endowed with the Poisson bivector
547  $\pi_{\mathcal{G}}$ given above.                                                                                      $\diamond$

548  As the infinitesimal version of a Lie group is a Lie algebra, the infinitesimal version of a
549  Poisson-Lie group is a *Lie bialgebra* $(\mathfrak{g}, \mathfrak{g}^*)$, where $\mathfrak{g}^*$ is the vector space dual of $\mathfrak{g}$ endowed
550  with a Lie bracket $[\cdot, \cdot]_*$ derived from the linearization of the Poisson bivector $\pi_{\mathcal{G}}$ at the
551  identity $1 \in \mathcal{G}$. Namely, recalling the earlier observation that $\pi_{\mathcal{G}}(1) = 0$, and denoting its
552  linearization at the identity by $\delta$,

$$\delta := d\pi_{\mathcal{G}}|_{g=1} : \mathfrak{g} \mapsto \mathfrak{g} \wedge \mathfrak{g}, \qquad (45)$$

553  one defines $[\cdot, \cdot]_* : \mathfrak{g}^* \wedge \mathfrak{g}^* \to \mathfrak{g}^*$ by dualizing $\delta$, i.e. demanding that for all $\alpha \in \mathfrak{g}$ and
554  $X_1, X_2 \in \mathfrak{g}^*$

$$\langle \alpha, [X_1, X_2]_* \rangle := \langle \delta(\alpha), X_1 \otimes X_2 \rangle. \qquad (46)$$

555  The Jacobi identity together with the multiplicativity condition for $\pi_{\mathcal{G}}$ then implies that
556  $\delta$ is a (Chevalley-Eilenberg) cocycle of $\mathfrak{g}$, $\delta \in (Z^1_{\mathrm{ad}}(\mathfrak{g}, \mathfrak{g} \otimes \mathfrak{g}), \partial_{\mathrm{CE}})$ [23, Def.8.1.1], or equiv-
557  alently that $(\mathfrak{g}^*, [\cdot, \cdot]_*)$ is a Lie algebra [50, Thm. 2.18].

558  In practice, one often considers a special class of cocycles which are coboundary, namely,
559  they are characterized in terms of an $r$-matrix in $\mathfrak{g} \otimes \mathfrak{g}$, i.e. $\delta = \partial_{\mathrm{CE}} r = [r, \cdot]$ [24, Ch.2].

560  Like regular Lie groups, it is fruitful to consider the action of Poisson-Lie groups on various
561  spaces. Of particular interest to physics is the action of Poisson-Lie groups on symplectic
562  or Poisson manifolds.

563  **Definition 2.8** (Poisson Action)**.** The left action $\sigma : \mathcal{G} \times P \to P$ of a Poisson-Lie group
564  $\mathcal{G}$ on a Poisson manifold $P$ is called a *Poisson action* if $\sigma$ is a Poisson map, where the
565  manifold $\mathcal{G} \times P$ has the product Poisson structure. Similarly for a right action.           $\diamond$

566  From any action we may define the induced maps $\sigma_p : \mathcal{G} \to P, \sigma_p(g) := \sigma(g, p) = gp$
567  and $\sigma_g : P \to P, \sigma_g(p) := \sigma(g, p) = gp$. The corresponding induced Lie algebra action
568  $\mho : \mathfrak{g} \to \mathfrak{X}(P)$ can then be defined as

$$\mho(\alpha)(p) = (\sigma_p)_*(\alpha) \quad \forall \alpha \in \mathfrak{g}, p \in P, \qquad (47)$$

569  where $(\sigma_p)_* := d\sigma_p|_{g=1} : \mathfrak{g} \to T_p P$.

570  This leads to the following Theorems which provide a Poisson-Lie group theoretic origin
571  for the analogous field space statements presented in §2.3.

572  **Theorem 2.9** (Lu and Weinstein [20])**.** Given a Poisson-Lie group $\mathcal{G}$ and a Poisson manifold
573  $P$, respectively with Poisson bivectors $\pi_{\mathcal{G}}$ and $\pi_P$, the following conditions are equivalent:

574      1. $\sigma : \mathcal{G} \times P \to P$ is a Poisson action;

575      2. for all $g \in \mathcal{G}$ and $p \in P$

$$\pi_P(gp) = (\sigma_g)_* \pi_P(p) + (\sigma_p)_* \pi_{\mathcal{G}}(g); \tag{48}$$

576      3. assuming $\mathcal{G}$ is connected, then for each $\alpha \in \mathfrak{g}$ we have,

$$\mathbb{L}_{\mathbb{O}(\alpha)} \pi_P = (\mathbb{O} \wedge \mathbb{O}) \delta(\alpha); \tag{49}$$

577      4. assuming that $\mathcal{G}$ is connected, then for any 1-forms $f_1, f_2 \in \Omega^1(P)$ we have,

$$(\mathbb{L}_{\mathbb{O}(\alpha)} \pi_P)(f_1, f_2) = \langle [X_{f_1}, X_{f_2}]_*, \alpha \rangle \tag{50}$$

578      where $X_f$ is the $\mathfrak{g}^*$-valued function on $P$ defined by

$$\langle X_f, \alpha \rangle = \langle f, \mathbb{O}(\alpha) \rangle, \; \alpha \in \mathfrak{g} \tag{51}$$

579      and $[X_{f_1}, X_{f_2}]_*$ denotes the pointwise bracket in $\mathfrak{g}^*$.

580      $\diamond$

581 **Theorem 2.10** (Lu [50, Theorem 3.7], Babelon and Bernard [22])**.** A left (right) action $\mathbb{O}$
582 on a symplectic manifold $(P, \Omega)$ is Poisson if and only if $\mathbb{Q} := \mathring{\mathbb{i}}_{\mathbb{O}(\cdot)} \Omega$ is such that

$$\mathbb{d}\mathbb{Q} + \frac{1}{2}[\mathbb{Q}, \mathbb{Q}]_* = 0 \qquad (\text{or } \mathbb{d}\mathbb{Q} - \frac{1}{2}[\mathbb{Q}, \mathbb{Q}]_* = 0), \tag{52}$$

583 that is, if and only if locally there exists a $Q$ such that $\mathbb{Q} = Q^{-1}\mathbb{d}Q$ (or $\mathbb{Q} = \mathbb{d}QQ^{-1}$,
584 respectively). $\diamond$

585 Note that this theorem generalizes the statement that an action $\mathbb{O}$ on $(P, \Omega)$ is symplectic,
586 namely $\mathbb{L}_{\mathbb{O}(\alpha)}\Omega = 0$ for all $\alpha \in \mathfrak{g}$ (cf. (49)), if and only if $\mathbb{Q} := \mathring{\mathbb{i}}_{\mathbb{O}(\cdot)}\Omega$ is $\mathbb{d}$-closed, that is, if
587 and only if locally there exists a $q$ such that $\mathbb{Q} = \mathbb{d}q$.

588 **Example 2.11.** Let $\mathcal{G}$ be a Lie group with corresponding Lie algebra $\mathfrak{g}$. Then the action of
589 $\mathfrak{g}^*$ in the right trivialization of the cotangent bundle $T^*\mathcal{G}$ by $(Y, (g, X_g)) \mapsto (g, X_g + l^*_{g^{-1}}Y)$
590 for $Y \in \mathfrak{g}^*, g \in \mathcal{G}$ and $X_g \in T^*_g\mathcal{G}$ is a Poisson action. In a left trivialization of $T^*\mathcal{G}$, this
591 action becomes $(Y, (g, X)) \mapsto (g, X + Y)$. Note that for $\mathfrak{g} = \mathfrak{su}(2)$, this is related to
592 Example 2.7. $\diamond$

593 It remains to elucidate one last piece of formalism inherent to Poisson-Lie groups which
594 we will need for their realization as symmetries in field theory. That is, the notion of
595 Heisenberg and Drinfel'd doubles. In a nutshell, the Heisenberg double is a group $\mathcal{D}$
596 equipped with a symplectic structure, while the Drinfel'd double is the same group $\mathcal{D}$ now
597 equipped with a (degenerate) Poisson-Lie structure, such that the action of the latter on
598 the former is Poisson.

599 **The Heisenberg & Drinfel'd Doubles, & Dressing Actions.** The notion of doubles are
600 a natural framework to characterize Poisson-Lie symmetries. This is in part because
601 given any Poisson-Lie group $\mathcal{G}$, there is a one-to-one correspondence between the induced
602 classical doubles and the induced Lie bialgebra structures it admits [24]. In this section,
603 we review standard material about such objects, which can be found in many references
604 such as [50], [24], and [23].

Let us consider a real finite dimensional Lie algebra $\mathfrak{g}$ and its dual Lie algebra $\mathfrak{g}^*$. On $\mathfrak{g}$ and $\mathfrak{g}^*$, we pick dual bases $\{\tau_a\}$ and $\{\tau_*^a\}$ respectively, satisfying $\langle \tau_*^a, \tau_b \rangle = \delta_b^a$, $[\tau_a, \tau_b] = c_{ab}{}^c \tau_c$, and $\left[\tau_*^a, \tau_*^b\right]_* = \tilde{c}^{ab}{}_c \tau_*^c$.

Because of the duality between $\mathfrak{g}$ and $\mathfrak{g}^*$ (note, $\mathfrak{g}^{**} \cong \mathfrak{g}$), we have both the co-adjoint action $\mathsf{ad}^*$ of $\mathfrak{g}$ on $\mathfrak{g}^*$ and the (dual) co-adjoint action $\widetilde{\mathsf{ad}}^*$ of $\mathfrak{g}^*$ on $\mathfrak{g}$. Thanks to this pair of actions, we can build a bracket on $\mathfrak{d} := \mathfrak{g} \oplus \mathfrak{g}^*$:[17]

$$[\alpha, \beta]_{\mathfrak{d}} := [\alpha, \beta], \quad [X, Y]_{\mathfrak{d}} := [X, Y]_* \quad [\alpha, X]_{\mathfrak{d}} := -\mathsf{ad}_\alpha^* X + \widetilde{\mathsf{ad}}_X^* \alpha, \quad \forall \alpha, \beta \in \mathfrak{g}, X, Y \in \mathfrak{g}^*. \tag{53}$$

This bracket is a Lie bracket if and only if the structure constants of $\mathfrak{g}$ and $\mathfrak{g}^*$ satisfy the compatibility condition

$$c_{ab}{}^c \tilde{c}^{de}{}_c = c_{ac}{}^d \tilde{c}^{ce}{}_b + c_{ac}{}^e \tilde{c}^{dc}{}_b - c_{bc}{}^d \tilde{c}^{ce}{}_a - c_{bc}{}^e \tilde{c}^{dc}{}_a \tag{54}$$

A theorem of Manin (see [50, Thm.2.22] or [24, Prop.1.3.4 and Lemma 1.3.5]) guarantees that equation (54) is *always* satisfied for $(\mathfrak{g}, \mathfrak{g}^*)$ a Lie bialgebra, along with the existence of an adjoint invariant symmetric bilinear pairing $\langle \cdot, \cdot \rangle : \mathfrak{d} \times \mathfrak{d} \to \mathbb{R}$ on $\mathfrak{d} := \mathfrak{g} \oplus \mathfrak{g}^*$ for which $\mathfrak{g}$ and $\mathfrak{g}^*$ are isotropic.[18] This leads to the following:

**Definition 2.12** (Classical Double)**.** Let $(\mathfrak{g}, \mathfrak{g}^*)$ be a Lie bialgebra. The Lie algebra structure on $\mathfrak{d} := \mathfrak{g} \oplus \mathfrak{g}^*$ defined by (53,54) along with a non-degenerate symmetric adjoint invariant bilinear pairing $\langle \cdot, \cdot \rangle : \mathfrak{d} \times \mathfrak{d} \to \mathbb{R}$ for which $\mathfrak{g}$ and $\mathfrak{g}^*$ are isotropic is called the *classical (Drinfel'd) double* and is denoted

$$\mathfrak{d} := \mathfrak{g} \bowtie \mathfrak{g}^* \cong \mathfrak{g}^* \bowtie \mathfrak{g}. \tag{55}$$

$\diamond$

The notation $\bowtie$ generalizes that of the semidirect products $\ltimes$ and $\rtimes$, emphasizing that each subalgebra carries a (non-trivial) action on the other. We will use the symbols $\triangleright$ and $\triangleleft$ to encode the different left and right actions[19] of $\mathfrak{g}$ on $\mathfrak{g}^*$ and vice versa. For all $\alpha, \beta \in \mathfrak{g}, X, Y \in \mathfrak{g}^*$, denote

$$\alpha \triangleright X := [\alpha, X]_{\mathfrak{d}}\big|_{\mathfrak{g}^*} = -\mathsf{ad}_\alpha^* X \quad \text{and} \quad \alpha \triangleleft X := [\alpha, X]_{\mathfrak{d}}\big|_{\mathfrak{g}} = \widetilde{\mathsf{ad}}_X^* \alpha \tag{56a}$$

where $\bullet_{|\mathfrak{g}}$ denotes the projection of the vector space $\mathfrak{d} = \mathfrak{g} \oplus \mathfrak{g}^*$ on the subspace $\mathfrak{g}$ etc. And similarly,

$$X \triangleleft \alpha := [X, \alpha]_{\mathfrak{d}}\big|_{\mathfrak{g}^*} = \mathsf{ad}_\alpha^* X \quad \text{and} \quad X \triangleright \alpha := [X, \alpha]_{\mathfrak{d}}\big|_{\mathfrak{g}} = -\widetilde{\mathsf{ad}}_X^* \alpha. \tag{56b}$$

Finally, note that by exponentiating the infinitesimal actions to a finite flow, setting $(h, \ell) = (e^\alpha, e^X)$, one finds

$$X \triangleleft h = h^{-1} \triangleright X = \mathsf{Ad}_h^* X \quad \text{and} \quad \alpha \triangleleft \ell = \ell^{-1} \triangleright \alpha = \widetilde{\mathsf{Ad}}_\ell^* \alpha. \tag{56c}$$

**Theorem 2.13** (Drinfel'd [24, Theorem 1.3.2])**.** If $(\mathcal{G}, \pi_{\mathcal{G}})$ is a Poisson-Lie group, then its linearization at $1 \in \mathcal{G}$ defines the classical double $\mathfrak{d} = \mathfrak{g} \bowtie \mathfrak{g}^*$, as per equations (45), (46), and Definition 2.12. Conversely, if $\mathcal{G}$ is connected and simply connected, then every classical double $\mathfrak{d} = \mathfrak{g} \bowtie \mathfrak{g}^*$ over $\mathfrak{g}$ defines a unique multiplicative Poisson structure $\pi_{\mathcal{G}}$ on $\mathcal{G}$ such that $(\mathfrak{g}, \mathfrak{g}^*)$ is the linearization of the Poisson-Lie group $(\mathcal{G}, \pi_{\mathcal{G}})$. $\diamond$

---

[17]The signs in the last formula can be checked as follows: $[\alpha, \bullet]_{\mathfrak{d}}$ and $[\bullet, X]_{\mathfrak{d}}$ are a left and a right action, respectively, thus so are $-\mathsf{ad}_\alpha^*$ and $\widetilde{\mathsf{ad}}_X^*$. This is because $\mathsf{ad}$ are by definition left actions and $\mathsf{ad}^*$ is its transpose. (Recall: with this convention, the co-adjoint *representation* is given by $-\mathsf{ad}^*$.)

[18]"Isotropic" means that $\langle \mathfrak{g}, \mathfrak{g} \rangle = \langle \mathfrak{g}^*, \mathfrak{g}^* \rangle = 0$.

[19]To keep the notation uncluttered, we use the same symbols in different cases, as the context should make it clear what acts on what and from which side.

Since the classical double $\mathfrak{d} = \mathfrak{g} \bowtie \mathfrak{g}^*$ is a Lie algebra with $\mathfrak{g}$ and $\mathfrak{g}^*$ as subalgebras, asumming that $\mathcal{G}$ is connected and $\mathcal{G}^*$ is connected and simply connected, then $\mathfrak{d}$ can be exponentiated to yield a unique connected and simply connected Lie group $\mathcal{D}$ which has $\mathcal{G}$ and $\mathcal{G}^*$ as subgroups. For simplicity, we assume that every element in $\mathcal{D}$ can be globally written as a product of elements in $\mathcal{G}$ and $\mathcal{G}^*$.[20] This tells that $\mathcal{D} \simeq \mathcal{G} \times \mathcal{G}^*$ and, since there are two ways of multiplying two group elements together, i.e. from the left and the right,

$$\forall G \in \mathcal{D} \quad \exists \ell, \tilde{\ell} \in \mathcal{G}^* \text{ and } h, \tilde{h} \in \mathcal{G} \text{ such that } G = \ell h = \tilde{h}\tilde{\ell}. \tag{57}$$

This two-fold decomposition allows one to define the *dressing actions*:[21]

$$\ell \rhd h := \tilde{h} \quad \text{and} \quad \ell \lhd h := \tilde{\ell}. \tag{58}$$

whence the notation

$$\mathcal{D} \simeq \mathcal{G}^* \bowtie \mathcal{G}. \tag{59}$$

Similarly, flipping the order of $\mathcal{G}^*$ and $\mathcal{G}$, $\mathcal{D} \simeq \mathcal{G} \bowtie \mathcal{G}^*$, one finds:

$$\tilde{h} \lhd \tilde{\ell} := h \quad \text{and} \quad \tilde{h} \rhd \tilde{\ell} := \ell. \tag{60}$$

To show that the actions $\lhd$ and $\rhd$ introduced here are consistent with those introduced earlier, it is enough to show that expanding the above expressions for $(h, \ell) = (e^\alpha, e^X)$ close to the identity, one recovers $[\alpha, X]_\mathfrak{d} = \alpha \rhd X + \alpha \lhd X$ (cf. equation (56)). To do so, notice that from the definitions above

$$h\ell = (h \rhd \ell)(h \lhd \ell). \tag{61}$$

Consider now the expansion $h = 1 + \alpha + \frac{1}{2}\alpha^2 + \ldots$ and $\ell = 1 + X + \frac{1}{2}X^2 + \ldots$ (done as if $\mathcal{D} = \mathcal{G} \bowtie \mathcal{G}^*$ was a matrix group). Then, using that $h \lhd 1 = h$ and $\ell \rhd 1 = 1$ etc and therefore that $\alpha \lhd 1 = \alpha$ and $X \rhd 1 = 0$ etc, a little algebra shows that the expansion of the left and right hand sides of the above equation yields the following expansion in the double:

$$1 + X + \alpha + \frac{1}{2}X^2 + \frac{1}{2}\alpha^2 + \alpha X + \ldots \tag{62}$$
$$= \left(1 + X + \frac{1}{2}X^2 + \alpha \rhd X + \ldots\right)\left(1 + \alpha + \frac{1}{2}\alpha^2 + \alpha \lhd X + \ldots\right)$$

whence, simplifying, we obtain at second order

$$[\alpha, X]_\mathfrak{d} \equiv \alpha X - X\alpha = \alpha \rhd X + \alpha \lhd X. \tag{63}$$

---

[20]See [20] for sufficient conditions that guarantee that this is the case.

[21]That this is an action, namely that $\ell_1 \rhd (\ell_2 \rhd h) = (\ell_1 \ell_2) \rhd h$ etc, can be proved as follows. Let $G = \ell_1 \ell_2 h \in \mathcal{D}$; using (58), compute in $\mathcal{D}$:

$$\ell_1 \ell_2 h = \underbrace{\big((\ell_1 \ell_2) \rhd h\big)}_{\in \mathcal{G}} \underbrace{\big((\ell_1 \ell_2) \lhd h\big)}_{\in \mathcal{G}^*}$$

and

$$\ell_1 \ell_2 h = \ell_1 (\ell_2 \rhd h)(\ell_2 \lhd h) = \underbrace{\big(\ell_1 \rhd (\ell_2 \rhd h)\big)}_{\in \mathcal{G}} \underbrace{\big(\ell_1 \lhd (\ell_2 \rhd h)\big)(\ell_2 \lhd h)}_{\in \mathcal{G}^*}.$$

Comparing these two expressions, and using the uniqueness of the decomposition of $\mathcal{D}$ into $\mathcal{G} \times \mathcal{G}^*$ and focusing on the projection on the left $\mathcal{G}$ factor, we find the sought result that $(\ell_1 \ell_2) \rhd h = \ell_1 \rhd (\ell_2 \rhd h)$.

The double $\mathcal{D}$ as a group can be equipped with two natural Poisson structures, one which makes $\mathcal{D}$ a Poisson-Lie group, called the *Drinfel'd* double $\mathcal{D}_D$, and another which makes $\mathcal{D}$ a symplectic space, called the *Heisenberg* double $\mathcal{D}_H$ [51]. The symplectic structure $\Omega$ on $\mathcal{D}_H$ is often named after Semenov-Tian-Shanski (STS) and reads

$$\Omega = \frac{1}{2}\Big( \langle \mathrm{d}\ell\ell^{-1} \wedge \mathrm{d}\tilde{h}\tilde{h}^{-1} \rangle + \langle \tilde{\ell}^{-1}\mathrm{d}\tilde{\ell} \wedge h^{-1}\mathrm{d}h \rangle \Big). \tag{64}$$

An expression of the multiplicative Poisson bivector on $\mathcal{D}_D$ can be found, e.g., in Proposition 2.34 of [50].

There are then two natural actions to consider, being either the left $L$ or right $R$ multiplication of $\mathcal{D}$ on itself. Remarkably, these are Poisson actions (Definition 2.8):

$$R, L : \mathcal{D}_D \times \mathcal{D}_H \to \mathcal{D}_H, \tag{65}$$

inducing the left and right infinitesimal actions:

$$\mathbb{r}, \mathbb{l} : \mathfrak{d} \to \mathfrak{X}(\mathcal{D}_H). \tag{66}$$

In particular these actions subsume the dressing transformations defined above. Infinitesimally, assuming a matrix group and denoting $h_0 = 1 + \alpha + \dots$ and $\ell_0 = 1 + Y + \dots$, one finds (recall, $G = \ell h = \tilde{h}\tilde{\ell}$):[22]

$$G \mapsto Gh_0 \implies \delta_\alpha^R G = G\alpha \implies \delta_\alpha^R(\ell, h) = (0, h\alpha), \quad \delta_\alpha^R(\tilde{h}, \tilde{\ell}) = \Big( \tilde{h}(\tilde{\ell} \rhd \alpha), \tilde{\ell}\alpha - (\tilde{\ell} \rhd \alpha)\tilde{\ell} \Big), \tag{67a}$$

$$G \mapsto G\ell_0 \implies \delta_Y^R G = GY \implies \delta_Y^R(\ell, h) = \Big( \ell(h \rhd Y), hY - (h \rhd Y)h \Big), \quad \delta_Y^R(\tilde{h}, \tilde{\ell}) = (0, \tilde{\ell}Y), \tag{67b}$$

$$G \mapsto h_0 G \implies \delta_\alpha^L G = \alpha G \implies \delta_\alpha^L(\ell, h) = \Big( \alpha\ell - \ell(\alpha \lhd \ell), (\alpha \lhd \ell)h \Big), \quad \delta_\alpha^L(\tilde{h}, \tilde{\ell}) = (\alpha\tilde{h}, 0), \tag{67c}$$

$$G \mapsto \ell_0 G \implies \delta_Y^L G = YG \implies \delta_Y^L(\ell, h) = (Y\ell, 0), \quad \delta_Y^L(\tilde{h}, \tilde{\ell}) = \Big( Y\tilde{h} - \tilde{h}(Y \lhd \tilde{h}), (Y \lhd \tilde{h})\tilde{\ell} \Big). \tag{67d}$$

It is easy to check using these expressions that the left and right actions are Poisson because they respectively lead to the Poisson-Lie flow equation of Theorem 2.10:

$$\mathbb{i}_{\mathbb{r}(\alpha)}\Omega = -\langle \alpha, \tilde{\ell}^{-1}\mathrm{d}\tilde{\ell} \rangle, \qquad\qquad \mathbb{i}_{\mathbb{r}(Y)}\Omega = \langle Y, h^{-1}\mathrm{d}h \rangle, \tag{68}$$

$$\mathbb{i}_{\mathbb{l}(\alpha)}\Omega = -\langle \alpha, \mathrm{d}\ell\ell^{-1} \rangle, \qquad\qquad \mathbb{i}_{\mathbb{l}(Y)}\Omega = \langle Y, \mathrm{d}\tilde{h}\tilde{h}^{-1} \rangle. \tag{69}$$

**Example 2.14** (The Cotangent Bundle as a Heisenberg Double)**.** Taking $\pi_{\mathcal{G}} = 0$ and thus $\mathcal{G}^* \simeq (\mathfrak{g}^*, +)$ Abelian, one finds the double $\mathcal{D}_H \simeq \mathcal{G}^* \bowtie \mathcal{G} \simeq \mathfrak{g}^* \rtimes \mathcal{G} \simeq T^*\mathcal{G}$ in the right trivialization. In particular in this case, the Heisenberg double symplectic structure is isomorphic to the canonical one on $T^*\mathcal{G}$:

$$\Omega = \langle \mathrm{d}X, \mathrm{d}hh^{-1} \rangle + \frac{1}{2}\langle X, \big[ \mathrm{d}hh^{-1}, \mathrm{d}hh^{-1} \big] \rangle. \tag{70}$$

Note in particular that $\Omega = \mathrm{d}\Theta$ with $\Theta := \langle X, \mathrm{d}hh^{-1} \rangle$, is in this case $\mathrm{d}$-exact. This is generally not the case e.g., for $\mathcal{G}^*$ non-Abelian and hence fully non-Abelian Heisenberg

---

[22]E.g. the formula $\alpha \rhd \ell = \alpha\ell - \ell(\alpha \lhd \ell)$ can be proved by taking the infinitesimal version $h = 1 + \alpha + \dots$ of the identity $h \rhd \ell = h\ell(h \lhd \ell)^{-1}$ which directly follows from the definition of the dressing action.

Doubles. As for the Drinfel'd double Poisson bivector $\pi_\mathcal{D}$, written in a right trivialization $\mathcal{D} \simeq \mathfrak{g}^* \rtimes \mathcal{G}$, it vanishes along $\mathcal{G}$ and restricts to the Kirillov-Kostant-Souriau bivector on $\mathfrak{g}^*$:

$$\pi_\mathcal{D}(h, X) = c_{ab}{}^c X_c \frac{\partial}{\partial X_a} \wedge \frac{\partial}{\partial X_b}. \tag{71}$$

In this case, the action of Drinfel'd double symmetries $\mathcal{D}_\mathrm{D}$ on the Heisenberg double $\mathcal{D}_\mathrm{H} \simeq T^*\mathcal{G}$, are given by the cotangent lift of the right/left action of $\mathcal{G}$ on itself, which is Hamiltonian and therefore Poisson, and by the action of $\mathcal{G}^* \simeq \mathfrak{g}^*$ on $T^*\mathcal{G}$ described in Example 2.7.

The infinitesimal left/right actions of $G_0 = (Y, 1 + \alpha + \dots) \in \mathcal{D}_\mathrm{D}$ on $G = (X, h) \in \mathcal{D}_\mathrm{H}$ are given by:

$$\delta_\alpha^R(X, h) = (0, h\alpha), \qquad\qquad \delta_\alpha^R(\tilde{h}, \tilde{X}) = (\tilde{h}\alpha, \mathsf{ad}_\alpha^* \tilde{X}) \tag{72a}$$

$$\delta_Y^R(X, h) = (h \triangleright Y, 0), \qquad\qquad \delta_Y^R(\tilde{h}, \tilde{X}) = (0, Y) \tag{72b}$$

$$\delta_\alpha^L(X, h) = (-\mathsf{ad}_\alpha^* X, \alpha h), \qquad\qquad \delta_\alpha^L(\tilde{h}, \tilde{X}) = (\alpha\tilde{h}, 0) \tag{72c}$$

$$\delta_Y^L(X, h) = (Y, 0), \qquad\qquad \delta_Y^L(\tilde{h}, \tilde{X}) = (0, Y \triangleleft \tilde{h}) \tag{72d}$$

where the corresponding dressing transformations are given by

$$X \triangleright h = h = \tilde{h} \quad \text{and} \quad X \triangleleft h = \mathsf{Ad}_h^* X = \tilde{X}, \tag{73}$$

$\diamond$

where we denoted by $\mathsf{Ad}^*$ the transpose of the adjoint representation $\mathsf{Ad}$ of $\mathcal{G}$ on $\mathfrak{g}$, namely $\langle X, \mathsf{Ad}_h \alpha \rangle =: \langle \mathsf{Ad}_h^* X, \alpha \rangle$ for all $X \in \mathfrak{g}^*, h \in \mathcal{G}$ and $\alpha \in \mathfrak{g}$.[23]

**Example 2.15** (Isometries of $\mathbb{R}^n$)**.** The group of isometries of $\mathbb{R}^n$ understood as the $n$-dimensional Euclidean space is $ISO(n) \simeq \mathbb{R}^n \rtimes SO(n) \simeq T^*SO(n)$ which we studied in the previous example. In this example the Drinfel'd double is the semidirect product of rotational ($\mathcal{G} \simeq SO(n)$) and translational ($\mathcal{G}^* \simeq \mathfrak{g}^* \simeq \mathbb{R}^n$) symmetries. $\diamond$

Using this example as an analogy, we will keep this terminology even when dealing with general Drinfel'd doubles, viz. if $\mathcal{D} \simeq \mathcal{G}^* \bowtie \mathcal{G}$ then $\mathcal{G} \subset \mathcal{D}_\mathrm{D}$ will be referred to as the *rotational symmetries* of $\mathcal{D}_\mathrm{H}$ and $\mathcal{G}^* \subset \mathcal{D}_\mathrm{D}$ as its *translational symmetries*.

# 3 Poisson-Lie Symmetry for a 0+1D Field Theory

The formulation of Poisson-Lie symmetries in a 0+1D field theory corresponds to the treatment of Poisson-Lie symmetries in which they were originally formulated for mechanical systems [3, 53]. Here, we consider two examples: the generalized spinning top in §3.1, and the deformed generalized spinning top in §3.2. By "generalized" spinning top we refer to a mechanical model with phase space $T^*\mathcal{G} \simeq \mathfrak{g}^* \rtimes \mathcal{G}$, with $\mathcal{G}$ not necessarily $SO(3)$. This phase space supports both Noetherian and Poisson-Lie symmetries, with the dynamics (Hamiltonian) determining which are physically realized. We instead refer to a "deformed" generalized spinning top if its phase space is not the cotangent bundle of $\mathcal{G}$ but a non-trivial Heisenberg double $\mathcal{D}_\mathrm{H} \simeq \mathcal{G}^* \bowtie \mathcal{G}$, i.e., having both curved configuration and momentum spaces.

---

[23]With this notation [52], the co-adjoint *representation* of $\mathcal{G}$ over $\mathfrak{g}^*$ is given by $(\mathsf{Ad}^*)^{-1}$, which is a *right* action.

### 3.1 The (Generalized) Spinning Top

Let $\mathfrak{g}$ be a finite dimensional (real) semi-simple Lie algebra and $\mathcal{G}$ its associated connected and simply connected Lie group. Denote the fields $(X, h) : \mathbb{R} \to T^*\mathcal{G} \simeq \mathfrak{g}^* \rtimes \mathcal{G}$ (using the right trivialization of $T^*\mathcal{G}$), and let[24]

$$\mathcal{L} = \langle X, \partial_t h h^{-1} \rangle - \mathcal{H}(X, h) \tag{74}$$

be the associated first order Lagrangian of the generalized spinning top whose dynamics and symmetries are determined by the choice of Hamiltonian $\mathcal{H}$. We will study two main families of Hamiltonians.

We start by studying the generalized spinning top's phase space using the CPS method. The field space is $\mathcal{F} = \{(X, h) : \mathbb{R} \to T^*\mathcal{G}\}$. Denote the variation of $\mathcal{H}$ in $X$ and $h$ as

$$\mathbb{d}\mathcal{H} = \langle \mathbb{d}X, \mathcal{H}'_X \rangle + \langle \mathcal{H}'_h, h^{-1}\mathbb{d}h \rangle, \tag{75}$$

where $\mathcal{H}'_X$ and $\mathcal{H}'_h$ are naturally valued in $\mathfrak{g}$ and $\mathfrak{g}^*$, as per

$$\delta^L_Y \mathcal{H} := \langle Y, \mathcal{H}'_X \rangle = \frac{d}{dt}\mathcal{H}(X + tY, h)|_{t=0} \quad \text{and} \quad \delta^R_\alpha \mathcal{H} := \langle \mathcal{H}'_h, \alpha \rangle = \frac{d}{dt}\mathcal{H}(X, he^{t\alpha})|_{t=0}. \tag{76}$$

Varying (74) and applying (10) we find the following equations of motion and (pre-)symplectic currents,

$$\mathcal{E} = \langle \mathbb{d}X, \partial_t h h^{-1} - \mathcal{H}'_X \rangle - \langle \partial_t(X \triangleleft h) + \mathcal{H}'_h, h^{-1}\mathbb{d}h \rangle \tag{77}$$

$$\theta = \langle X, \mathbb{d}h h^{-1} \rangle \tag{78}$$

$$\omega := \mathbb{d}\theta = \langle \mathbb{d}X, \mathbb{d}h h^{-1} \rangle + \tfrac{1}{2}\langle X, \left[ \mathbb{d}h h^{-1}, \mathbb{d}h h^{-1} \right] \rangle, \tag{79}$$

where $X \triangleleft h := \mathrm{Ad}^*_h X$ and $h \triangleright X = (\mathrm{Ad}^*_h)^{-1} X$. Recall that according to the CPS method, identifying the symplectic structure $\Omega$ of the theory requires going on-shell ($\approx$) and integrating $\omega$ over a choice of Cauchy hypersurface. In 0+1D, this amounts to choosing a point, say $t_{\mathrm{in}} \in \mathbb{R}$, at which to evaluate the pSF current $\omega$, that is

$$\Omega \approx \omega(t_{\mathrm{in}}). \tag{80}$$

On-shell $\Omega$ is independent of the choice of $t_{\mathrm{in}}$. The space $(\mathcal{F}_{\mathrm{EL}}, \Omega)$ is isomorphic to $T^*\mathcal{G} = \mathcal{D}_{\mathrm{H}} \simeq \mathfrak{g}^* \rtimes \mathcal{G}$ with $\Omega$ corresponding to the semi-Abelian case (70) of the STS symplectic form (64).

The kinematical symmetries of $\mathcal{D}_{\mathrm{H}}$ are the special case of the Drinfel'd double PL group $\mathcal{D}_{\mathrm{D}} \simeq \mathfrak{g}^* \rtimes \mathcal{G}$ written in (72a)-(72d) wherein one has $\mathcal{G}^* \simeq (\mathfrak{g}^*, +)$.

Since the right and left Drinfel'd double actions ($\mathbb{r}(\cdot)$ and $\mathbb{l}(\cdot)$, respectively) are by construction generating PL type flow equations of the form (36) (cf. (68) and (69)), to check which are PL symmetries compatible with the dynamics of the generalized spinning top we need to check which actions preserve the equations of motion i.e., which of the actions define vector fields on $\mathcal{F}_{\mathrm{EL}}$, thereby satisfying point $1'$ of Definition 2.5. For this, we need to fix the Hamiltonian. We are now going to study a few explicit cases.

---

[24]Note that the kinetic term in this Lagrangian arises from the familiar coadjoint orbit method for constructing a geometric action associated to the group $\mathcal{G}$, with $\langle \cdot, \cdot \rangle$ the natural pairing between $\mathfrak{g}$ and its dual $\mathfrak{g}^*$.

**Case 1: Noetherian Symmetries.** A natural choice of Hamiltonian is given by

$$\mathcal{H}(X, h) = \frac{1}{2}\kappa(X, X), \tag{81}$$

where the $\kappa$ is a non-degenerate, $\mathsf{Ad}^*$-invariant, bilinear form on $\mathfrak{g}^*$, e.g. the inverse of the Killing bilinear form on $\mathfrak{g}$.

Identifying $\mathfrak{g}^*$ and $\mathfrak{g} \simeq \mathfrak{g}^{**}$ through $\kappa : X \mapsto X_\kappa := \kappa(X, \cdot)$, the equations of motion (77) with this Hamiltonian read:

$$\partial_t h h^{-1} - X_\kappa \approx 0 \quad \text{and} \quad \partial_t(X \triangleleft h) \approx 0 \tag{82}$$

In view of the $\mathsf{Ad}^*$-invariance of $\kappa$ one has $(X \triangleleft h)_\kappa = \mathsf{Ad}_{h^{-1}}(X_\kappa)$, which enables us to rewrite the above as

$$\partial_t h h^{-1} - X_\kappa \approx 0 \quad \text{and} \quad \partial_t X_\kappa \approx 0, \tag{83}$$

after exploiting the linearity and time independence of $\kappa$, as well as going on-shell of the first equation of motion. The first equation essentially defines the canonical momenta ($X$) in terms of the velocities in configuration space ($\partial_t h$). The second equation is a momentum conservation equation. The two versions of momentum conservation are equivalent on-shell of the first equation owing to the $\mathsf{Ad}^*$ invariance of $\kappa$.

The present choice of Hamiltonian is clearly not compatible with the translational symmetries (72b) and (72d), which shift $X$ and leave $h$ invariant. However, the rotational symmetries (72a) and (72c) are easily seen to be Noetherian symmetries, since

$$\delta_\alpha^R \mathcal{L} = 0 = \delta_\alpha^L \mathcal{L}, \tag{84}$$

Recall, as mentioned in §2.3, this implies (on-shell) preservation of the equations of motion. Proceeding with the CPS analysis one quickly arrives at the $\mathfrak{g}^*$-valued Noether charges (momentum maps) $q^R = X \triangleleft h$ and $q^L = X$, viz.

$$\mathring{\mathbb{i}}_{\mathbb{r}(\alpha)}\Omega := \mathbb{Q}^R(\alpha) = -\mathbb{d}\langle (X \triangleleft h)(t_{\text{in}}), \alpha \rangle \tag{85}$$

$$\mathring{\mathbb{i}}_{\mathbb{l}(\alpha)}\Omega := \mathbb{Q}^L(\alpha) = -\mathbb{d}\langle X(t_{\text{in}}), \alpha \rangle \tag{86}$$

The conservation of these charges corresponds to the momentum conservation equation expressed above and, as noted there, they are essentially equivalent on-shell of $\partial_t h h^{-1} \approx X_\kappa$.

Physically, the system we just described corresponds to a particle moving on a group manifold with geodesic motion associated to the metric obtained from $\kappa$ by right transport. Since $\kappa$ is $\mathsf{Ad}$-invariant, this metric has global symmetries given by right and left "rotations" on the group. One then obtains conserved charges generating such transformations in the usual way.

Another physical interpretation of this system is that of a (generalized) spherically symmetric spinning top, with $X$ being its angular momentum vector and $\kappa$ (the inverse of) its rotationally invariant moment of inertia.

From the perspective of the spinning top it is natural to consider a moment of inertia that is not rotationally invariant, thus replacing $\mathcal{H}(X) = \frac{1}{2}\kappa(X, X)$ with $\mathcal{H}(X) = \frac{1}{2}I(X, X)$ for $I$ a non-degenerate symmetric bilinear form which fails to be $\mathsf{Ad}^*$-invariant. This breaks the left rotational symmetry of the system (cf. (72c)), while continuing to leave $\mathcal{L}$ invariant under right rotations (72a), $\delta_\alpha^R \mathcal{L} = 0$. Thus one concludes that only $q^R = X \triangleleft h$ is a conserved Noetherian charge and momentum map.

The physical interpretation of this fact is that $X$ and $\tilde{X} := X \triangleleft h$ are the angular momenta as seen respectively in the Lagrangian (comoving) and Eulerian (laboratory) frames of the spinning top. Only the latter is expected to be conserved for a non-symmetric top.

**Case 2: Poisson-Lie Symmetries.** Switching the role of configuration and momentum space, one can choose a Hamiltonian that only depends on $h$ and not on $X$. Assuming $\mathcal{G}$ is a matrix Lie group, let us consider

$$\mathcal{H}(h) = c_1 \text{Tr}(h) - c_2 \tag{87}$$

where $c_1$ and $c_2$ are appropriate normalization constants. In this case, $\mathcal{H}(h)$ is neither invariant under the right or left rotations ((72a) and (72c)), but it is invariant under the right and left translation symmetries of $X$ that leave $h$ invariant ((72b) and (72d)).

These translational symmetries are an illustration of non-trivial Poisson-Lie symmetries, i.e. Poisson-Lie symmetries which are truly non-Noetherian. For example, the corresponding Lagrangian is *not* invariant under either right or left translations, not even up to a total derivative, since

$$\delta_Y^R \mathcal{L} = \langle Y, h^{-1} \partial_t h \rangle, \quad \delta_Y^L \mathcal{L} = \langle Y, \partial_t h h^{-1} \rangle. \tag{88}$$

To prove that both of these transformations are PL symmetries as per Definition 2.5, we need to prove that they preserve the equations of motion and that they satisfy the PL flow equation.

First, we plug our choice of Hamiltonian $\mathcal{H}(h) = c_1 \text{Tr}(h) - c_2$ into the equations of motion (77), to find

$$\partial_t h \approx 0 \quad \text{and} \quad \langle \partial_t (X \lhd h), h^{-1} \text{d} h \rangle + c_1 \text{Tr}(h \ (h^{-1} \text{d} h)) \approx 0. \tag{89}$$

Next, we prove that they are preserved (on-shell) by the symmetry action. As mentioned, any function of $h$ is invariant under these right/left translations, so we only need to check that

$$\delta_Y^R \partial_t (X \lhd h) = \partial_t (\delta_Y^R X \lhd h) = \partial_t ((h \rhd Y) \lhd h) = \partial_t ((Y \lhd h^{-1}) \lhd h) = \partial_t Y = 0 \tag{90}$$

$$\delta_Y^L \partial_t (X \lhd h) = \partial_t (\delta_Y^L X \lhd h) = \text{ad}_{h^{-1}\partial_t h}^* (Y \lhd h) \approx 0 \tag{91}$$

where in the last equation we went on-shell of the first equation of motion $\partial_t h \approx 0$.

Proceeding with the CPS analysis, it is not hard to recover the appropriate two corresponding Poisson-Lie momentum maps of the possible actions determined by (68) and (69), namely

$$\mathring{\mathbb{i}}_{\mathfrak{r}(Y)} \Omega := \mathbb{Q}^R(Y) = \langle h^{-1}(t_{\text{in}}) \text{d} h(t_{\text{in}}), Y \rangle, \tag{92}$$

$$\mathring{\mathbb{i}}_{\mathbb{l}(Y)} \Omega := \mathbb{Q}^L(Y) = \langle \text{d} h(t_{\text{in}}) h^{-1}(t_{\text{in}}), Y \rangle, \tag{93}$$

where $Y \in \mathfrak{g}^*$, which are associated to the conserved group-valued Poisson-Lie charge

$$h(t_{\text{in}}) : \mathcal{F}_{EL} \to \mathcal{G} \tag{94}$$

thereby satisfying our Definition 2.5 of Poisson-Lie symmetries.

Notice the dualization of the roles of $\mathcal{G}$ and $\mathfrak{g}^*$ with respect to the previous example, where (configuration space $\sim$ symmetry parameter) $\leftrightarrow$ (momentum $\sim$ target of the momentum map).

We note that in this system, the would-be canonical momentum is encoded in $h$ while the configuration variable is morally given by $\tilde{X} := X \lhd h$ (89). Taking e.g. $\mathcal{G} \simeq SU(2)$, this system could be interpreted therefore as a particle moving in $\mathbb{R}^3$, but subject to a *curved,*

*and in fact closed, momentum space $\mathcal{G} \simeq S_3$.* In particular, this results in a non-trivial Poisson bracket between the coordinates on configuration space: inverting $\Omega$, one finds in particular $\{\tilde{X}^i, \tilde{X}^j\} = \epsilon^{ij}{}_k \tilde{X}^k$. This specific system encodes the notion of a spinless particle in 3D Euclidean gravity [54]. It can be used to encode the notion of particle excitations for a field theory over of a non-commutative spacetime of the Lie algebra type. The (Poisson) non-commutativity can also be interpreted as a relative notion of locality [55, 56]. The quantization of this system would be done using the representation theory of the quantum group arising from the Poisson-Lie symmetry [57].

From this perspective, taking a sensible Abelian limit, means taking a limit of small momenta that do not "see" the curvature of $\mathcal{G}$. Writing $h = e^p = 1 + p + \frac{1}{2}p^2 + \dots$, for $p = \frac{i}{\sqrt{2}}\vec{\sigma} \cdot \vec{p} \in \mathfrak{su}(2)$ with $\vec{\sigma}$ the three Pauli matrices, and keeping the first non-vanishing orders, we have that $\mathcal{H} = 2 - \mathrm{Tr}(h) = \frac{1}{2}|\vec{p}|^2 + \dots$, $\delta_Y^{R,L}(h, X) = (0, Y)$, and $\mathbb{Q}^{R,L}(Y) = \mathrm{d}\vec{p} \cdot \vec{Y}$. Thus, in the limit of small momenta one recovers a free particle moving in $\mathbb{R}^3$, subject to a translational symmetry generated by its linear momentum.

## 3.2 The Deformed (Generalized) Spinning Top

We now turn to the *deformed* generalized spinning top. The phase space of this 0+1D field theory is described by a non-trivial Heisenberg double $\mathcal{D}_H \simeq \mathcal{G}^* \bowtie \mathcal{G}$ acted upon by the corresponding Drinfel'd double Poisson-Lie group $\mathcal{D}_D$. By non-trivial, we mean that $\mathcal{D}_H$ is *not* isomorphic to $T^*\mathcal{G}$, but has both $\mathcal{G}$ and $\mathcal{G}^*$ non-Abelian. Since the symplectic 2-form on $\mathcal{D}_H$ is not exact, a priori there is not an obvious way to recover this phase space through a CPS analysis, not even from a first order Lagrangian.

In this section, to accommodate for a non-exact symplectic 2-form in the CPS, we divert from the usual action formulation for 0+1D field theories and adopt a different formalism inspired by [31, 58]. It involves an action functional for a non-covariant "topological string" with target space being the deformed spinning top's phase space $\mathcal{D}_H$. The word "topological" here simply means that ultimately only the end points of the string matter, while the word "non-covariant" refers to the worldsheet theory heavily relying on a choice of 1+1 foliation.

The article [31] only looks at the "action principle" as a way to generate the equations of motion. However, the imposition of the initial/final conditions for the variational action principle to be well-defined requires more care. This is because the proposed initial/final conditions in [31] are too strong for its equations of motion, for they end up fixing the "$q$'s and $p$'s" at both the initial and final times. We discuss more of the subtleties of this point in §3.2.2.

Here, we improve on this setup, and find a rather different picture, namely a string that closes on-shell. In particular, when going on-shell, the two endpoints collapse on a single trajectory that runs along the physical history of the mechanical system, while the bulk of the string is free to wobble unconstrained by the equations of motion.[25]

This example will serve as an introduction to the realm of 1+1D field theories in the next section.

We start by putting forward the general framework for a mechanical system with (finite dimensional) phase space $P$, (non-exact) symplectic 2-form $\omega$, and Hamiltonian $\mathcal{H}$, before

---

[25]Note that this is a drastically different scenario from [31] where it was proposed that one end point of the string is kept fixed at all times.

842 specializing it to the case of the deformed generalized spinning top's finite dimensional
843 phase space $(P, \omega) = (\mathcal{D}_{\mathrm{H}}, \Omega)$ where $\Omega$ is the Heisenberg double sympelctic form (64).

844 ### 3.2.1 Setup: Action Principle

845 Let $(P, \omega)$ be a symplectic manifold, and denote its points by $z$. A history is a curve
846 $\xi : \mathbb{R} \to P$, $t \mapsto z = \xi(t)$. We "extend" such curves to a worldsheet $\Sigma \simeq \mathbb{R} \times [0, 1]$:

$$\gamma : \Sigma \to P, \quad (t, \sigma) \mapsto z = \gamma(t, \sigma) \tag{95}$$

847 by thinking of the two end points of the worldsheet as spanning two histories of the
848 mechanical system:

$$\begin{cases} \xi_0(t) := \gamma(t, 0) \\ \xi_1(t) := \gamma(t, 1) \end{cases} \tag{96}$$

849 There are three relevant manifolds: the string's worldsheet $\Sigma$, the string's target space $P$,
850 and the infinite dimensional string's field space

$$\mathcal{F} := \{\gamma : \Sigma \to P\} = C^\infty(\Sigma, P). \tag{97}$$

851 We denote the Cartan's calculus operations on $\Sigma$, $P$, and $\mathcal{F}$, respectively by $(d, i_\bullet, L_\bullet)$,
852 $(d^{(P)}, i_\bullet^{(P)}, L_\bullet^{(P)})$ and $(\mathrm{d}, \hat{\mathrm{i}}_\bullet, \mathbb{L}_\bullet)$. The closure of $\omega$ over $P$ gives us the following useful
853 identities:

854 **Lemma 3.1.** Let $\omega \in \Omega^2(P)$ be closed, and $\gamma \in \mathcal{F} = C^\infty(\Sigma, P)$. Then, in an abstract index
855 notation where $\bullet^a$ are indices on $\Sigma \ni x = (t, \sigma)$ and $\bullet^I$ are indices on $P \ni z$, the following
856 identities hold:

$$\mathrm{d}(\gamma^*\omega) = -d\vartheta \quad \text{and} \quad \mathrm{d}\vartheta = -d\varpi, \tag{98}$$

857 where $\gamma^*\omega \in \Omega^{2,0}_{(\mathrm{loc})}(\Sigma \times \mathcal{F})$, $\vartheta \in \Omega^{1,1}_{(\mathrm{loc})}(\Sigma \times \mathcal{F})$ and $\varpi \in \Omega^{0,2}_{(\mathrm{loc})}(\Sigma \times \mathcal{F})$ are given by

$$(\gamma^*\omega)(x) = \frac{1}{2}\omega_{IJ}(\gamma(x))d\gamma^I(x) \wedge d\gamma^J(x) = \omega_{IJ}(\gamma(x))\partial_t\gamma^I(x)\partial_\sigma\gamma^J(x)dt \wedge d\sigma, \tag{99}$$

$$\vartheta(x) := \omega_{IJ}(\gamma(x))\mathrm{d}\gamma^I(x) \wedge d\gamma^J(x) = \omega_{IJ}(\gamma(x))\mathrm{d}\gamma^I(x) \wedge \partial_a\gamma^J(x)dx^a \tag{100}$$

$$\varpi(x) := \frac{1}{2}\omega_{IJ}(\gamma(x))\mathrm{d}\gamma^I(x) \wedge \mathrm{d}\gamma^J(x) \tag{101}$$

858 $\diamond$

859 We provide two proofs of this Lemma: one computational but elementary, and one more
860 conceptual.

861 *Proof 1.* Equation (99) is obvious. Recalling that $\mathrm{d}$ and $d$ anticommute, we can vary it to
862 find:

$$\mathrm{d}(\gamma^*\omega)(x) = \frac{1}{2}\partial_K\omega_{IJ}(\gamma(x))\mathrm{d}\gamma^K(x) \wedge d\gamma^I(x) \wedge d\gamma^J(x) - \omega_{IJ}(\gamma(x))d\mathrm{d}\gamma^I(x) \wedge d\gamma^J(x). \tag{102}$$

863 Similarly, from the definition of $\vartheta$, we get:

$$d\vartheta(x) = d\Big(\omega_{IJ}(\gamma(x))\mathrm{d}\gamma^I(x) \wedge d\gamma^J(x)\Big)$$

$$= \partial_K \omega_{IJ}(\gamma(x)) d\gamma^K(x) \wedge \mathrm{d}\gamma^I(x) \wedge d\gamma^J(x) + \omega_{IJ}(\gamma(x)) d\mathrm{d}\gamma^I(x) \wedge d\gamma^J(x) \quad (103)$$

The first identity of the Lemma follows from the closure of $\omega$, since (henceforth suppressing $x$):

$$
\begin{aligned}
\mathrm{d}(\gamma^*\omega) + d\vartheta &= \frac{1}{2}\partial_K\omega_{IJ}(\gamma)\mathrm{d}\gamma^K \wedge d\gamma^I \wedge d\gamma^J + \partial_K\omega_{IJ}(\gamma)d\gamma^K \wedge \mathrm{d}\gamma^I \wedge d\gamma^J \\
&= \gamma^*\left(\partial_I\omega_{JK} + \frac{1}{2}\partial_K\omega_{IJ}\right)\mathrm{d}\gamma^K \wedge d\gamma^I \wedge d\gamma^J \\
&= \frac{1}{2}\gamma^*(\partial_I\omega_{JK} - \partial_J\omega_{IK} + \partial_K\omega_{IJ})\mathrm{d}\gamma^K \wedge d\gamma^I \wedge d\gamma^J \\
&= \frac{1}{2}\gamma^*(d\omega)_{IJK}d\gamma^I \wedge d\gamma^J \wedge \mathrm{d}\gamma^K = 0,
\end{aligned}
\quad (104)
$$

whence

$$\mathrm{d}(\gamma^*\omega) = -d\vartheta. \quad (105)$$

The second identity can be proven in a similar way. $\qquad\square$

*Proof 2.* Consider the evaluation map $\mathrm{ev} : \Sigma \times \mathcal{F} \to P$, $(x, \gamma) \mapsto z = \gamma(x)$. Then, since the space of local forms on $\Sigma \times \mathcal{F}$ can be equipped with the total differential $\mathbf{d} := d + \mathrm{d}$, $\mathbf{d}^2 = d\mathrm{d} + \mathrm{d}d = 0$, it is immediate to verify that:

$$(\mathrm{ev}^*\omega)(x, \gamma) = \frac{1}{2}\omega_{IJ}(\gamma(x))\mathbf{d}\gamma^I(x) \wedge \mathbf{d}\gamma^J(x) = (\gamma^*\omega)(x) + \vartheta(x) + \varpi(x) \quad (106)$$

The closure of $\omega$ then implies:

$$
\begin{aligned}
0 = \mathrm{ev}^*(d^{(P)}\omega) &= \mathbf{d}(\mathrm{ev}^*\omega) \\
&= (d + \mathrm{d})(\gamma^*\omega + \vartheta + \varpi) \\
&= \underbrace{d\gamma^*\omega}_{(3,0)} + \underbrace{(\mathrm{d}(\gamma^*\omega) + d\vartheta)}_{(2,1)} + \underbrace{(\mathrm{d}\vartheta + d\varpi)}_{(1,2)} + \underbrace{\mathrm{d}\varpi}_{(0,3)}.
\end{aligned}
\quad (107)
$$

In the last line we organized the forms by their mixed $(p,q)$-degree as per

$$\Omega^3_{(\mathrm{loc})}(\Sigma \times \mathcal{F}) = \sum_{\substack{p,q \geq 0 \\ p+q=3}} \Omega^{p,q}_{(\mathrm{loc})}(\Sigma \times \mathcal{F}). \quad (108)$$

The conclusion follows from the fact that terms in different (mixed) degree must vanish independently. Note that the remaining equations, $d(\gamma^*\omega) = 0$ and $\mathrm{d}\varpi = 0$, also follow from the $d^{(P)}$ closure of $\omega$. $\qquad\square$

We are now ready to introduce the "topological string" action $\Psi : \mathcal{F} \to \mathbb{R}$ associated to the mechanical system $(P, \omega, \mathcal{H})$ and perform its CPS analysis. Let

$$\Psi(\gamma) := \int_\Sigma \gamma^*\omega - \partial_\sigma\mathcal{H}(\gamma(t,\sigma))d\sigma dt = \int_\Sigma \gamma^*\omega - \int_\mathbb{R}(\mathcal{H}(\xi_1(t)) - \mathcal{H}(\xi_0(t)))dt. \quad (109)$$

Note that the first term is coordinate independent whereas the second one heavily relies on the $t$-foliation of $\Sigma$. The second term also concentrates at end points of the string, i.e. depends only on the two histories of the mechanical system $(\xi_0, \xi_1)$ – rather than on the entire string's worldsheet.

If $\mathcal{H} = 0$, then we obtain the action $\Psi_0(\gamma) = \int_\Sigma \gamma^*\omega$ which is the topological string action obtained by integrating out the field $A_I$ from the (topological) Poisson $\sigma$-model

$$\Psi_0^{\text{Poiss}}(\gamma, A) := \int_\Sigma A_I \wedge d\gamma^I + \frac{1}{2}\pi(\gamma)^{IJ}A_I \wedge A_J, \tag{110}$$

where $\pi^{IJ} \in \mathfrak{X}^2(P)$ is the Poisson bivector on $P$ obtained from inverting $\omega_{IJ} \in \Omega^2(P)$, and $A_I$ is a $(T^*P)$-valued 1-form on $\Sigma$. The (supersymmetric extension of the) resulting action is sometimes referred to as the A-model (e.g. [59]).

**Proposition 3.2.** The action $\Psi$ for $\gamma \in \mathcal{F}$ over $\Sigma = [t_{\text{in}}, t_{\text{fin}}] \times [0, 1]$ yields the following: equations of motion $(\mathcal{E}_I^0, \mathcal{E}_I^1)$ for the string's endpoints $(\xi_0(t), \xi_1(t))$; pre-symplectic potential (pSP) $\Theta$ for the worldsheet; and pre-symplectic 2-form $\Omega$ on $C = [0, 1]$, where $C$ is the time $t_{\text{in}}$ Cauchy surface of $\Sigma$ stretching from $\gamma(t_{\text{in}}, 0) = \xi_0(t_{\text{in}})$ to $\gamma(t_{\text{in}}, 1) = \xi_1(t_{\text{in}})$:

$$\mathcal{E}_I^\bullet = -\omega_{IJ}(\xi_\bullet)\partial_t\xi_\bullet^J - \partial_I\mathcal{H}(\xi_\bullet) \quad \text{for} \quad \bullet \in \{0, 1\}, \tag{111}$$

$$\Theta = -\int_C \vartheta = -\int_0^1 d\sigma\, \omega_{IJ}(\gamma)\partial_\sigma\gamma^I \mathbb{d}\gamma^J \tag{112}$$

$$\Omega := \mathbb{d}\Theta = \varpi\Big|_{(t_{\text{in}},\sigma=0)}^{(t_{\text{in}},\sigma=1)} = \frac{1}{2}\omega_{IJ}(\xi_\bullet)\mathbb{d}\xi_\bullet^I \wedge \mathbb{d}\xi_\bullet^J\Big|_{\bullet=0}^{\bullet=1}. \tag{113}$$

$\diamond$

Before proceeding to the proof, we note that the equations of motion are nothing else than two copies of Hamilton's equations of motion expressed as the Hamiltonian flow equations for the *0+1D* dynamical system $(P, \omega, \mathcal{H})$ traced by the two endpoints of the string. Similarly, the CPS symplectic structure $\Omega$ only depends on the worldlines of the two endpoints $(\xi_0(t), \xi_1(t))$, and not on the entire worldsheet $\gamma(t, \sigma)$.

*Proof.* We start by varying $\Psi$ using the identities of Lemma 3.1:

$$\begin{aligned}
\mathbb{d}\Psi(\gamma) &= \int_\Sigma \mathbb{d}(\gamma^*\omega) - \partial_\sigma\mathbb{d}\mathcal{H}(\gamma)d\sigma dt \\
&= \int_\Sigma -d\vartheta - \partial_\sigma\partial_I\mathcal{H}(\gamma)\mathbb{d}\gamma^I d\sigma dt \\
&= \underbrace{\int_0^1 \left[-\vartheta_\sigma\right]_{t=t_{\text{in}}}^{t=t_{\text{fin}}} d\sigma}_{=:\Theta(t_{\text{fin}})-\Theta(t_{\text{in}})} + \int_{t_{\text{in}}}^{t_{\text{fin}}} \underbrace{\left[-\vartheta_t - \partial_I\mathcal{H}(\gamma)\mathbb{d}\gamma^I\right]_{\sigma=0}^{\sigma=1}}_{=:\mathcal{E}} dt
\end{aligned} \tag{114}$$

Using (100), we recognize the sought formula for $\Theta$. To recognize the claimed form of $\mathcal{E}$, using the boundary conditions $\gamma(t, \sigma = 0, 1) = \xi_{0,1}(t)$, we write:

$$\mathcal{E} = \left(-\vartheta_t - \partial_I\mathcal{H}(\gamma)\mathbb{d}\gamma^I\right)\Big|_{\sigma=0}^{\sigma=1} = \left(-\omega_{IJ}(\xi_\bullet)\partial_t\xi_\bullet^J - \partial_I\mathcal{H}(\xi_\bullet)\right)\mathbb{d}\xi_\bullet^I\Big|_{\bullet=0}^{\bullet=1} =: \mathcal{E}_I^1 \mathbb{d}\xi_1^I - \mathcal{E}_I^0 \mathbb{d}\xi_0^I. \tag{115}$$

Finally, using again Lemma 3.1 and that $\mathbb{d}\gamma(t, \sigma = 0, 1) = \mathbb{d}\xi_{0,1}$, we find:

$$\Omega := \mathbb{d}\Theta = -\int_C \mathbb{d}\vartheta = \int_C d\varpi = \varpi\Big|_{\sigma=0}^{\sigma=1} = \frac{1}{2}\omega_{IJ}(\xi_\bullet)\mathbb{d}\xi_\bullet^I \wedge \mathbb{d}\xi_\bullet^J\Big|_{\bullet=0}^{\bullet=1}. \tag{116}$$

$\square$

### 3.2.2 Aside: The Variational Action Principle for $\Psi(\gamma)$

We have just shown how a covariant phase space (CPS) analysis of the "topological string" action $\Psi(\gamma)$ yields the symplectic 2-form and the Hamilton equations of motion of any mechanical system, even one whose phase space fails to be a cotangent bundle – like a system with a compact phase space. More precisely, what one arrives at is *two copies* of such a system, with opposite symplectic structure.

In the following we will simply focus on one of these copies.

The CPS derives the equations of motion and the symplectic structure from the bulk ($\mathcal{E}$) and boundary ($d\Theta$) decomposition of $\mathrm{d}\Psi(\gamma)$. This decomposition is solely dictated by Takens' theorem (10) [8].

Traditionally, one thinks of the action as generating the equations of motion based on a variational action principle *at fixed initial/final configuration*. To provide a well-defined variational principle for $\Psi(\gamma)$ one then has to prescribe appropriate initial/final conditions that make the boundary contribution of $\mathrm{d}\Psi$, namely $d\Theta$, vanish.

Looking at the expression (112) for $\Theta$, one naively might expect that fixing $\mathrm{d}\gamma^I = 0$ at the initial and final times is a good boundary condition. However, this means fixing in particular $\mathrm{d}\xi_{0,1} = 0$ at the initial and final times and, since $\xi$ is valued in the *phase space* of the mechanical system, rather than its configuration space, this is too strong a boundary condition for Hamilton's equations of motion, for it corresponds to fixing "the $q$'s *and* the $p$'s" at both initial and final times.[26]

Upon a closer look – assuming for a moment that global Darboux coordinates $z^I = (q^i, p_i)$ exist, namely that one can write globally $\omega = \mathrm{d}q^i \wedge \mathrm{d}p_i$ – one sees that, e.g., at $t = t_{\mathrm{in}}$, the boundary term

$$\Theta(t_{\mathrm{in}}) = -\int_0^1 d\sigma \ \left(\partial_\sigma q^i \mathrm{d}p_i - \partial_\sigma p_i \mathrm{d}q^i\right), \tag{117}$$

can be made to vanish provided that one fixes

$$\begin{cases} \mathrm{d}q^i(t_{\mathrm{in}}, \sigma) = 0 \\ \partial_\sigma q^i(t_{\mathrm{in}}, \sigma) = 0 \end{cases} \tag{118}$$

One can proceed similarly at $t = t_{\mathrm{fin}}$, where the same condition can be imposed – possibly with respect to a different set of Darboux coordinates. The crucial point is that this condition fixes only *half* of the degrees of freedom

$$\mathrm{d}q^i(t_{\mathrm{in/fin}}) = 0, \tag{119}$$

and not all of them as the condition $\mathrm{d}\gamma(t_{\mathrm{in/fin}}) = 0$ would do.

Abstracting from these considerations, we arrive at the following initial/final conditions to be imposed on variations of $\gamma$ to obtain a well defined action principle for $\Psi(\gamma)$:[27] given a pair of Lagrangian submanifolds $(L_{\mathrm{in}}, L_{\mathrm{fin}}) \subset (P, \omega)$, one demands that

$$\gamma^I(t_\bullet, \sigma) \in L_\bullet \quad \text{for} \quad \bullet \in \{\mathrm{in}, \mathrm{fin}\}. \tag{120}$$

---

[26]Missing this point is the oversight committed in [31].

[27]In adapted (local) Darboux coordinates $z = (q_\bullet^i, p_i^\bullet)$, where $L_\bullet = \{q_\bullet^i = 0\}$, this is the same as asking that $q_\bullet^i(t_\bullet, \sigma) = 0$ at all $\sigma \in [0, 1]$. Since the coordinates are adapted, this does not necessarily mean that the initial and final configurations coincide, just that "the initial and final $q$'s are each fixed and constant wrt to $\sigma$".

Note that these initial/final conditions involve *both* the initial and final value of $\xi_0(t)$ and $\xi_1(t)$, which are those relevant for solving the equations of motion *of the topological string*, and also the initial and final values of the *bulk* of the string, even though the bulk of the string is not governed by said equations of motion. This situation is not only unusual, but it also has an unexpected consequence: the histories solving the variational action principle correspond to two *identical* copies of the same mechanical system, including the fact that they must satisfy the *same initial/final conditions*. This is because the initial/final conditions (118) (or (120)) require that $q^i(t_{\text{in/fin}}, \sigma)$ are constant in $\sigma$ and thus that they are the same at $\sigma = 0$ and 1.

For histories satisfying the variational action principle, then, the string is a closed string with *(i)* a marked point following the physical 0+1D trajectory $\xi(t) :\approx \xi_0(t) \approx \xi_1(t)$, and *(ii)* an unconstrained bulk, free to wobble in phase space as long as at the initial/final time it all lies in $L_{\text{in/fin}}$. It is important to note that the string closes only on-shell and provided that the initial/final conditions are imposed. Had we imposed that the string was closed from the start, we would have found a completely trivial system, possessing identically vanishing equations of motions and symplectic 2-form.

We conclude this section by noting a couple of issues with this variational setup. Firstly, although the boundary conditions (118) (or (120)) are necessary to induce a well-defined variational action principle, they are an overkill from a dynamical perspective, where it suffices to ask that $\xi_0(t_{\text{in/fin}})$ and $\xi_1(t_{\text{in/fin}})$ belong *separately* to pairs of Lagrangian submanifolds of $P$, $(L^0_{\text{in}}, L^0_{\text{fin}})$ and $(L^1_{\text{in}}, L^1_{\text{fin}})$ respectively. Second, and maybe more seriously, one finds that strings which satisfy the variational action principle carry – *on-shell* – vanishing symplectic currents and thus vanishing charges (whatever their symmetries might be), since the two copies of the identical mechanical system appear with opposite signs and their respective contributions cancel out.

For these reasons, in the following, we shall focus on a single copy of the mechanical system, e.g. the one attached to the $\sigma = 0$ end. Whether the second one is taken identical to the first, or independent from it, depends on how much importance one attaches to the variational action principle. We will henceforth ignore this question.

### 3.2.3 Symmetries of the Deformed (Generalized) Spinning Top

We now specialize these results to the deformed generalized spinning top, where $\omega$ is the STS form (64) on the Heisenberg double $P = \mathcal{D}_{\text{H}}$. We recall our notation according to which an element of $\mathcal{D}_{\text{H}}$ is denoted $z \equiv G$.

Using the decompositions $\mathcal{D}_{\text{H}} \simeq \mathcal{G}^* \bowtie \mathcal{G} \simeq \mathcal{G} \bowtie \mathcal{G}^*$ we get maps $z \mapsto (\ell, h)$ and $z \mapsto (\tilde{h}, \tilde{\ell})$ respectively, where $G = \ell h = \tilde{h}\tilde{\ell}$ with $\ell, \tilde{\ell} \in \mathcal{G}^*$ and $h, \tilde{h} \in \mathcal{G}$.

With this notation, the action is determined by a Hamiltonian $\mathcal{H}(\ell, h)$, which we will address soon, together with the pullback of the STS form (64) to the topological "worldsheet" $\Sigma$:

$$
\gamma^*\omega = \frac{1}{2}\Big(\langle d\ell\ell^{-1}, d\tilde{h}\tilde{h}^{-1}\rangle + \langle \tilde{\ell}^{-1}d\tilde{\ell}, h^{-1}dh\rangle\Big)
$$
$$
= \frac{1}{2}\Big(\langle \partial_I\ell\ell^{-1}, \partial_J\tilde{h}\tilde{h}^{-1}\rangle + \langle \tilde{\ell}^{-1}\partial_I\tilde{\ell}, h^{-1}\partial_J h\rangle\Big)d\gamma^I \wedge d\gamma^J, \tag{121}
$$

where $\partial_I := \partial/\partial z^I$ and $d = dx^\mu\partial_\mu = dx^\mu\partial/\partial x^\mu$, with $z^I$ and $x^\mu = (t, \sigma)$ corresponding to coordinates on $P = \mathcal{D}_{\text{H}}$ and $\Sigma$ respectively.

973 Focusing on either endpoint of the string, say $\sigma = 0$, the CPS analysis then yields the
974 following two-form on the space of its worldlines $\xi(t)$:

$$\Omega|_{\sigma=0} = \frac{1}{2}\Big(\langle \mathrm{d}\ell\ell^{-1} \wedge \mathrm{d}\tilde{h}\tilde{h}^{-1}\rangle + \langle \tilde{\ell}^{-1}\mathrm{d}\tilde{\ell} \wedge h^{-1}\mathrm{d}h\rangle\Big). \tag{122}$$

975 Note that this 2-form fails to be $\mathrm{d}$-exact.

976 We now briefly consider some choices of Hamiltonian $\mathcal{H}(\ell, h)$ and investigate the corre-
977 sponding PL symmetries. Recall that a PL symmetry is a transformation of the (off-shell)
978 worldine $\xi(t)$ which must preserve the equations of motion and yield a PL flow equation
979 in the CPS, for some group valued momentum map $Q$.

980 Since the symmetries we are interested in are in fact symmetries of the string's target
981 space, they act pointwise on $\xi(t)$ and therefore admit an obvious extension to the entire
982 string $\gamma(t, \sigma)$, where the symmetry parameters are taken to be constant in $\sigma$ and $\tau$.

983 As in the case of the (non-deformed) generalized spinning top, we can choose either
984 $\mathcal{H} = \mathcal{H}(\ell)$ or $\mathcal{H} = \mathcal{H}(h)$ which respectively select as symmetries a subset of the possible
985 Drinfel'd double actions. These choices are of course "dual" to each other, and combine a
986 situation where both configuration and momentum space are deformed, or "curved". The
987 discussions from the previous section should make clear what the resulting PL momentum
988 map and group valued PL charges are associated to these choices. For example, choosing
989 the dynamics $\mathcal{H} = c_1\mathrm{Tr}(h) - c_2$ for e.g., $\mathcal{G} = SU(2)$, leads to the general case of a deformed
990 spinning top whose (angular) momentum space fails to be linear if $\mathcal{G}^* \neq (\mathfrak{g}^*, +)$.

991 A notable example closely related to the deformed spinning top is that of a deformed
992 spinor, wherein one can choose $\mathcal{G} = SU(2)$ and $\mathfrak{g}^* = \mathfrak{an}(2)$, so that $\mathcal{D}_H = SL(2, \mathbb{C})$ (seen
993 as a 6-dimensional real group) [60].

994 In the next section, we will consider a genuinely 1+1D field theory, i.e. the Klimčík-Ševera
995 (KS) model for an open string, rather than a 0+1D theory with an auxiliary non-covariant
996 dimension introduced to deal with a non-exact symplectic structure as was considered
997 here. Although the second order formulation of the KS model looks quite distant from the
998 example treated here, when written in a first order formulation the similarities are quite
999 striking. Curiously, in the second order formulation the PL symmetries act non-locally on
1000 the space of fields, while in the first order formulation they act locally, but the charges
1001 localize at the string's endpoint in a way not so dissimilar to the formulation considered
1002 in this section. The non-locality, in this case, hides in the physical interpretation of the
1003 first order variables, since the relevant "Legendre" transform is itself non-local (cf. §4.5
1004 for details).

## 4 Poisson-Lie Symmetry for a 1+1D Field Theory

1006 In this section we will be revisiting the Poisson-Lie symmetric non-linear $\sigma$-model[28] of
1007 Klimčík and Ševera [32–34, 61]. The model describes a string moving in a Lie group
1008 target manifold $\mathcal{G}$. The model can be used to study an open or closed string, with the
1009 resulting symmetries being heavily dependent on this choice. Unlike the original analyses
1010 which were done from a Hamiltonian point of view using a convenient foliation (or space-
1011 time coordinatization) of the 2D worldsheet, we will adopt the manifestly-covariant and
1012 coordinate-independent CPS formalism.

---

[28]Here, we only consider the KS model in its version *without* "spectator coordinates". See [36] and
references therein.

To the best of our knowledge, a CPS analysis of such a model has not been done before. Beside offering a better understanding of its covariance properties and the origin of the "Poisson-Lie T-duality" discovered by Klimčík and Ševera, the analysis presented here suggests some (slight) generalization of the original model and most importantly sheds new light on issues of locality in the Poisson-Lie symmetry context.

## 4.1 Formulating the Covariant Klimčík & Ševera Open String

This section serves as a first introduction to the Klimčík & Ševera (KS) model as we formulate it covariantly. We establish the notations relevant to this entire chapter and construct the Lagrangian of the model. A key proposition establishing a tight link between the geometry of the target space and the Poisson-Lie notion of a classical double is proven, and a preliminary CPS analysis of the model including its (pre-)symplectic structure and equations of motion is given.

**Notations.** Consider a field

$$\tilde{h} : \Sigma \to \mathcal{G}, \quad x \mapsto \tilde{h}(x). \tag{123}$$

valued in the (real) finite dimensional Lie group $\mathcal{G}$, with Lie algebra $\mathfrak{g}$, on a 2-dimensional spacetime[29] $\Sigma$. We call $\Sigma$ the *worldsheet*, and $\mathcal{G}$ the *target space*. The (Lorentzian) metric on $\Sigma$ is denoted $\gamma_{\mu\nu}$ and is here considered fixed once and for all. We use Greek indices $\mu, \nu, \ldots$ on the worldsheet and Latin indices $a, b, c, \ldots$ on $\mathfrak{g}$ or $\mathfrak{g}^*$.

In the following, unless otherwise and explicitly specified, we consider open strings only, meaning

$$\Sigma \simeq [0, \pi] \times \mathbb{R}, \tag{124}$$

leaving our discussion of the closed string for §4.4. This will require introducing boundary conditions to ensure the conservation of the charges.

There are two different spaces on which we want to be doing differential geometry.[30] Following our notation for the CPS formalism, we will use $(d, i_\bullet, L_\bullet)$ for Cartan's calculus on $\Sigma$, and $(\mathbb{d}, \mathbb{i}_\bullet, \mathbb{L}_\bullet)$ for the calculus on the infinite dimensional field space:

$$\mathcal{F} := C^\infty(\Sigma, \mathcal{G}) \ni \tilde{h}. \tag{125}$$

Denote by $\Delta \tilde{h}$ the pullback to $\Sigma$ by a field $\tilde{h} : \Sigma \to \mathcal{G}$ of the right Maurer-Cartan form on $\mathcal{G}$:

$$\Delta \tilde{h} := d\tilde{h}\tilde{h}^{-1} \in \Omega^1(\Sigma) \otimes \mathfrak{g}. \tag{126}$$

We will sometimes adopt the following mixed notations:

$$\Delta \tilde{h} \equiv (\Delta_\mu^a \tilde{h})\tau_a dx^\mu \equiv \nabla_\mu \tilde{h}\tilde{h}^{-1} dx^\mu, \tag{127}$$

with $\nabla_\mu$ the Levi-Civita covariant derivative on $(\Sigma, \gamma_{\mu\nu})$ and $\tau_a$ a basis of $\mathfrak{g}$.

The group multiplication on $\mathcal{G}$ lifts pointwise to field space, turning $\mathcal{F}$ into an infinite dimensional Lie group. Denote $\text{Lie}(\mathcal{F}) := C^\infty(\Sigma, \mathfrak{g})$ its Lie algebra, and $\mathbb{\Delta}\tilde{h}$ the left Maurer-Cartan form on $\mathcal{F}$:

$$\mathbb{\Delta}\tilde{h} := \tilde{h}^{-1}\mathbb{d}\tilde{h} \in \Omega^1(\mathcal{F}) \otimes \text{Lie}(\mathcal{F}). \tag{128}$$

---

[29]Note the change in notation for "spacetime" from $M \rightsquigarrow \Sigma$.

[30]Differential geometry on the finite dimensional target space $\mathcal{G}$ is also important, but can be somewhat circumvented. For this reason, we avoid introducing a third, separate, notation for it.

1044  The two "deltas", $\Delta$ and $\mathbb{A}$, satisfy the following useful identity:

$$\mathbb{d}\Delta\tilde{h} = \mathrm{Ad}_{\tilde{h}}(d\mathbb{A}\tilde{h}) = \tilde{h}(d\mathbb{A}\tilde{h})\tilde{h}^{-1} \tag{129}$$

1045  where $\mathrm{Ad}_{\tilde{h}}$ is the adjoint action of $\tilde{h} \in \mathcal{G}$ on $\mathfrak{g}$, lifted pointwise to the adjoint action
1046  (denoted with the same symbol) of $\mathcal{F}$ on $\mathrm{Lie}(\mathcal{F})$.

1047  **The Model.**  Over the field space $\mathcal{F}$, consider the following second order Lagrangian of
1048  Klimčík and Ševera [32, 34], here only slightly generalized, and written in a manifestly
1049  covariant form:

$$\mathcal{L}(x, \tilde{h}) = \frac{1}{2} E_{ab}^{\mu\nu}(\tilde{h}(x))\Delta_\mu^a \tilde{h}(x)\Delta_\nu^b \tilde{h}(x). \tag{130}$$

1050  Henceforth, more succinctly:

$$\mathcal{L} := \frac{1}{2} E_{ab}^{\mu\nu}\Delta_\mu^a\tilde{h}\Delta_\nu^b\tilde{h}. \tag{131}$$

1051  Without loss of generality, we assume that $E_{ab}^{\mu\nu}$ is "symmetric" in $(a,\mu) \leftrightarrow (b,\nu)$, namely
1052  $E_{ab}^{\mu\nu} = E_{ba}^{\nu\mu}$. Moreover we demand that $E_{ab}^{\mu\nu}$ is *invertible*, meaning that there exists a $\widetilde{E}_{\mu\nu}^{ab}$
1053  such that

$$\delta_\rho^\mu \delta_a^c = E_{ab}^{\mu\nu}\widetilde{E}_{\nu\rho}^{bc}, \tag{132}$$

1054  and that $E_{ab}^{\mu\nu}$ satisfies the *KS condition*:

$$\mathbb{d}E_{ab}^{\mu\nu} =: E_{abc}^{\mu\nu}\mathbb{A}^c\tilde{h}, \quad E_{abc}^{\mu\nu} := \tilde{c}^{a'b'}{}_c E_{aa''}^{\mu\mu'}E_{bb''}^{\nu\nu'}(\mathrm{Ad}_{\tilde{h}})^{a''}{}_{a'}(\mathrm{Ad}_{\tilde{h}})^{b''}{}_{b'}\epsilon_{\mu'\nu'} \tag{133}$$

1055  for $\epsilon_{\mu\nu}$ the covariant Levi-Civita tensor on the worldsheet $(\Sigma, \gamma_{\mu\nu})$, and $\tilde{c}^{ab}{}_c$ structure
1056  constants of a (finite dimensional) Lie algebra defined on the dual vector space $\mathfrak{g}^*$ in the
1057  dual basis $\tau_*^a$. We have also introduced the following notation for the matrix elements of
1058  the adjoint action in $\mathfrak{g}$:

$$(\mathrm{Ad}_{\tilde{h}})^b{}_c := \langle \tau_*^b, \mathrm{Ad}_{\tilde{h}}\tau_c \rangle. \tag{134}$$

1059  The yet-unspecified tensor $E_{ab}^{\mu\nu}$ encodes all the important information of the non-linear
1060  $\sigma$-model. If $E_{ab}^{\mu\nu}$ were equal to $\gamma^{\mu\nu}\kappa_{ab}$, for $\kappa_{ab}$ a non-degenerate, symmetric, bilinear,
1061  $\mathrm{Ad}$-invariant 2-form on $\mathfrak{g}$, then we would be looking at the principal chiral model for
1062  which $\mathbb{d}E_{ab}^{\mu\nu} = 0$ (cf. §4.2). The KS condition is meant to generalize this case while
1063  preserving, and deforming, all its relevant symmetries. In this sense, the KS condition
1064  (133) is responsible for the Poisson-Lie symmetries of the model as we will soon see in
1065  detail.

1066  As mentioned, by considering an open string we require appropriate boundary conditions
1067  in order to have a well-posed (and non-Abelian) Poisson-Lie symmetry with conserved
1068  charges. Namely, the following "Neumann" open-string boundary conditions need to be
1069  imposed at the string's timelike boundary $B = \partial\Sigma \simeq \{0, \pi\} \times \mathbb{R}$ [34]:

$$s_\mu E_{ab}^{\mu\nu}\Delta_\nu^b\tilde{h}|_B = 0. \tag{135}$$

1070  where $s_\mu$ is the spacelike unit conormal to $B$. As we will see, these open-string boundary
1071  conditions have two advantages: they give rise to a conserved symplectic structure and
1072  they are compatible with the relevant symmetries – thus leading to conserved charges.
1073  (Note that Dirichlet boundary conditions, $\tilde{h}|_{\partial\Sigma} = 1$, would not be symmetry-compatible;
1074  this will become obvious later on.)[31]

---

[31]See [62] for a discussion of Dirichlet boundary conditions, generalized to $\tilde{h}|_{\partial\Sigma}$ belonging to a dressing orbit, or D-brane. These D-brane conditions emerge naturally in the dual model from the present Neumann conditions in the original model.

1075 The following Proposition shows that for the KS condition (133) to have solutions, the
1076 structure constants $c_{ab}{}^c$ and $\tilde{c}^{ab}{}_c$ of $\mathfrak{g}$ and $\mathfrak{g}^*$ are such that they define a classical double
1077 $\mathfrak{d} = \mathfrak{g} \bowtie \mathfrak{g}^*$ in the sense of Definition 2.12:

1078 **Proposition 4.1** (Klimčík and Ševera [32])**.** Consider an invertible mixed tensor $E_{ab}^{\mu\nu}$, then
1079 the KS condition (133) is satisfied only if $\mathfrak{g}$ and $\mathfrak{g}^*$ form a classical double $\mathfrak{d} := \mathfrak{g} \bowtie \mathfrak{g}^*$. $\diamond$

1080 *Proof.* First, using the invertibility condition (132), we have

$$\mathsf{d}\widetilde{E}^{bc}_{\nu\rho} = -\widetilde{E}^{bd}_{\nu\tau}\mathsf{d}E^{\tau\sigma}_{de}\widetilde{E}^{ec}_{\sigma\rho}. \tag{136}$$

1081 allowing (133) to be re-written more succinctly in terms of the inverse $\widetilde{E}^{ab}_{\mu\nu}$:

$$\mathsf{d}\widetilde{E}^{ab}_{\mu\nu} = -\tilde{c}^{a'b'}{}_c(\mathrm{Ad}_{\tilde{h}})^a{}_{a'}(\mathrm{Ad}_{\tilde{h}})^b{}_{b'}\Delta^c\tilde{h}\,\epsilon_{\mu\nu}. \tag{137}$$

1082 This is equivalent to demanding that, for all $X := X_a\tau_*^a$ and $Y := Y_a\tau_*^a$ elements of $\mathfrak{g}^*$,

$$\mathsf{d}(X_aY_b\widetilde{E}^{ab}_{\mu\nu}) = X_aY_b\mathsf{d}\widetilde{E}^{ab}_{\mu\nu} = -\langle[\mathrm{Ad}^*_{\tilde{h}}X, \mathrm{Ad}^*_{\tilde{h}}Y]_*, \Delta\tilde{h}\rangle\epsilon_{\mu\nu}, \tag{138}$$

1083 Therefore, a necessary condition for (133) to have solutions, is that for all $X, Y \in \mathfrak{g}^*$ the
1084 variation of the right hand side vanishes:

$$0 \equiv \mathsf{d}^2(X_aY_b\widetilde{E}^{ab}_{\mu\nu}) = -\mathsf{d}\langle[\mathrm{Ad}^*_{\tilde{h}}X, \mathrm{Ad}^*_{\tilde{h}}Y]_*, \Delta\tilde{h}\rangle\epsilon_{\mu\nu} \tag{139}$$

1085 After some algebra this gives the necessary condition:

$$\langle[X^{\tilde{h}}, Y^{\tilde{h}}]_*, \frac{1}{2}[\Delta\tilde{h}, \Delta\tilde{h}]\rangle = \langle[\mathrm{ad}^*_{\Delta\tilde{h}}X^{\tilde{h}}, Y^{\tilde{h}}]_*, \Delta\tilde{h}\rangle + \langle[X^{\tilde{h}}, \mathrm{ad}^*_{\Delta\tilde{h}}Y^{\tilde{h}}]_*, \Delta\tilde{h}\rangle \tag{140}$$

1086 where we used $\mathsf{d}\mathrm{Ad}^*_{\tilde{h}}X = \mathrm{ad}^*_{\Delta\tilde{h}}\mathrm{Ad}^*_{\tilde{h}}X$ and introduced for the sake of this proof the more
1087 compact notation $X^{\tilde{h}} \equiv \mathrm{Ad}^*_{\tilde{h}}X = X \lhd \tilde{h}$ etc.

1088 Evaluating this at a fixed $x_0 \in \Sigma$, and noting that $\Delta\tilde{h}(x_0)$ can be identified with the left
1089 Maurer-Cartan form on $\mathcal{G}$ at $g = \tilde{h}(x_0) \in \mathcal{G}$, we find that the above is satisfied if and only
1090 if for all $\alpha, \beta \in \mathfrak{g}$

$$\begin{aligned}\langle[X^g, Y^g]_*, [\alpha, \beta]\rangle = &\langle[\mathrm{ad}^*_\alpha X^g, Y^g]_*, \beta\rangle + \langle[X^g, \mathrm{ad}^*_\alpha Y^g]_*, \beta\rangle \\ &- \langle[\mathrm{ad}^*_\beta X^g, Y^g]_*, \alpha\rangle - \langle[X^g, \mathrm{ad}^*_\beta Y^g]_*, \alpha\rangle.\end{aligned} \tag{141}$$

1091 Marginalizing over $X, Y, \alpha, \beta$, this integrability condition gives the following condition on
1092 the structure constants $c_{ab}{}^c$ and $\tilde{c}^{ab}{}_c$ of $\mathfrak{g}$ and $\mathfrak{g}^*$, respectively:

$$c_{ab}{}^c\tilde{c}^{de}{}_c = c_{ac}{}^d\tilde{c}^{ce}{}_b + c_{ac}{}^e\tilde{c}^{dc}{}_b - c_{bc}{}^d\tilde{c}^{ce}{}_a - c_{bc}{}^e\tilde{c}^{dc}{}_a. \tag{142}$$

1093 This is the Jacobi identity (54) for the classical double. $\qquad\square$

1094 This proposition gives us a necessary condition for the existence of solutions to the KS
1095 equation. Klimčík and Ševera showed how to build explicit solutions to it by writing $\widetilde{E}^{ab}_{\mu\nu}$
1096 in terms of the corresponding Drinfel'd Poisson bivector on $\mathcal{G}$ [32–34, 61].

1097 We review and slightly generalize their construction in Appendix A.

1098 As discovered by Klimčík and Ševera, the relationship between the KS model and Poisson-
1099 Lie groups does not stop here. We will see in the later sections that the CPS, as determined
1100 by the dynamics of the model, displays a Poisson-Lie symmetry as a generalized Noether

symmetry. For this we need to study the equations of motion and (pre-)symplectic structure of the model.

Using that $\mathbb{\Delta}\tilde{h}$ is a basis of source forms on $\mathcal{F}$, we write the variation of the Lagrangian in terms of the equations of motion $\mathcal{E}_c$ and the pSP current $\theta^\mu$ according to:

$$\mathbb{d}\mathcal{L} =: \mathcal{E}_c \mathbb{\Delta}^c \tilde{h} + \nabla_\mu \theta^\mu. \tag{143}$$

where

$$\mathcal{E}_c = \frac{1}{2} E_{abc}^{\mu\nu} \Delta_\mu^a \tilde{h} \Delta_\nu^b \tilde{h} - \nabla_\mu j_c^\mu \quad \text{and} \quad \theta^\mu = \langle j^\mu, \mathbb{\Delta}\tilde{h} \rangle, \tag{144}$$

with[32]

$$j^\nu := \text{Ad}_{\tilde{h}}^* \left( E_{ab}^{\mu\nu} \Delta_\mu^a \tilde{h} \, \tau_*^b \right) \quad \text{or, equivalently,} \quad j_c^\nu = \left( E_{ab}^{\mu\nu} \Delta_\mu^a \tilde{h} \right) (\text{Ad}_{\tilde{h}})^b{}_c. \tag{145}$$

The pSF current is then

$$\omega^\mu = \mathbb{d}\theta^\mu = \langle \mathbb{d}j^\mu, \mathbb{\Delta}\tilde{h} \rangle - \frac{1}{2} \langle j^\mu, \left[ \mathbb{\Delta}\tilde{h}, \mathbb{\Delta}\tilde{h} \right] \rangle. \tag{146}$$

For an open string, the worldsheet $\Sigma$ has a timelike boundary $B = \{0, \pi\} \times \mathbb{R}$. It is then not a priori guaranteed that the symplectic structure $\Omega \approx \int_C n_\mu \omega^\mu$ is conserved, where $C \simeq [0, \pi]$ is a Cauchy surface. However, conservation is indeed achieved if the open-string boundary conditions (135) are imposed. Note that they can be rewritten as

$$s_\mu j^\mu|_{\partial\Sigma} = 0. \tag{147}$$

Indeed, from the on-shell conservation of the pSF current $\omega^\mu$ (13) it follows that:

$$\nabla_\mu \omega^\mu \approx 0 \implies \Omega|_{C_2} - \Omega|_{C_1} \approx \int_{C_2 \sqcup \overline{C}_1} n_\mu \omega^\mu \approx \int_{B_\pi \sqcup B_0} s_\mu \omega^\mu \approx \int_{B_\pi \sqcup B_0} \mathbb{d}\langle s_\mu j^\mu, \mathbb{\Delta}\tilde{h} \rangle = 0. \tag{148}$$

This concludes our formulation of the covariant KS model. We begin our exposition of the symmetries of the model in the next section with the simpler "warm up" case in which $E_{ab}^{\mu\nu}$ is taken to be field independent.

## 4.2  Warm up: Noetherian Symmetries of the Principal Chiral Model

To highlight the distinction between Noetherian symmetries and Poisson-Lie symmetries, we first illustrate the usual Noetherian symmetry of the principal chiral model obtained by considering a field-independent $E_{ab}^{\mu\nu}$. In section §4.3, we will then see how the Poisson-Lie symmetries of the KS model arise as a "deformation" of the Noetherian symmetries discussed here.

By definition $E_{ab}^{\mu\nu}$ is field-independent if and only if $\mathbb{d}E_{ab}^{\mu\nu} = 0$. From (133) it is clear this holds if and only if the dual structure constants vanish, $\tilde{c}^{ab}{}_c = 0$. This, in particular, implies that $E_{ab}^{\mu\nu}(\tilde{h})$ is invariant under ($\Sigma$-global) right rotations, $\tilde{h} \to \tilde{h}h_0$ with $h_0 \in \mathcal{G}$:

$$E_{ab}^{\mu\nu}(\tilde{h}h_0) = E_{ab}^{\mu\nu}(\tilde{h}). \tag{149}$$

---

[32]Recall that in our notation $\text{Ad}_{\tilde{h}}^*$ is the dual map to $\text{Ad}_{\tilde{h}}$, which acts from the right: $\langle \text{Ad}_{\tilde{h}}^* X, \alpha \rangle = \langle X, \text{Ad}_{\tilde{h}} \alpha \rangle$.

Since $\Delta^a_\mu \tilde{h}$ is invariant under right rotations, it is clear then that right rotations are Noetherian symmetries that leave the Lagrangian (130) invariant. Taking the infinitesimal right rotation $h_0 = 1 + \alpha + \cdots$, $\alpha \in \mathfrak{g}$, we denote its action

$$\mathbb{r} : \mathfrak{g} \to \mathfrak{X}(\mathcal{F}), \quad \alpha \mapsto \mathbb{r}(\alpha) \quad \text{with} \quad \left(\mathbb{r}(\alpha)f\right)(\tilde{h}) := \frac{d}{dt}\bigg|_{t=0} f(\tilde{h}e^{t\alpha}) \quad \text{iff} \quad \mathring{\mathbb{i}}_{\mathbb{r}(\alpha)}\Delta\tilde{h} = \alpha. \tag{150}$$

so that indeed

$$\mathbb{L}_{\mathbb{r}(\alpha)}\mathcal{L} = 0. \tag{151}$$

The corresponding conserved Noether current equals $j^\mu$ itself,

$$j^\mu = \mathring{\mathbb{i}}_{\mathbb{r}(\cdot)}\theta^\mu, \tag{152}$$

while the equations of motion (144) reduce to the conservation equations for $j^\mu_c$ (145):

$$\mathbb{d}E^{\mu\nu}_{ab} = 0 \implies \mathcal{E}_c = -\nabla_\mu j^\mu_c \approx 0. \tag{153}$$

Moreover, as one can see from the first expression in (145), $j^\mu$ is equivariant under right rotations, that is

$$j^\mu \mapsto \mathrm{Ad}^*_{h_0} j^\mu. \tag{154}$$

With this, we see that the infinitesimal transformations of $\tilde{h}$ and $j$ are

$$\tilde{h}^{-1}\delta^R_\alpha \tilde{h} = \mathring{\mathbb{i}}_{\mathbb{r}(\alpha)}\Delta\tilde{h} = \alpha \tag{155}$$

$$\delta^R_\alpha j^\mu = \mathbb{L}_{\mathbb{r}(\alpha)} j^\mu = \mathrm{ad}^*_\alpha j^\mu, \tag{156}$$

thus implying that (cf. (146))

$$\mathring{\mathbb{i}}_{\mathbb{Q}(\alpha)}\omega^\mu = -\mathbb{d}\langle j^\mu, \alpha \rangle. \tag{157}$$

Assuming $\Sigma$ is globally hyperbolic and choosing a Cauchy surface $C \hookrightarrow \Sigma$ of the worldsheet we recover a flow equation (cf. (18)) by integrating (157) over $C$,

$$\mathring{\mathbb{i}}_{\mathbb{Q}(\alpha)}\Omega = -\mathbb{d}\langle q, \alpha \rangle, \quad q := \int_C n_\mu j^\mu, \tag{158}$$

where on-shell, neither $q$ nor $\Omega$ dependss on the choice of $C \hookrightarrow \Sigma$. We emphasize again that this $C$-invariance is the covariant field space version of the statement that $q$ is conserved and generates a Noetherian symmetry, reflecting the usual notion of momentum map.

If the target space is taken to be $\mathcal{G} \simeq (\mathbb{R}^n, +)$, then this corresponds to the conservation of the total linear momentum of the string moving in $\mathbb{R}^n$.

These results are reminiscent of our comments in §2.3, ascertaining that if $\mathfrak{g}^*$ is Abelian, then the Poisson-Lie variational charge reduces to a $\mathbb{d}$-exact Noether variational charge. Unsurprisingly then, in the next sections we will see how having $\mathfrak{g}^*$ non-Abelian deforms the Noether symmetry discussed here into a Poisson-Lie symmetry.

## 4.3 Poisson-Lie Symmetries of the Covariant Klimčík & Ševera Open String

We will now illustrate how Poisson-Lie symmetries arise in the Klimčík and Ševera model. Just as the invariance condition (149) gave rise to a Noether symmetry, it is the generalization of this condition into one of the form of (133) with non-zero structure constants $\tilde{c}^{ab}{}_c \neq 0$ of the dual Lie algebra $\mathfrak{g}^*$ which will give rise to a Poisson-Lie symmetry. There

are three properties of the model which are beautifully tied to each other in this gener-alization: *(i)* $\mathfrak{g}$ and $\mathfrak{g}^*$ form a classical double *(ii)* the equations of motion take on the form of a spacetime flatness equation for $j^\mu$ *(iii)* there exists a symmetry admitting a Poisson-Lie momentum map.

Crucially, the corresponding symmetry action *fails* both to leave the Lagrangian invariant (even up to a boundary term), and (in a second order formalism) to be spacetime local. Therefore, the PL symmetries of the KS model go genuinely beyond the usual Noetherian framework.

### 4.3.1 Deformed Conservation: Flatness

Imposing (133) with $\tilde{c}^{ab}{}_c \neq 0$ causes the usual Noether conservation equation (153) to be "deformed" into a *spacetime* flatness condition.[33] A $\mathfrak{g}^*$-valued 1-form $J = J_{a\mu}\tau_*^a d\sigma^\mu$ is said to be flat if and only if the 2-form

$$F_c := dJ_c - \frac{1}{2}\tilde{c}^{ab}{}_c J_a \wedge J_b \tag{159}$$

vanishes (the relative sign is conventional, since it can be flipped by sending $J \mapsto -J$).

By means of a Hodge dualization with respect to the (Lorentzian) worldsheet metric $\gamma_{\mu\nu}$, the 1-form $J$ can be mapped onto a current $j$:

$$J = \star(\gamma_{\mu\nu}j^\mu dx^\nu) \quad \text{i.e.} \quad J_\nu = j^\mu\epsilon_{\mu\nu}. \tag{160}$$

Then, flatness of $J$ holds if and only if $j$ satisfies the "deformed" conservation law,

$$-\frac{1}{2}\epsilon^{\mu\nu}F_{\mu\nu} = \nabla_\mu j^\mu - \frac{1}{2}[j^\mu, j^\nu]_*\epsilon_{\mu\nu} \approx 0. \tag{161}$$

Counting form degrees, it becomes clear that *this deformation of the conservation equation for j is specific to 2D spacetimes.*[34]

The key point is that the deformed conservation law for the current $j^\mu$ is exactly the equation of motion of the KS model (130) if and only if (133) holds:

**Proposition 4.2.** The KS model (130) admits the deformed conservation law as its equation of motion,

$$\mathcal{E}_c = \frac{1}{2}\epsilon^{\mu\nu}F_{\mu\nu c} = -\left(\nabla_\mu j_c^\mu - \frac{1}{2}[j^\mu, j^\nu]_{*c}\epsilon_{\mu\nu}\right) \tag{162}$$

if and only if $\mathbb{d}E_{ab}^{\mu\nu}$ is given by the KS condition (133). $\diamond$

*Proof.* From the general equations of motion (144), plug in (133), group terms into $j_c^\nu$ factors:

$$\mathcal{E}_c = \frac{1}{2}E_{abc}^{\mu\nu}\Delta_\mu^a h \Delta_\nu^b h - \nabla_\mu j_c^\mu = \frac{1}{2}\tilde{c}^{a'b'}{}_c\Delta_\mu^a h \Delta_\nu^b h E_{aa''}^{\mu\mu'}E_{bb''}^{\nu\nu'}(\mathrm{Ad}_h)^{a''}{}_{a'}(\mathrm{Ad}_h)^{b''}{}_{b'}\epsilon_{\mu'\nu'} - \nabla_\mu j_c^\mu$$

$$= \frac{1}{2}\tilde{c}^{a'b'}{}_c(E_{aa''}^{\mu\mu'}\Delta_\mu^a h(\mathrm{Ad}_h)^{a''}{}_{a'})(E_{bb''}^{\nu\nu'}\Delta_\nu^b h(\mathrm{Ad}_h)^{b''}{}_{b'})\epsilon_{\mu'\nu'} - \nabla_\mu j_c^\mu$$

---

[33]This spacetime flatness equation, which is highly dependent on the model at hand, is not to be confused with the field space flatness condition (34) satisfied in general by Poisson-Lie variational charges. Although, the two are related as we will soon see.

[34]Recall all currents, by namesake, are codimension-1 forms in spacetime. This fact plus the fact that $j^\mu$ must be a 1-form to satisfy the flatness equation (161), restricts these Klimčík & Ševera type models to 2D spacetimes. As motivated in §2.3 and as we will later discuss in §5, not all Poisson-Lie symmetric field theories are restricted to 2d (or lower dimensional) spacetimes.

$$= \frac{1}{2}\tilde{c}^{a'b'}{}_c j_{a'}^{\mu'} j_{b'}^{\nu'} \epsilon_{\mu'\nu'} - \nabla_\mu j_c^\mu = -\nabla_\mu j_c^\mu + \frac{1}{2}[j^\mu, j^\nu]_{*c}\epsilon_{\mu\nu} \tag{163}$$

$\square$

An important implication of this equation of motion which will be crucial to our discussion is the form of its solutions. If $\Sigma \simeq [0,\pi] \times \mathbb{R}$ is the worldsheet of an open string (more generally, if it is simply connected) then we have that, on-shell, (162) implies that there exists a map

$$\tilde{\ell} : \mathcal{F} \to C^\infty(\Sigma, \mathcal{G}^*) \tag{164}$$

such that

$$J \approx \Delta\tilde{\ell} \iff j^\mu \approx \nabla_\nu \tilde{\ell}\, \tilde{\ell}^{-1} \epsilon^{\nu\mu}, \tag{165}$$

where $\mathcal{G}^*$ denotes the (simply connected) Lie group induced by exponentiating $(\mathfrak{g}^*, [\cdot,\cdot]_*)$. There is of course an ambiguity in the definition of $\tilde{\ell}$ from $J$. We fix it by picking on $\partial\Sigma$ a point $x_0$, and defining $\tilde{\ell}$ at any $x \in \Sigma$ as the path-ordered exponential of $J$ from $x_0$ to $x$:

$$\tilde{\ell}(x) :\approx \text{Pexp} \int_{x \leftarrow x_0} J. \tag{166}$$

This way, $\tilde{\ell}(x_0) \equiv 1 \in \mathcal{G}^*$. Since on-shell $J$ is flat, all homotopically equivalent paths from $x_0$ to $x$ will give the same result.

In fact, due to the boundary conditions (147), the pullback of $J$ to the boundary $\partial\Sigma = \{0,\pi\} \times \mathbb{R}$ vanishes[35], thus implying that

$$\iota_{\partial\Sigma}^* J = 0 \implies \tilde{\ell}|_{\{0\}\times\mathbb{R}} \approx 1 \quad \text{and} \quad \tilde{\ell}|_{\{\pi\}\times\mathbb{R}} \approx const, \tag{167}$$

where we assumed that $x_0$ belongs to the $\{0\} \times \mathbb{R}$ boundary component. As we will see the constant value $Q := \tilde{\ell}(\pi, \tau) \in \mathcal{G}^*$, with $\tau \in \mathbb{R}$ arbitrary but fixed (to denote a specific Cauchy slice in the foliation $\Sigma \simeq C \times \mathbb{R}$, $C \simeq [0,\pi]$), is the conserved Poisson-Lie charge of the open string (cf. Theorem 4.8). Had we chosen $x_0$ in the bulk of $\Sigma$, $\tilde{\ell}$ would still be constant on each boundary component, while the relevant $Q$ would be $\tilde{\ell}(\pi,\tau)\tilde{\ell}^{-1}(0,\tau)$.

### 4.3.2 Twisted Right Rotations & Their Poisson-Lie Charge

In the previous section we have seen that for the equations of motion to take the form of a (dual) flatness equation, $E_{ab}^{\mu\nu}$ must have a specific field-dependence expressed by the KS condition (133). Moreover, we established with Proposition 4.1 that this field dependence of $E_{ab}^{\mu\nu}$ has solutions only if $\mathfrak{g} \bowtie \mathfrak{g}^*$ has the structure of a (classical) Drinfel'd double:

$$\mathfrak{d} = \mathfrak{g} \bowtie \mathfrak{g}^*. \tag{168}$$

This condition is the main ingredient behind the results of this section.

We now show that the so-defined KS model enjoys a generalization of the right rotational symmetry of the principal chiral model. As we shall see, this symmetry is non-local and can be expressed as a "twisted" right rotation of the field $\tilde{h}^{-1}\delta_\alpha \tilde{h} = \overline{\alpha}$ where the right rotation parameter $\overline{\alpha}$ is *field dependent*, namely $\overline{\alpha} := \widetilde{\text{Ad}}^*_{\tilde{\ell}^{-1}}\alpha$, with $\widetilde{\text{Ad}}^*$ denoting the transpose of the (dual) adjoint action $\widetilde{\text{Ad}}$ of $\mathcal{G}^*$ on $\mathfrak{g}^*$. That is, for all $\tilde{\ell} \in \mathcal{G}^*, X \in \mathfrak{g}^*$ and $\alpha \in \mathfrak{g}$, $\langle \widetilde{\text{Ad}}_{\tilde{\ell}^{-1}}(X), \alpha\rangle =: \langle X, \widetilde{\text{Ad}}^*_{\tilde{\ell}^{-1}}\alpha\rangle$.[36]

---

[35]Indeed, if $t^\mu$ is tangent to $\partial\Sigma$ and $\epsilon_{\mu\nu} \propto (\underline{n} \times \underline{s})_{\mu\nu}$ with $n_\mu$ the conormal to $C$ and $s_\nu$ the conormal to $\partial\Sigma$, using $t^\mu s_\mu = 0$, it is easy to see that $t^\mu J_\mu \propto s_\mu j^\mu$. This does not depend on the choice of $C \hookrightarrow \Sigma$.

[36]With this notation [52], the co-adjoint *representation* of $\mathcal{G}^*$ over $\mathfrak{g}$ is given by $(\widetilde{\text{Ad}}^*)^{-1}$, which is a *left* action.

More in detail, recall that the action of a right rotation on $\mathcal{F}$ is defined as the map

$$\mathbb{r} : \mathfrak{g} \to \mathfrak{X}(\mathcal{F}) \quad \text{such that} \quad \left( \mathbb{i}_{\mathbb{r}(\alpha)} \Delta \tilde{h} \right)(x) = \alpha \quad \forall x \in \Sigma. \tag{169}$$

**Definition 4.3** (Twisted right rotation). *Twisted* right rotations[37]

$$\mathbb{Q} : \mathfrak{g} \to \mathfrak{X}(\mathcal{F}_{\mathrm{EL}}) \tag{170}$$

are the field-space vector fields defined by

$$\left( \mathbb{i}_{\mathbb{Q}(\alpha)} \Delta \tilde{h} \right)(x) :\approx \widetilde{\mathrm{Ad}}^*_{\tilde{\ell}^{-1}(x)} \alpha \quad \forall \alpha \in \mathfrak{g}, x \in \Sigma \tag{171}$$

where $\tilde{\ell}(x) \approx \mathrm{Pexp} \int_{x \leftarrow x_0} J$ (166) is a point-dependent, non-local, function of $\tilde{h} \in \mathcal{F}_{\mathrm{EL}}$. $\diamond$

Note that twisted right rotations differ from "standard" right rotations only if $\mathcal{G}^*$ is non-Abelian, i.e. $\widetilde{\mathrm{Ad}} \neq \mathrm{id}$. In particular, seen as vector fields on $\mathcal{F}_{\mathrm{EL}} \subset \mathcal{F}$, they are non-local.

We now prove that this action is well-defined, namely it is a homomorphism of Lie algebras, and defines a Poisson-Lie symmetry of the KS model. This means in particular that it is compatible with the equations of motion and, in the case of the open string, compatible with its Neumann boundary conditions. These facts rely on $(\mathfrak{g}, \mathfrak{g}^*)$ forming a Drinfel'd double structure.

Before formulating the main theorem, we prove some preliminary results.

**Lemma 4.4.** Recall that $\tilde{\ell}(x) \approx \mathrm{Pexp} \int_{x \leftarrow x_0} J$. Then, $\overline{\alpha}(x) \approx \widetilde{\mathrm{Ad}}^*_{\tilde{\ell}^{-1}(x)} \alpha$ iff

$$d_J \overline{\alpha} := d\overline{\alpha} + \widetilde{\mathrm{ad}}^*_J(\overline{\alpha}) \approx 0 \quad \text{and} \quad \overline{\alpha}(x_0) \approx \alpha. \tag{172}$$

$\diamond$

*Proof.* The first of (172) is an over determined first order PDE. If a solution exists, it is uniquely determined by the value of $\overline{\alpha}$ at (any) one point $x_0 \in \Sigma$. Crucially, $J \approx d\tilde{\ell} \, \tilde{\ell}^{-1}$ is flat *on-shell* whence this equation has solutions for any "initial" value $\alpha$ of $\overline{\alpha}$ at $x_0$. It is easily checked that this unique solution is $\overline{\alpha} :\approx \widetilde{\mathrm{Ad}}^*_{\tilde{\ell}^{-1}} \alpha$. Indeed, $\overline{\alpha}(x_0) = \alpha$ and, observing that $\overline{\alpha} = \widetilde{\mathrm{Ad}}^*_{\tilde{\ell}^{-1}} \alpha = \alpha \lhd \tilde{\ell}^{-1} = \tilde{\ell} \rhd \alpha$ coincides with the projection on the first factor of $\mathfrak{d} = \mathfrak{g} \bowtie \mathfrak{g}^*$ of the adjoint action of $\tilde{\ell} \in \mathcal{G}^* \subset \mathcal{D} := \exp(\mathfrak{d})$ on $\alpha \in \mathfrak{g}$ (56), we readily compute:

$$d\overline{\alpha} = d(\tilde{\ell} \, \alpha \, \tilde{\ell}^{-1})|_{\mathfrak{g}} = [\Delta \tilde{\ell}, \overline{\alpha}]_{\mathfrak{d}}|_{\mathfrak{g}} = -\widetilde{\mathrm{ad}}^*_{\Delta \tilde{\ell}} \overline{\alpha} \approx -\widetilde{\mathrm{ad}}^*_J \overline{\alpha}. \tag{173}$$

$\square$

As a consequence of (169) and (171) we can write the twisted right rotations as

$$\mathbb{Q}(\alpha) \approx \mathbb{r}(\overline{\alpha}), \tag{174}$$

which is nothing else than a convenient short-hand.

---

[37]At this stage we do not know whether twisted right rotations preserve the equations of motion. Therefore, it would be more accurate to define twisted right rotations as a map $\mathfrak{g} \times \mathcal{F}_{\mathrm{EL}} \to T_{\mathcal{F}_{\mathrm{EL}}} \mathcal{F}$, $(\alpha, \tilde{h}) \mapsto (\tilde{h}, \mathbb{Q}(\alpha)|_{\tilde{h}})$. However, we will prove in Proposition 4.6 that they are indeed well defined as vector fields on $\mathcal{F}_{\mathrm{EL}}$.

**Proposition 4.5.** Under the action of a twisted right rotation, the $\mathfrak{g}^*$-valued momentum current $j$ transforms as follows

$$\delta_\alpha j :\approx \mathbb{i}_{\mathbb{Q}(\alpha)} \mathbb{d} j \approx \mathrm{ad}^*_{\overline{\alpha}} j. \tag{175}$$

$\diamond$

*Proof.* We start by computing the variation of $j_c^\nu = E_{ab}^{\mu\nu}\Delta_\mu^a \tilde{h}(\mathrm{Ad}_{\tilde{h}})^b{}_c$. For this, we use the identities

$$\mathbb{d} E_{ab}^{\mu\nu} = \tilde{c}^{a'b'}{}_f E_{aa''}^{\mu\mu'} E_{bb''}^{\nu\nu'}(\mathrm{Ad}_{\tilde{h}})^{a''}{}_{a'}(\mathrm{Ad}_{\tilde{h}})^{b''}{}_{b'}\epsilon_{\mu'\nu'}\Delta^f\tilde{h} \tag{176}$$

$$\mathbb{d}\Delta_\mu^a\tilde{h} = (\mathrm{Ad}_{\tilde{h}})^a{}_{b'}\nabla_\mu\Delta^{b'}\tilde{h} \tag{177}$$

$$\mathbb{d}(\mathrm{Ad}_{\tilde{h}})^b{}_c = \Delta^f\tilde{h}\ c_{fc}{}^d(\mathrm{Ad}_{\tilde{h}})^b{}_d \tag{178}$$

to find:

$$\begin{aligned}
\mathbb{d} j_c^\nu =& \tilde{c}^{a'b'}{}_f E_{aa''}^{\mu\mu'} E_{bb''}^{\nu\nu'}(\mathrm{Ad}_{\tilde{h}})^{a''}{}_{a'}(\mathrm{Ad}_{\tilde{h}})^{b''}{}_{b'}\epsilon_{\mu'\nu'}\Delta^f\tilde{h}\Delta_\mu^a\tilde{h}(\mathrm{Ad}_{\tilde{h}})^b{}_c \\
&+ E_{b''b}^{\nu'\nu}(\mathrm{Ad}_{\tilde{h}})^{b''}{}_{b'}(\nabla_{\nu'}\Delta^{b'}\tilde{h})(\mathrm{Ad}_{\tilde{h}})^b{}_c + E_{ab}^{\mu\nu}\Delta_\mu^a\tilde{h}\Delta^f\tilde{h}\ c_{fc}{}^d(\mathrm{Ad}_{\tilde{h}})^b{}_d \tag{179} \\
=& E_{bb''}^{\nu\nu'}(\mathrm{Ad}_{\tilde{h}})^{b''}{}_{b'}J_{\nu'a'}\ \tilde{c}^{a'b'}{}_f\Delta^f\tilde{h}(\mathrm{Ad}_{\tilde{h}})^b{}_c \\
&+ E_{b''b}^{\nu'\nu}(\mathrm{Ad}_{\tilde{h}})^{b''}{}_{b'}(\nabla_{\nu'}\Delta^{b'}\tilde{h})(\mathrm{Ad}_{\tilde{h}})^b{}_c + \Delta^f\tilde{h}\ c_{fc}{}^d j_d^\nu \tag{180} \\
=& E_{ab}^{\mu\nu}(\mathrm{Ad}_{\tilde{h}})^a{}_{b'}(\mathrm{Ad}_{\tilde{h}})^b{}_c\left(\nabla_\mu\Delta^{b'}\tilde{h} + J_{\mu a'}\ \tilde{c}^{a'b'}{}_f\Delta^f\tilde{h}\right) + \Delta^f\tilde{h}\ c_{fc}{}^d j_d^\nu \tag{181}
\end{aligned}$$

where in the second step we used that $E_{ab}^{\mu\nu} = E_{ba}^{\nu\mu}$. Using the definition (171) of twisted right rotations, we infer then how $j$ transforms under $\mathbb{Q}$:

$$\delta_\alpha j_c^\nu :\approx \mathbb{i}_{\mathbb{Q}(\alpha)}\mathbb{d} j_c^\nu = E_{ab}^{\mu\nu}(\mathrm{Ad}_{\tilde{h}})^a{}_{b'}(\mathrm{Ad}_{\tilde{h}})^b{}_c\left(\nabla_\mu\overline{\alpha}^{b'} + J_{\mu a'}\ \tilde{c}^{a'b'}{}_f\overline{\alpha}^f\right) + \overline{\alpha}^f\ c_{fc}{}^d j_d^\nu, \tag{182}$$

We can conclude our proof if we show that the first term on the right hand side above vanishes. Indeed, since $\overline{\alpha} \approx \widetilde{\mathrm{Ad}}^*_{\ell-1}(\alpha)$, this follows from Lemma 4.4. To see that this is the case, namely that

$$\nabla_\mu\overline{\alpha}^{b'} + J_{\mu a'}\ \tilde{c}^{a'b'}{}_f\overline{\alpha}^f \approx 0, \tag{183}$$

we write it using explicitly the dual basis $\tau_*^{b'} \in \mathfrak{g}^*$ and re-arrange, giving,

$$0 \approx \langle \tau_*^{b'}, \nabla_\mu\overline{\alpha}\rangle + \langle \widetilde{\mathrm{ad}}_{J_\mu}(\tau_*^{b'}), \overline{\alpha}\rangle := \langle \tau_*^{b'}, \nabla_\mu\overline{\alpha} + \widetilde{\mathrm{ad}}^*_{J_\mu}(\overline{\alpha})\rangle. \tag{184}$$

We recognize in the second term the co-adjoint action of $J \in \mathfrak{g}^*$ on $\alpha \in \mathfrak{g}$, so that we can rewrite this expression in the desired form:

$$d\overline{\alpha} + \widetilde{\mathrm{ad}}^*_J(\overline{\alpha}) \approx 0. \tag{185}$$

$\square$

The proof of this proposition could be run backwards showing that there is only one twisting of the right action compatible with the "equivariance" of $j$, namely: $\mathbb{i}_{\mathbb{r}(\overline{\alpha})}\mathbb{d} j \approx \mathrm{ad}^*_{\overline{\alpha}} j$ if and only if $d_J\overline{\alpha} \approx 0$.

**Proposition 4.6** (Well-definedness). The twisted right rotations preserve the equations of motion, i.e. $\mathbb{L}_{\mathbb{Q}(\alpha)}\mathcal{E} \approx 0$ or equivalently

$$\delta_\alpha F \approx 0, \tag{186}$$

and—in the case of an open string—the Neumann boundary conditions $s_\mu j^\mu|_B \approx 0$ at $B = \{0, \pi\} \times \mathbb{R}$ ((135) and (147)). Twisted right rotations are therefore well-defined as vector fields over $\mathcal{F}_{\text{EL}}$.

Moreover, they define an infinitesimal action of $\mathfrak{g}$ on $\mathcal{F}_{\text{EL}}$ since they are a Lie algebra homomorphism, i.e.

$$[\mathcal{Q}(\alpha_1), \mathcal{Q}(\alpha_2)] \approx \mathcal{Q}([\alpha_1, \alpha_2]) \quad \forall \alpha_1, \alpha_2 \in \mathfrak{g}, \tag{187}$$

where the bracket on the left hand side is a Lie bracket of vector fields on $\mathcal{F}_{\text{EL}}$, and the bracket on the right hand side is the Lie bracket on $\mathfrak{g}$.      $\diamond$

*Proof.* To see that the twisted rotations preserve the equations of motion recall that $\mathcal{E} = -\frac{1}{2}\epsilon^{\mu\nu}F_{\mu\nu} \approx 0$. Therefore, $\mathbb{L}_{\mathcal{Q}(\alpha)}\mathcal{E} \approx 0$ iff $\delta_\alpha F \equiv \mathbb{L}_{\mathcal{Q}(\alpha)}F \approx 0$. This follows through a straightforward computation using the expression (160) for $J$, Lemma 4.4, and Proposition 4.5. That they preserve the boundary conditions follows directly from Proposition 4.5.

To prove $\mathcal{Q}$ is a Lie algebra homomorphism, we start from the fact that the right rotation $\mathbb{r}$ is known to be a Lie algebra homomorphism. Then, using the relationship $\mathcal{Q}(\alpha) \approx \mathbb{r}(\overline{\alpha})$, we compute[38]

$$[\mathcal{Q}(\alpha_1), \mathcal{Q}(\alpha_2)] \approx [\mathbb{r}(\overline{\alpha_1}), \mathbb{r}(\overline{\alpha_2})] \approx \mathbb{r}(\llbracket \overline{\alpha_1}, \overline{\alpha_2} \rrbracket) \quad \text{where} \quad \llbracket \overline{\alpha_1}, \overline{\alpha_2} \rrbracket :\approx [\overline{\alpha_1}, \overline{\alpha_2}] + \delta_1\overline{\alpha_2} - \delta_2\overline{\alpha_1}, \tag{188}$$

and $\delta_1\overline{\alpha_2} \equiv \mathbb{r}(\overline{\alpha_1})\overline{\alpha_2}$ etc. Therefore, our claim then follows if we can prove that

$$\llbracket \overline{\alpha_1}, \overline{\alpha_2} \rrbracket \approx \overline{[\alpha_1, \alpha_2]}. \tag{189}$$

Owing to Lemma 4.4 and using its notation, this holds iff

$$d_J\llbracket \overline{\alpha_1}, \overline{\alpha_2} \rrbracket \approx 0 \quad \text{and} \quad \llbracket \overline{\alpha_1}, \overline{\alpha_2} \rrbracket(x_0) \approx [\alpha_1, \alpha_2]. \tag{190}$$

We will now show that these two equations hold. We start from the second one.

Since $\tilde{\ell}(x_0) \equiv 1$ for every (on-shell) configuration $h$, one has $\overline{\alpha}(x_0) = \widetilde{\text{Ad}}^*_{\ell^{-1}(x_0)}\alpha \equiv \alpha$ and thus:

$$[\overline{\alpha_1}, \overline{\alpha_2}](x_0) \equiv [\overline{\alpha_1}(x_0), \overline{\alpha_2}(x_0)] \approx [\alpha_1, \alpha_2] \quad \text{and} \quad \delta_1\overline{\alpha_2}(x_0) \equiv \mathbb{i}_{\mathbb{r}(\overline{\alpha_1})}\mathbb{d}\overline{\alpha_2}(x_0) \approx 0 \tag{191}$$

where in the last equality we used the fact that the value $\alpha_2$ of $\overline{\alpha_2}$ at $x_0$ is independent of the field configuration: namely $\mathbb{d}\overline{\alpha_2}(x_0) = \mathbb{d}\alpha_2 = 0$, even before contraction with $\mathcal{Q}(\alpha_1)$. Combing these equations, it is then easy to see that the second of (190) follows from the definition of the bracket $\llbracket \cdot, \cdot \rrbracket$.

To show the first of the equations (190), we first observe that varying the defining equation of $\overline{\alpha_2}$, i.e. $d_J\overline{\alpha_2} \approx 0$, yields

$$0 \approx \mathbb{d}(d_J\overline{\alpha_2}) = d_J(\mathbb{d}\overline{\alpha_2}) + \widetilde{\text{ad}}^*_{\mathbb{d}_J\overline{\alpha_2}}, \tag{192}$$

which, when contracted with $\mathcal{Q}(\alpha_1)$, gives:

$$d_J(\delta_1\overline{\alpha_2}) \approx -\widetilde{\text{ad}}^*_{\delta_1 J}\overline{\alpha_2}. \tag{193}$$

---

[38]The third equality in this formula is valid for any (arbitrary) field-dependent symmetry parameters $\overline{\alpha}_{1,2} : \mathcal{F} \to C^\infty(\Sigma, \mathfrak{g})$, and not just those of the form we are interested in here. This formula can also be easily checked for matrix groups by explicitly computing $\mathbb{r}(\overline{\alpha_1})\mathbb{r}(\overline{\alpha_2})h - \mathbb{r}(\overline{\alpha_2})\mathbb{r}(\overline{\alpha_1})h$, using $\mathbb{r}(\overline{\alpha})h = h\overline{\alpha}$.

Using this result together with the equivariance of $J$, namely $\delta_1 J \approx \mathrm{ad}^*_{\overline{\alpha_1}} J$ (Proposition 4.5), we can now compute:

$$
\begin{aligned}
d_J[\![\overline{\alpha_1}, \overline{\alpha_2}]\!] &\approx d_J[\overline{\alpha_1}, \overline{\alpha_2}] + \widetilde{\mathrm{ad}}^*_{\delta_1 J} \overline{\alpha_2} - \widetilde{\mathrm{ad}}^*_{\delta_2 J} \overline{\alpha_1} \\
&\approx [d\overline{\alpha_1}, \overline{\alpha_2}] + [\overline{\alpha_1}, d\overline{\alpha_2}] + \widetilde{\mathrm{ad}}^*_J [\overline{\alpha_1}, \overline{\alpha_2}] + \widetilde{\mathrm{ad}}^*_{\delta_1 J} \overline{\alpha_2} - \widetilde{\mathrm{ad}}^*_{\delta_2 J} \overline{\alpha_1} \\
&\approx -[\widetilde{\mathrm{ad}}^*_J \overline{\alpha_1}, \overline{\alpha_2}] - [\overline{\alpha_1}, \widetilde{\mathrm{ad}}^*_J \overline{\alpha_2}] + \widetilde{\mathrm{ad}}^*_J [\overline{\alpha_1}, \overline{\alpha_2}] + \widetilde{\mathrm{ad}}^*_{\delta_1 J} \overline{\alpha_2} - \widetilde{\mathrm{ad}}^*_{\delta_2 J} \overline{\alpha_1} \\
&\approx [[\overline{\alpha_1}, \overline{\alpha_2}]_\mathfrak{d}, J]_\mathfrak{d} + [[\overline{\alpha_2}, J]_\mathfrak{d}, \overline{\alpha_1}]_\mathfrak{d} + [[J, \overline{\alpha_1}]_\mathfrak{d}, \overline{\alpha_2}]_\mathfrak{d} \equiv 0,
\end{aligned}
\tag{194}
$$

where the last step is most easily proved backwards and using equation (53) or, equivalently follows—after some careful bookkeeping—from the Drinfel'd double compatibility condition (54). This proves that equation (189) follows from (54), and therefore so does equation (187). $\qquad\square$

Note that Dirichlet boundary conditions, $\mathbb{d}\tilde{h}|_{\partial\Sigma} = 0$, for the open string would not be compatible with the twisted right rotations. As emphasized throughout, this is a consequence of dynamics really selecting the symmetries.

We can now prove our two main theorems of this section. In the first, we show how the non-local twisted right rotations encode a right dressing action on the Drinfel'd double. In the second, we provide a manifestly covariant, second order, version of results by Klimčík and Ševera [34], namely we prove that twisted right rotations are Poisson-Lie symmetries of the second order KS model. These two theorems will be brought together in the next section.

**Theorem 4.7** (Dressing action from twisted rotations). Recall $\mathcal{D} \simeq \mathcal{G} \bowtie \mathcal{G}^*$ and define the map between infinite dimensional manifolds

$$
\begin{aligned}
\Phi: \quad \mathcal{F}_{\mathrm{EL}} &\to C^\infty(\Sigma, \mathcal{G} \bowtie \mathcal{G}^*) &\qquad (195) \\
\tilde{h} &\mapsto (\tilde{H}, \tilde{L}) = (\tilde{h}, \tilde{\ell}(\tilde{h})) &\qquad (196)
\end{aligned}
$$

where $\tilde{\ell}$ is the non-local function of the on-shell field $\tilde{h}$ defined in (165). Then, the pushforward along $\Phi$ of the twisted right rotation with parameter $\alpha \in \mathfrak{g}$ is a *pointwise* dressing transformation on $C^\infty(\Sigma, \mathcal{G} \bowtie \mathcal{G}^*)$ with the same parameter, i.e.

$$
\Phi_* \mathcal{Q}(\alpha) = \delta^R_\alpha.
\tag{197}
$$

$\qquad\qquad\qquad\qquad\qquad\qquad\qquad\qquad\qquad\qquad\qquad\qquad\qquad\qquad\qquad\qquad\qquad\qquad\qquad\quad \diamond$

*Proof.* Let $f$ be a real-valued function of $(\tilde{H}, \tilde{L})$. Using the notation $\mathbb{d}f =: \langle f'_{\tilde{H}}, \triangle\tilde{H} \rangle + \langle f'_{\tilde{L}}, \triangle\tilde{L} \rangle$, one has that at $(\tilde{H}, \tilde{L}) = \Phi(\tilde{h})$:

$$
\begin{aligned}
\Phi_* \mathcal{Q}(\alpha) f|_{(\tilde{H}, \tilde{L})} = \mathcal{Q}(\alpha)(f \circ \Phi)|_{\tilde{h}} &= \langle f'_{\tilde{H}}, \mathbb{i}_{\mathcal{Q}(\alpha)} \triangle\tilde{h} \rangle + \langle f'_{\tilde{L}}, \mathbb{i}_{\mathcal{Q}(\alpha)} \triangle\tilde{\ell} \rangle \\
&\overset{?}{=} \langle f'_{\tilde{H}}, \tilde{H}^{-1} \delta^R_\alpha \tilde{H} \rangle + \langle f'_{\tilde{L}}, \tilde{L}^{-1} \delta^R_\alpha \tilde{L} \rangle = \delta^R_\alpha f|_{(\tilde{H}, \tilde{L})}
\end{aligned}
\tag{198}
$$

where the equality with the question mark is the one we need to prove.

Recall (67a), and note that the (pointwise) infinitesimal dressing action of $\mathfrak{g}$ on $(\tilde{H}, \tilde{L}) \in C^\infty(\Sigma, \mathcal{G} \bowtie \mathcal{G}^*)$ is given by $\delta^R_\alpha G(x) = G(x)\alpha$ for $G(x) = \tilde{H}(x)\tilde{L}(x)$, namely

$$
\delta^R_\alpha \left( \tilde{H}, \tilde{L} \right) = \left( \tilde{H}(\tilde{L} \triangleright \alpha), \tilde{L}\alpha - (\tilde{L} \triangleright \alpha)\tilde{L} \right).
\tag{199}
$$

Recall Definition 4.3 together with (56), and note that the (non-local) twisted right action of $\mathfrak{g}$ on $\tilde{h}$ is given by:

$$\mathbb{Q}(\alpha)\tilde{h} = \tilde{h}\,\overline{\alpha} = \tilde{h}\,\widetilde{\mathrm{Ad}}^*_{\tilde{\ell}^{-1}}\alpha = \tilde{h}\,(\tilde{\ell}\rhd\alpha). \tag{200}$$

Using the previous two equations, we readily see that $\mathring{\mathbb{i}}_{\mathbb{Q}(\alpha)}\Delta\tilde{h} = \tilde{\ell}\rhd\alpha = \tilde{L}\rhd\alpha = \tilde{H}^{-1}\delta^R_\alpha\tilde{H}$ as desired.

To show that also $\mathring{\mathbb{i}}_{\mathbb{Q}(\alpha)}\Delta\tilde{\ell} = \alpha - \tilde{\ell}^{-1}(\tilde{\ell}\rhd\alpha)\tilde{\ell} = \alpha - \tilde{L}^{-1}(\tilde{L}\rhd\alpha)\tilde{L} = \tilde{L}^{-1}\delta^R_\alpha\tilde{L}$, we use the following observation. Since $\tilde{\ell}$ is the (non-local) function of $\tilde{h}$ given by $\tilde{\ell}(\tilde{h})(x) := \mathrm{Pexp}\int_{x\leftarrow x_0}J(\tilde{h})$ (166), which is in turn defined by the conditions

$$J = \Delta\tilde{\ell} \quad and \quad \tilde{\ell}(x_0) = 1, \tag{201}$$

we observe that $\mathbb{Q}(\alpha)\tilde{\ell} = \tilde{\ell}\alpha - (\tilde{\ell}\rhd\alpha)\tilde{\ell} \equiv \delta^R_\alpha\tilde{\ell}$ as desired iff $\mathbb{Q}(\alpha)J = \delta^R_\alpha\Delta\tilde{\ell}$ and $\delta^R_\alpha\tilde{\ell}(x_0) = 0$. Moreover, from Proposition 4.5 we also know that $\mathbb{Q}(\alpha)J = \mathrm{ad}^*_\alpha J$. Therefore, to conclude the proof of the theorem, what we need to show is

$$\delta^R_\alpha(\Delta\tilde{\ell}) \stackrel{?}{=} \mathrm{ad}^*_\alpha\Delta\tilde{\ell} \quad and \quad \delta^R_\alpha\tilde{\ell}(x_0) \stackrel{?}{=} 0. \tag{202}$$

The second condition is readily proved using that $\tilde{\ell}(x_0) = 1$:

$$\delta^R_\alpha\tilde{\ell}(x_0) = \tilde{\ell}(x_0)\alpha - (\tilde{\ell}(x_0)\rhd\alpha)\tilde{\ell}(x_0) = \alpha - \alpha = 0. \tag{203}$$

To prove the first condition, we compute instead:

$$\begin{aligned}
\delta^R_\alpha(\Delta\tilde{\ell}) &= \tilde{\ell}d(\tilde{\ell}^{-1}\delta^R_\alpha\tilde{\ell})\tilde{\ell}^{-1} = \tilde{\ell}d(\alpha - \tilde{\ell}^{-1}(\tilde{\ell}\rhd\alpha)\tilde{\ell})\tilde{\ell}^{-1} \\
&= 0 + [\Delta\tilde{\ell}, \tilde{\ell}\rhd\alpha]_\mathfrak{d} - (\Delta\tilde{\ell})\rhd(\tilde{\ell}\rhd\alpha) = [\Delta\tilde{\ell}, \overline{\alpha}]_\mathfrak{d} - \Delta\tilde{\ell}\rhd\overline{\alpha} = (\Delta\tilde{\ell})\lhd\overline{\alpha} = \mathrm{ad}^*_{\overline{\alpha}}\Delta\tilde{\ell},
\end{aligned} \tag{204}$$

where in the second line we used the definition of the Drinfel'd double algebra, that is $[X,\beta]_\mathfrak{d} = X\lhd\beta + X\rhd\beta = X\lhd\beta - \beta\lhd X = \mathrm{ad}^*_\beta X - \widetilde{\mathrm{ad}}^*_X\beta$ (cf. (56)). $\qquad\square$

**Theorem 4.8** (PL symmetry of the second order KS model)**.** Twisted right rotations $\mathbb{Q}$ are a Poisson-Lie symmetry (Definition 2.5) of the second order KS model (130) for the open string $\Sigma \simeq [0,\pi]\times\mathbb{R}$ with Neumann boundary conditions (147). In particular, the PL flow equation[39] (36),

$$\mathring{\mathbb{i}}_{\mathbb{Q}(\alpha)}\Omega \approx -\langle Q^{-1}\mathrm{d}Q, \alpha\rangle, \tag{205}$$

holds for the *conserved* PL charge

$$Q \approx \mathrm{Pexp}\int_{\pi\leftarrow 0}J \approx \tilde{\ell}|_{\{\pi\}\times\mathbb{R}}, \tag{206}$$

where the integral is performed along $C \simeq [0,\pi]\subset\Sigma$.

Contact with equation (29) is obtained by choosing the constant reference solution $\varphi_0 \equiv \tilde{h}_0 = 1$, which is such that $Q(\varphi_0) = 1$ for all choices of Cauchy surface. $\qquad\diamond$

---

[39]Up to an inconsequential sign.

<sub>1325</sub> *Proof.* In Proposition 4.6 we have already shown that $\mathbb{Q}$ preserves the equations of motion
<sub>1326</sub> and, for the open string, the Neumann boundary conditions. Therefore $\mathbb{Q}$ is a vector field
<sub>1327</sub> on $\mathcal{F}_{\text{EL}}$, satisfying point 1' of Definition 2.5.

<sub>1328</sub> We also have a Drinfel'd double $\mathfrak{d} = \mathfrak{g} \bowtie \mathfrak{g}^*$ by construction. We now need to prove the
<sub>1329</sub> PL flow equation (205) for the charge (206) through a CPS analysis.

<sub>1330</sub> Recall the pSF current $\omega^\mu$ (146), the definition of the twisted right rotation of equation
<sub>1331</sub> (171), and the corresponding transformation property of $j^\mu$ (Proposition 4.5). With these,
<sub>1332</sub> we compute the variational current

$$\langle \mathbb{J}^\mu, \alpha \rangle :\approx \mathbb{i}_{\mathbb{Q}(\alpha)}\omega^\mu \approx \langle \text{ad}^*_{\overline{\alpha}} j^\mu, \Delta\tilde{h} \rangle - \langle \mathbb{d}j^\mu, \overline{\alpha} \rangle - \langle j^\mu, \left[ \overline{\alpha}, \Delta\tilde{h} \right] \rangle \approx -\langle \mathbb{d}j^\mu, \overline{\alpha} \rangle \approx -\langle \mathbb{d}(\nabla_\nu \tilde{\ell}\,\tilde{\ell}^{-1}\epsilon^{\nu\mu}), \overline{\alpha} \rangle, \tag{207}$$

<sub>1333</sub> where in the third equality we used the invariance property of the pairing to cancel the
<sub>1334</sub> first and last terms, and in the last equality we used the on-shell expression (165) of $j^\mu$ in
<sub>1335</sub> terms of $\tilde{\ell}$.

<sub>1336</sub> To define the main object of interest, the variational charge $\mathbb{Q}$, we integrate the pSP current
<sub>1337</sub> over a Cauchy hypersurface $C \simeq [0, \pi]$ with unit surface conormal $n_\mu$,

$$\Omega = \int_C n_\mu \omega^\mu, \tag{208}$$

<sub>1338</sub> Combining this and the previous equation, we compute the variational charge to be,

$$\begin{aligned}
\mathbb{i}_{\mathbb{Q}(\alpha)}\Omega \approx \langle \mathbb{Q}, \alpha \rangle &\approx \int_C n_\mu \langle \mathbb{J}^\mu, \alpha \rangle \approx -\int_C \langle n_\mu \mathbb{d}j^\mu, \overline{\alpha} \rangle \\
&\approx -\int_C n_\mu \langle \mathbb{d}(\nabla_\nu \tilde{\ell}\,\tilde{\ell}^{-1}\epsilon^{\nu\mu}), \overline{\alpha} \rangle \approx -\int_C \langle \tilde{\ell}d(\tilde{\ell}^{-1}\mathbb{d}\tilde{\ell})\tilde{\ell}^{-1}, \overline{\alpha} \rangle \\
&= -\int_C \langle \tilde{\ell}d(\tilde{\ell}^{-1}\mathbb{d}\tilde{\ell})\tilde{\ell}^{-1}, \widetilde{\text{Ad}}^*_{\tilde{\ell}^{-1}}\alpha \rangle = -\int_C d\langle \tilde{\ell}^{-1}\mathbb{d}\tilde{\ell}, \alpha \rangle \\
&= -\int_{\partial C} \langle \tilde{\ell}^{-1}\mathbb{d}\tilde{\ell}, \alpha \rangle = -\langle \tilde{\ell}^{-1}(\tau, \pi)\mathbb{d}\tilde{\ell}(\tau, \pi), \alpha \rangle
\end{aligned} \tag{209}$$

<sub>1339</sub> where $\tau \in \mathbb{R}$ in this equation is arbitrary but fixed (it denotes the specific Cauchy slice
<sub>1340</sub> in the foliation $\Sigma \simeq C \times \mathbb{R}$, $C \simeq [0, \pi]$). In the last step, we used that according to our
<sub>1341</sub> definition, and the Neumann boundary condition on $\{0\} \times \mathbb{R} \subset \partial\Sigma$, $\tilde{\ell}(\tau, 0) = 1$.

<sub>1342</sub> Note that the Neumann boundary condition on $\{\pi\} \times \mathbb{R} \subset \partial\Sigma$ guarantees the constancy
<sub>1343</sub> in $\tau$ of $\tilde{\ell}(\pi, \tau)$, namely the conservation of $Q$ [34] (cf. (167)).

<sub>1344</sub> Therefore, not only is $\mathbb{Q}$ generated by the group-valued charge $Q$ (point 2' of Definition
<sub>1345</sub> 2.5), but the charge

$$Q \approx \text{Pexp} \int_{\pi \leftarrow 0} J(\tilde{h}) \approx \tilde{\ell}(\tilde{h})|_{\{\pi\} \times \mathbb{R}} \tag{210}$$

<sub>1346</sub> can be expressed as a (non-local) function of $\tilde{h}$ and its derivatives at (any) surface
<sub>1347</sub> $C \simeq [0, \pi] \hookrightarrow \Sigma$ (point 3' of Definition 2.5). This results relies on the Neumann boundary
<sub>1348</sub> conditions and the expression (145) of the current $J_\nu = j^\mu \epsilon_{\mu\nu}$.

<sub>1349</sub> We conclude that $(\mathcal{F}_{\text{EL}}, \Omega, \mathbb{Q}, Q)$ defines a Poisson-Lie symmetry in the covariant framework
<sub>1350</sub> of Definition 2.5. $\qquad\square$

<sub>1351</sub> Conservation of $Q$ from that of $\mathbb{Q}$ was discussed in general terms in §2.3, where $Q$ was
<sub>1352</sub> defined as a path-ordered exponential of $\mathbb{Q}$ over a path in field space from a reference
<sub>1353</sub> configuration $\varphi_0$ (29). Here, that construction follows from equation (148) and the choice

of the reference configuration $\varphi_0 \in \mathcal{F}_{\text{EL}}$ to be $\tilde{\ell} = 1$ throughout $\Sigma$, from which one can easily verify that $J(\varphi_0) = 0$ and thus $Q(\varphi_0) = 1$, as desired.

As twisted right rotations generalize the right rotation symmetry of the principal chiral model (151), the non-Abelian PL charge generalizes the principal chiral model's one, wherein $\mathfrak{g}^*$ is Abelian and the corresponding Noetherian charge was found to be $q = \int_C n_\mu j^\mu = \int_C J$ (158), viz:

$$Q \approx \text{Pexp} \int_C J \xrightarrow{\mathfrak{g}^* \text{ Abelian}} \exp \int_C J = \exp(q). \tag{211}$$

We recall that in the Abelian limit of $\mathcal{G}^*$, $q$ is the total momentum (charge) generating the "rotational" symmetry of the target space; if the target space is also Abelian, $\mathcal{G} \simeq (\mathbb{R}^n, +)$, then target space "rotations" are nothing else than global translations, and $Q$ is nothing else than the exponential of the total linear momentum of the string moving in $\mathbb{R}^n$.

As we have previously emphasized in §2.3, it is important to realize that, contrary to Noetherian symmetries, PL symmetries do not leave the Lagrangian invariant. In fact, in the present case the situation is even more serious, since the twisted right rotations PL symmetry is not even defined off-shell and, since on-shell all variations of the Lagrangian are pure-boundary, the Noetherian criterion is void for twisted right rotations.

Concerning the question of locality, we stress that a twisted right rotation is a *non-local* action, providing an (infinitesimal) non-local redefinition of the field $\tilde{h}$, namely $\tilde{h}^{-1} \delta_\alpha \tilde{h} = \widetilde{\text{Ad}}^*_{\tilde{\ell}^{-1}} \alpha$ for $\tilde{\ell} \approx \text{Pexp} \int J$ a *non-local* function of $\tilde{h}$.

As discussed in §2.3 below (34), the need for some non-locality can be gleaned already from the analog of equation (33) as computed from (205) of Theorem 4.8, i.e.[40]

$$\mathbb{L}_{\mathbb{Q}(\alpha)} \Omega = \mathbb{d} \langle \mathbb{Q}, \alpha \rangle \approx \frac{1}{2} \langle [\mathbb{Q}, \mathbb{Q}]_*, \alpha \rangle. \tag{212}$$

Indeed, whereas $\Omega$ on the left hand side is by construction a local functional of the fields, the commutator on the right hand side, a priori involves a bi-local expression (if the number of spatial dimension is strictly larger than zero, each $\mathbb{Q}$ is a priori an integral over $C$).

Beside the case where $\mathfrak{g}^*$ is Abelian so that the commutator on the right hand side simply vanishes, we are aware of two possible solutions to this tension. In the first scenario, the action $\mathbb{Q}$ is itself non-local; this is what happens for the twisted right rotations of the second order KS model discussed here.[41] In the second scenario, the integral defining $\mathbb{Q}$ localizes at a point of the Cauchy surface/string; this is what happens in the first order KS model (see §4.5). In this case, although everything appears local[42] (the action of the symmetry, the commutator in (212)), it is the physical interpretation of the charge $Q$ that is in tension with locality, since then "*total* momentum" of the string with respect to the given (global!) symmetry is captured by the value of the worldsheet fields at a *single* point of the string. In the first order KS model this happens because, in passing from the second- to the first order formulation, one employs a non-local field redefinition. See §4.5.

In the following sections we briefly discuss two variations of the above results. One involves a closed string version of the model (either in second or first order), and the other a first

---

[40]As observed in the theorem, there is an inconsequential sign different between the PL flow equation in the KS model and the one discussed in section §2.3.

[41]In the second order model, $Q = \tilde{\ell}(\pi) = \text{Pexp} \int_C J$ where $J$ is a local function of the field $\tilde{h}$.

[42]Essentially because now $\tilde{\ell}$ is treated as an independent "conjugate momentum" field with respect to the "configuration field" $\tilde{h}$.

order Lagrangian for the open string KS model. The closed string reveals the difficulties of implementing non-Abelian Poisson-Lie symmetries in the absence of boundaries. The first order formulation, on the other hand, is the one mostly used by Klimčík and Ševera in their works for it enjoys a non-Abelian T-duality.

## 4.4 Some Comments on the Klimčík & Ševera Closed String

To close the KS string as it was formulated in §4.1, we need only replace the open string Neumann boundary conditions (135) (equivalently (147)) with the appropriate closed string periodic boundary condition:

$$\tilde{h}(0,\tau) = \tilde{h}(\pi,\tau).$$ (213)

All of our CPS symmetry analysis from the previous sections still holds, up to the point of checking the compatibility of our twisted right rotations (Definition 4.3) with the closed string boundary condition given above.

Recall from §4.3.2 that $\overline{\alpha}(x) :\approx \widetilde{\mathrm{Ad}}^*_{\tilde{\ell}^{-1}(x)}\alpha$ where $\tilde{\ell}$ is the non-local function of the on-shell field $\tilde{h}$ defined in (165). It can be easily checked that imposing (213) leads to the following additional condition on the symmetry parameter $\alpha$:

$$\overline{\alpha}(\pi) = \overline{\alpha}(0) \quad \text{i.e.} \quad \widetilde{\mathrm{Ad}}^*_{\tilde{\ell}^{-1}(\pi)}\alpha = \widetilde{\mathrm{Ad}}^*_{\tilde{\ell}^{-1}(0)}\alpha \equiv \alpha.$$ (214)

Namely, $\alpha$ needs to be stabilized by $\tilde{\ell}(\pi) \in \mathcal{G}^*$ (recall that by construction $\tilde{\ell}(0) = 1$).

For this to be a global symmetry in the usual sense, this must happen for all symmetry generators $\alpha$ and all $\tilde{\ell}(\pi)$ in the phase space. This condition is trivially satisfied whenever $\mathcal{G}^*$ is Abelian, since then $\widetilde{\mathrm{Ad}} = \widetilde{\mathrm{Ad}}^* \equiv 1$, which is the usual Noetherian scenario of a "trivial" PL symmetry. If $\mathcal{G}^*$ is non-Abelian, on the other hand, the above condition generally fails and one is left with the choice of whether to restrict the set of symmetries (constrain $\alpha$) or the set of allowed configurations (constrain $\tilde{\ell}(\pi)$).

For $\mathfrak{g}^*$ semisimple, however, there is no $\alpha$ that satisfies the above condition for all $\tilde{\ell} \in \mathcal{G}^*$. Indeed, consider the same condition for $\tilde{\ell} = 1 + X + ...$, then contracting with an arbitrary $Y \in \mathfrak{g}^*$ we obtain

$$\langle \alpha, [X,Y]_* \rangle = 0, \quad \forall X, Y \in \mathfrak{g}^*,$$ (215)

which means that $\alpha$ annihilates the derived subalgebra $[\mathfrak{g}^*, \mathfrak{g}^*]_* \subset \mathfrak{g}^*$. But, for $\mathfrak{g}^*$ semisimple $[\mathfrak{g}^*, \mathfrak{g}^*]_* = \mathfrak{g}^*$, and thus $\alpha$ must be zero since the canonical pairing is non-degenerate.

Alternatively, one can constrain $\tilde{\ell}(\pi)$ to be in the center of $\mathcal{G}^*$, whence $\widetilde{\mathrm{Ad}}_{\tilde{\ell}^{-1}(\pi)} \equiv 1$. Assuming $\mathcal{G}^*$ semisimple, $\tilde{\ell}(\pi)$ is therefore constrained to take a discrete set of values. Since $\tilde{\ell}(\pi)$ is also the momentum map for the open string, imposing this constraint while keeping a non-degenerate sympelctic structure over phase space requires a symplectic reduction procedure. In particular, the demand that $\tilde{\ell}(\pi) = 1$ trivializes the charge and closes the string in momentum space (see the previous section). This way, one obtains, as proposed by Klimčík and Ševera [33, Section 2], a closed string with $\mathcal{D}$ as a target space where $\mathcal{G}$ and $\mathcal{G}^*$ play symmetric roles. The "meta-symmetry" of interchanging $\mathcal{G}$ and $\mathcal{G}^*$ is called (Poisson-Lie) T-duality [34, 61, 62], see §4.5.

A different perspective on $\tilde{\ell}(\pi)$ being in the center of $\mathcal{G}^*$ can be obtained by thinking about what it would take to have conservation of a (*group!*) $\mathcal{G}^*$-valued charge $Q$ on the worldsheet of a closed string, $\Sigma \simeq S^1 \times \mathbb{R}$, if this charge is given by the (path-ordered)

integral of an (on-shell) flat current $J$ on $S^1$ (206). In this scenario, it is clear that the value of $Q$ depends in general on the choice of a base point for the path-ordered integral, with different choices yielding charges conjugate to each other. Then, conservation can at best be expected up to conjugation, namely $Q_{\text{fin}} \approx L Q_{\text{in}} L^{-1}$ for some[43] $L \in \mathcal{G}^*$—*unless* $Q = L Q L^{-1}$ for arbitrary $L \in \mathcal{G}^*$, which is precisely the condition that $Q$ is valued in the center of $\mathcal{G}^*$.

In sum, this means that in general a group-valued charge $Q$ for the closed-string can be expected to be conserved only if $\mathcal{G}^*$ is Abelian or if it is somehow constrained to be valued in its center. It is thus relevant to recall that Poisson-Lie symmetries for an Abelian dual group $\mathcal{G}^*$ fall into the Noetherian formulation. We conclude that the closed string KS model can possess no Poisson-Lie symmetries with non-trivial charge.

## 4.5 Some Comments on the First Order Formulation of the Klimčík & Ševera Model

To express the string KS model in the first order formalism one first takes a canonical 1+1 split of the worldsheet $\Sigma$, wherein $\Sigma = \mathbb{R} \times C \ni (\tau, \sigma)$, with fixed choice of foliation specified by the Cauchy surface $C \simeq [0, \pi]$.

Doing the Legendre transform of the KS Lagrangian (130), a first order action can then be written in terms of the configuration variable $\tilde{h}$ and canonical momentum $\tilde{p} := j^\tau(\tilde{h})$ valued in $\mathfrak{g}^*$ (cf. (144) and (145)). However, as Klimčík and Ševera comment [the words are theirs, the notation is adapted to ours]: "Our crucial trick is the following: we parametrize the canonically conjugated momentum $\tilde{p}$ by a field $\tilde{\ell}$ (valued in $\mathcal{G}^*$) such that $\tilde{p} = \partial_\sigma \tilde{\ell} \tilde{\ell}^{-1}$. The ambiguity of this parametrization is fixed by requiring that $\tilde{\ell}(\tau, \sigma = 0) = 1$" [34]. In other words, they observe that in 1-dimension (we are on the Cauchy surface $C \simeq [0, \pi]$), the $\mathfrak{g}^*$-valued field $\tilde{p}$ can always[44] be replaced by a $\mathcal{G}^*$-valued field $\tilde{\ell}(\sigma) = \text{Pexp} \int_{\sigma \leftarrow 0} \tilde{p}$, so that $\tilde{p} = \partial_\sigma \tilde{\ell} \tilde{\ell}^{-1}$. The re-parametrization $\tilde{p} \mapsto \tilde{\ell}$ involves an integration and is therefore *non-local* on $C$.

Therefore, following Klimčík and Ševera in passing to a first order formulation, instead of $(\tilde{h}, \tilde{p})$, we introduce the pair of fields $(\tilde{h}, \tilde{\ell}) : \Sigma \to \mathcal{G} \bowtie \mathcal{G}^* \simeq \mathcal{D}$. The first order version of the KS Lagrangian (130) then takes the form [34]

$$\mathcal{L}_{1st} = \langle \partial_\sigma \tilde{\ell} \tilde{\ell}^{-1}, \tilde{h}^{-1} \partial_\tau \tilde{h} \rangle - \mathcal{H}(\partial_\sigma G G^{-1}), \tag{216}$$

where, as a result of Theorem 4.7, $G = \tilde{h} \tilde{\ell} \in \mathcal{D}$, and an explicit form of the "$E$-model" Hamiltonian $\mathcal{H}(\partial_\sigma G G^{-1})$ can be found in [34, 63, 64].

Omitting the details, and using the notations

$$\text{d}\mathcal{H} =: \langle \mathcal{H}'_{\tilde{h}}, \triangle \tilde{h} \rangle + \langle \mathcal{H}'_{\tilde{\ell}}, \triangle \tilde{\ell} \rangle \quad \text{and} \quad \mathcal{E} = \langle \mathcal{E}_{\tilde{h}}, \triangle \tilde{h} \rangle + \langle \mathcal{E}_{\tilde{\ell}}, \triangle \tilde{\ell} \rangle, \tag{217}$$

the CPS data for this Lagrangian are:

$$\mathcal{E}_{\tilde{h}} = - \left( \partial_\tau + \text{ad}^*_{\triangle_\sigma \tilde{h}^{-1}} \right) \triangle_\sigma \tilde{\ell} - \mathcal{H}'_{\tilde{h}} \tag{218}$$

$$\mathcal{E}_{\tilde{\ell}} = -\widetilde{\text{Ad}}^*_{\tilde{\ell}} \left( \left( \partial_\sigma + \widetilde{\text{ad}}^*_{\triangle_\sigma \tilde{\ell}} \right) \triangle_\tau \tilde{h}^{-1} \right) - \mathcal{H}'_{\tilde{\ell}} \tag{219}$$

$$\Theta = \int_0^\pi d\sigma \, \langle \triangle_\sigma \tilde{\ell}, \triangle \tilde{h} \rangle \tag{220}$$

---

[43]The element $L \in \mathcal{G}^*$ is nothing else than the path ordered integral of $J$ along a path connecting the base points on $\Sigma_{\text{in}}$ and $\Sigma_{\text{fin}}$.

[44]Note: the field redefinition from $\tilde{p} \mapsto \tilde{\ell}$ does not require using the equation of motion for $\tilde{p}$, i.e. the flatness of, $j^\mu$.

from which

$$\Omega := \mathbb{d}\Theta = \int_0^\pi d\sigma \ \langle \widetilde{\mathrm{Ad}}_{\tilde{\ell}} \partial_\sigma \mathbb{\Delta}\tilde{\ell}, \mathbb{\Delta}\tilde{h} \rangle - \frac{1}{2} \langle \Delta_\sigma \tilde{\ell}, [\mathbb{\Delta}\tilde{h}, \mathbb{\Delta}\tilde{h}] \rangle. \tag{221}$$

As proved in Theorem 4.7, the *non-local* action of twisted right rotations on the second order Lagrangian is mapped in the first order formalism to the dressing action (67a) of the Drinfel'd double studied by Klimčík and Ševera [34]:

$$G \mapsto Gh_0 \implies \delta_\alpha^R G = G\alpha \implies \delta_\alpha^R(\tilde{h}, \tilde{\ell}) = \left( \tilde{h}(\tilde{\ell} \rhd \alpha), \tilde{\ell}\alpha - (\tilde{\ell} \rhd \alpha)\tilde{\ell} \right). \tag{222}$$

We have thus shown that our analysis of the second order KS model and its symmetries matches their original analysis of the first order model.

Indeed, using (221) and (222), it is straightforward to show that (cf. the proof of Theorem 4.8)

$$\mathbb{i}_{\mathbb{Q}(\alpha)} \Omega = -\langle \mathbb{\Delta} Q, \alpha \rangle \quad \text{with} \quad Q = \tilde{\ell}(\pi). \tag{223}$$

At this point it is worth noting the following facts. The Lagrangian $\mathcal{L}_{1\text{st}}$ is invariant up-to-boundary terms under the transformation (222), with remainder

$$\delta_\alpha^R \mathcal{L}_{1\text{st}} = \nabla_\mu \langle R^\mu, \alpha \rangle \quad \text{where} \quad R^\mu = -\epsilon^{\mu\nu} \tilde{\ell}^{-1} \partial_\nu \tilde{\ell}, \tag{224}$$

whence one concludes that the dressing symmetries (222) are Noetherian. Moreover, computing the corresponding Noetherian charge one obtains

$$\langle q, \alpha \rangle = \mathbb{i}_{\mathbb{Q}(\alpha)} \Theta - \int_0^\pi d\sigma \langle R^\tau, \alpha \rangle = 0, \tag{225}$$

These two facts are puzzling at first sight. Namely, the dressing transformations (222) being Noetherian (224) is seemingly in contradiction with the fact (223) that they are a PL symmetry with group-valued momentum map (cf. the discussions in §2.3), as well, the corresponding Noetherian charge (225) identically vanishing is per se puzzling for a global symmetry. Of course, the two puzzles and their resolutions are related.

First, the theorem stating that to any (global) Noetherian $\mathcal{G}$-symmetry one assigns a conserved $\mathfrak{g}^*$-valued charge that generates the corresponding symmetry flow in phase space holds on *closed* Cauchy surfaces—and not necessarily on open ones (cf. postulates 1–3 in §2.3). Therefore, in the open string case, there is no contradiction between the last three formulas, since the relation between $\mathbb{Q} := \mathbb{i}_{\mathbb{Q}(\alpha)} \Omega$ and $\mathbb{d}q$ holds only *up to corner terms*, consistently with $q$ being identically zero and $Q$ being supported on $\partial C$.

Second, since $\tilde{\ell}(0) = 1$, closing the first order string requires not only that $\tilde{h}(0) = \tilde{h}(\pi)$ as discussed in the previous section, but also that $Q = \tilde{\ell}(\pi) = 1$. Therefore, in the closed case $\mathbb{Q} = 0$ both from (223) and from (225) computed from Noether's theorem.

In other words, whereas the open first order KS string possesses a non-trivial PL symmetry and charge, the closed first order string is characterized by a "gauged" global symmetry with trivial charge. Namely, due to the closed string constraint $Q = \tilde{\ell}(\pi) = 1$, to obtain a *symplectic* phase space, one has to perform symplectic reduction of the right dressing action by $\mathcal{G}$. This yields a string model whose phase space is the based loop group $\Omega \mathcal{D} \simeq L\mathcal{D}/\mathcal{D}$ [33, Section 2].

Note that, in light of this observation, the closed first order KS string provides an important example of a charge constrained to be valued in the center of $\mathcal{G}^*$ when the Cauchy surface is closed—as per the discussion of §4.4.

As discovered by Klimčík and Ševera [34, 61], one of the most interesting features of the KS closed string model is that it features a *non-Abelian T-duality*, meaning that the model is invariant under the interchange of $\mathcal{G}$ and $\mathcal{G}^*$. This feature is highly non-trivial in the second order formulation, but can be proved rather easily in the first order one. This is because in the first order formulation, the non-Abelian T-duality takes the form of a dynamical[45] symmetry that interchanges configurations and momenta.

In the case of a point-particle such a symmetry takes the form $(q, p) \mapsto (\tilde{q}, \tilde{p}) = (-p, q)$. This symplectomorphism is a dynamical symmetry of the theory if and only if $H(q, p) = H(\tilde{p}, -\tilde{q})$. At the Lagrangian level, this transformation is seen to be a symmetry because it leaves the Lagrangian invariant up to a boundary term coming from the transformation of the Liouville term $p\dot{q}$:

$$L_{1\text{st}}(q, p) := p\dot{q} - H(q, p) = L_{1\text{st}}(\tilde{p}, -\tilde{q}) - \frac{d}{dt}(\tilde{p}\tilde{q}). \tag{226}$$

For a Lagrangian quadratic in the momenta, this (dynamical) duality is only realized by the harmonic oscillator. Indeed, $H(p, q) = \frac{1}{2m}(p^2 + q^2) + Apq$ is bounded from below iff $1 - m^2 A^2 \geq 0$, in which case the corresponding Hamilton equations reproduce those for a harmonic oscillator of angular frequency $\sqrt{1 - m^2 A^2}/m$.

To see that the KS model enjoys a form of non-Abelian T-duality, it is enough to provide a transformation that exchanges $\mathcal{G}$ and $\mathcal{G}^*$ while leaving the Hamiltonian and symplectic structure invariant. This is of course given by the isomorphism

$$\mathcal{G}^* \bowtie \mathcal{G} \simeq \mathcal{G} \bowtie \mathcal{G}^*, \quad (\tilde{\ell}, \tilde{h}) \mapsto (h, \ell) = (\tilde{\ell} \triangleright \tilde{h}, \tilde{\ell} \triangleleft \tilde{h}). \tag{227}$$

Since $G = \tilde{h}\tilde{\ell} = \ell h$ is invariant under the above isomorphism, the KS Hamiltonian is manifestly invariant. To prove the well-known fact that the *closed*-string symplectic structure is similarly invariant, we make use here of the following property of the KS Liouville term [34, Eq. (31)]:[46]

$$\langle \partial_\sigma \tilde{\ell}\tilde{\ell}^{-1}, \tilde{h}^{-1}\partial_\tau \tilde{h} \rangle \, d\sigma \wedge d\tau = \langle \partial_\sigma h h^{-1}, \ell^{-1}\partial_\tau \ell \rangle \, d\sigma \wedge d\tau - \gamma^* \omega_{\text{STS}}(\ell, h), \tag{228}$$

where $\omega_{\text{STS}}$ is the STS symplectic form on the Heisenberg double $\mathcal{D}_{\text{H}}$ (64) and $\gamma : \Sigma \mapsto \mathcal{D}_{\text{H}}$ is the history $x \mapsto G(x) := \tilde{h}(x)\tilde{\ell}(x) = \ell(x)h(x)$.

This equality, together with the invariance of the KS Hamiltonian, implies that

$$\mathcal{L}_{1\text{st}}(\tilde{h}, \tilde{\ell}) = \mathcal{L}_{1\text{st}}(\ell, h) - \gamma^* \omega_{\text{STS}}(\ell, h). \tag{229}$$

Since the STS symplectic form is not $d$-exact, the last term $\gamma^* \omega_{\text{STS}}$ is not boundary as in the particle mechanics case—it is, however, $d$-closed. We studied the contribution of

---

[45]Meaning that not only the kinematics, i.e. symplectic structure, but also the dynamics of the theory are preserved under the symmetry.

[46]This identity is proven by repeatedly using the ribbon identity $\tilde{h}\tilde{\ell} = \ell h$ and the isotropy of the symmetric pairing $\langle \cdot, \cdot \rangle$ in $\mathfrak{d}$, namely the fact that $\langle \mathfrak{g}, \mathfrak{g} \rangle = 0 = \langle \mathfrak{g}^*, \mathfrak{g}^* \rangle$. In particular, these facts allow one to show that:

$$\omega_{\text{STS}} := \langle \tilde{\ell}^{-1}d\tilde{\ell}, h^{-1}dh \rangle + \langle d\ell\ell^{-1}, d\tilde{h}\tilde{h}^{-1} \rangle = \left( -\langle \tilde{\ell}^{-1}\partial_\tau \tilde{\ell}, h^{-1}\partial_\sigma h \rangle + \langle \partial_\sigma \ell\ell^{-1}, \partial_\tau \tilde{h}\tilde{h}^{-1} \rangle \right) d\sigma \wedge d\tau,$$

and that

$$\langle \partial_\sigma \tilde{\ell}\tilde{\ell}^{-1}, \tilde{h}^{-1}\partial_\tau \tilde{h} \rangle - \langle \partial_\sigma h h^{-1}, \ell^{-1}\partial_\tau \ell \rangle = \langle \partial_\sigma \ell\ell^{-1}, \partial_\tau \tilde{h}\tilde{h}^{-1} \rangle - \langle \tilde{\ell}^{-1}\partial_\tau \tilde{\ell}, h^{-1}\partial_\sigma h \rangle,$$

which together prove the asserted formula. In the main text, $\omega_{\text{STS}}(\ell, h)$ is given by the formal expression above with $(\tilde{h}, \tilde{\ell}) = (\ell \triangleright h, \ell \triangleleft h)$ understood as functions of $(h, \ell)$. With this understanding, $\omega_{\text{STS}}(\ell, h) = -\omega_{\text{STS}}(\tilde{h}, \tilde{\ell})$.

this term to the action in §3.2 where it was denoted $\Psi_0(\gamma) := \int_\Sigma \gamma^* \omega_{\text{STS}}$. There, we found that its variation is such that it will not contribute to the equations of motion nor to the symplectic structure of the *closed* KS string (cf. Lemma 3.1, Proposition 3.2, and Equation (110)). In other words, since the difference between $\mathcal{L}_{1\text{st}}(\tilde{h}, \tilde{\ell})$ and $\mathcal{L}_{1\text{st}}(\ell, h)$ is a "topological string action" (A-model) which bares no consequence on the theory of the closed KS string, the two Lagrangians are dynamically equivalent, hence the first order closed string KS model enjoys a non-Abelian T-duality.

We close this section with the curious remark that since $\omega_{\text{STS}}(\tilde{\ell}, \tilde{h}) = -\omega_{\text{STS}}(h, \ell)$, as can be seen from (228), one can introduce a self-T-dual Lagrangian for the closed first order KS string:

$$\mathcal{L}_{\text{self}}(\tilde{h}, \tilde{\ell}) := \mathcal{L}_{1\text{st}}(\tilde{h}, \tilde{\ell}) - \frac{1}{2} \gamma^* \omega_{\text{STS}}(\tilde{h}, \tilde{\ell}). \tag{230}$$

In the open string case [62], one must augment the KS action by an appropriate boundary term $\partial_\sigma \mathcal{K}(\tilde{h}, \tilde{\ell})$ to have the desired boundary conditions fixed by means of the equations of motion. If one's starting point is the self-T-dual Lagrangian $\mathcal{L}_{\text{self}}$, then one ends up with an action consisting of the "naive" open KS string coupled at its endpoints to a pair of deformed spinning tops with Hamiltonian $2\mathcal{K}$, see §3.2:

$$\mathcal{L}_{\text{self}} + \partial_\sigma \mathcal{K} = \mathcal{L}_{1\text{st}} - \frac{1}{2} \left( \gamma^* \omega_{\text{STS}} - 2 \partial_\sigma \mathcal{K} \right) = \mathcal{L}_{1\text{st}} - \frac{1}{2} \mathcal{L}_{\text{def.sp.tops}}. \tag{231}$$

Upon dualization, this coupled system yields by the above formulas another open KS string coupled to another pair of deformed spinning tops at its endpoints, this time equipped with different dynamics implementing "dual" boundary conditions (we don't expect $\mathcal{K}$ to be self-dual).

# 5 Poisson-Lie Symmetry for a 2+1D Field Theory

Before we begin our illustration of a 2+1D field theory with a non-trivial Poisson-Lie symmetry, we find it important to summarize and reflect on some of the lessons learned thus far.

To build examples of field theories with Poisson-Lie symmetries, it is convenient to think of them as deformations of field theories with standard, namely Noetherian, symmetries. In such theories the charges are built as integrals of conserved currents over a Cauchy surface. This is possible because these currents are valued in a linear space (the dual of the Lie algebra), as well as the charges and momentum maps. In theories with (non-trivial) PL symmetries, however, the momentum map is valued in a non-Abelian group, $\mathcal{G}^*$, which is not a linear space, whence the question arises of how to build such an object from a current.

In 0+1D, currents are simply functions of time, and conserved currents are constant functions of time, which can be readily identified with the associated conserved charge. A non-trivial PL symmetry yields in this case a $\mathcal{G}^*$-valued conserved charge $Q$, with $\mathcal{G}^*$ non-Abelian. In 1+1D, Hodge duality relates 1-forms and 1-vectors over spacetime, which allows one to identify currents with 1-forms and therefore "deform" the usual conservation equation $dJ \approx 0$ into a flatness equation $dJ + \frac{1}{2}[J, J]_* \approx 0$, which involves spacetime 2-forms, once $\mathfrak{g}^*$ is given a (non-Abelian) Lie algebra structure. This deformed conservation equation is at the root of the PL symmetries in 1+1D, yielding (with appropriate

boundary conditions) conserved $\mathcal{G}^*$-valued charges obtained as path-ordered integrals of $J$ on the (bounded) Cauchy slice, $Q = \text{Pexp} \int J.$[47]

In 2+1D, it is unclear how to canonically obtain a non-Abelian $\mathcal{G}^*$-valued conserved charge by appropriately integrating local quantities on the (bounded) Cauchy surface. Something like a surface-ordered integral seems necessary, and the theory of 2-connections and 2-gauge symmetries comes straight to mind [65]. Here, we will not pursue this generalization, but rather work with a 2+1D topological field theory ($BF$ theory/Chern-Simons theory) over a "filled cylinder" $M = D^2 \times \mathbb{R}$, that "holographically projects" to the boundary "worldsheet" $\Sigma \simeq \partial M = S^1 \times \mathbb{R}$ and thus reducing, de facto, to a 1+1D theory.

In this setup, boundary PL symmetries emerge as global symmetries of the would-be-gauge boundary degrees of freedom of the topological bulk *gauge* theory. In other words, they correspond to reducibility parameters of the boundary gauge symmetries (which are, here, treated as broken and therefore as encoding physical degrees of freedom). This correspondence with reducibility parameters readily implies that the associated charges must vanish – as is also expected for a closed string (if $\Sigma = \partial M$, then $\partial \Sigma = \emptyset$).

The trick to recover non-trivial PL symmetries with non-vanishing charges is to introduce a cellular decomposition of $\Sigma$ and therefore of $\partial \Sigma$. This is natural in a "lattice" approach to $BF$ theory, where one considers the theory on a cellular complex and associates data, a priori, to all its $k$-cells. This way, a discretization of $D^2$ (or $S^2$) into e.g. triangles, leaves us to consider data associated not only to $\partial D^2 \simeq S^1$ but also to the sides of the triangles, that is, open strings $C \simeq [0, \pi]$, as well as its vertices, $\partial C \simeq \{0\} \cup \{\pi\}$. From the viewpoint of 2+1D $BF$ theory, the PL charges are associated to codimension-3 objects, in the same way that the PL charges of the 1+1D Klimčík-Ševera model turned out to be associated to codimension-2 objects (Theorem 4.8).

This idea to recover the Klimčík-Ševera model of a closed string from a "bulk" Chern-Simons theory was expounded in great detail by Ševera in [36]. Similar ideas were independently developed in the context of 2+1D quantum gravity by one of the authors and his collaborators – see in particular [35]. Here we follow the latter exposition, where 2+1D (Euclidean, first order Palatini-Cartan) quantum gravity with a negative cosmological constant is described in terms of a deformed $SU(2)$-$BF$ action, rather than by a standard $SL(2, \mathbb{C})$ Chern-Simons theory (the relationship between the two will become clear in due time).

## 5.1   3D Gravity With Cosmological Constant

The first order Palatini-Cartan action for (Euclidean) 3D gravity with cosmological constant $\Lambda$ is

$$S(e, A) = \frac{1}{8\pi G} \int_M e^a \wedge \left( F_a(A) + \frac{\Lambda}{6} \epsilon_{abc} e^b \wedge e^c \right). \tag{232}$$

Here, $e = e^a \tau_a^*$ is a 1-form valued in $\mathfrak{su}(2)^*$ and $F(A)$ is the $\mathfrak{su}(2)$-valued curvature 2-form of a $\mathfrak{su}(2)$ connection 1-form $A = A_a \tau^a$. Note that we have adopted the following change in notation as compared to the previous sections, $\tau_a \rightsquigarrow \tau^a, \tau_*^a \rightsquigarrow \tau_a^*$, so as to keep consistent with the primary reference for this section [35]. We consider the underlying spacetime to be $M \simeq C \times \mathbb{R}$ where $C$ is a 2d surface with boundaries. In what follows $C$ will be a 2-disk $D^2$.

---

[47]Recall that contact with equation (29) from the general CPS treatment of PL symmetries is obtained by choosing a constant reference solution $\varphi_0 \in \mathcal{F}_{\text{EL}}$, which is such that $Q(\varphi_0) = 1$ for all choices of Cauchy surface.

1604 The gauge symmetries of this action are, for $\alpha$ and $Y$ functions valued in $\mathfrak{su}(2)$ and $\mathfrak{su}(2)^*$
1605 respectively, given by:

$$\text{rotations}: \quad \begin{cases} \delta_\alpha A = d_A \alpha \\ \delta_\alpha e = \mathrm{ad}^*_\alpha e \end{cases} \quad \text{and translations}: \quad \begin{cases} \delta_Y A_c = \Lambda \epsilon_{abc} e^a Y^b \\ \delta_Y e^c = dY^c + \epsilon^{abc} A_a Y_b =: (d_A Y)^c \end{cases} \tag{233}$$

1606 The equations of motion are:

$$F_a(A) + \frac{\Lambda}{2} \epsilon_{abc} e^b \wedge e^c = 0 \tag{234}$$

$$d_A e^a = 0, \tag{235}$$

1607 and its associated pSF (pre-symplectic form) current is

$$\omega = \mathbb{d} e^a \wedge \mathbb{d} A_a. \tag{236}$$

1608 In the flat case, i.e. if $\Lambda = 0$, it is immediate to see that one has a $\mathfrak{su}(2)$ $BF$-theory, whose
1609 solutions are given by flat $\mathfrak{su}(2)$ connections $A = h^{-1}dh$ and covariantly constant "$B$-fields"
1610 $e = \mathrm{Ad}^*_h dX$ for some $X \in C^\infty(D^2, \mathfrak{g}^*)$. These solutions are pure-gauge. More compactly,
1611 one can express the solutions to this model as being given by flat $\mathfrak{iso}(3)$ connections
1612 $\mathcal{A} = A_a \tau^a + e^a \tau^*_a$ where $\tau^a$ and $\tau^*_a$ are a basis of the $\mathfrak{su}(2)$ and $\mathfrak{su}(2)^* \simeq (\mathbb{R}^3, +)$ subalgebras
1613 of rotations and translations respectively. Since $\mathfrak{iso}(3)$ is nothing else than the semiflat
1614 Drinfel'd double of $\mathfrak{su}(2)$, it is not too surprising that 2+1D flat (quantum) gravity exhibits
1615 such a symmetry group [66,67]. Another way to assess this finding is to rephrase the $\mathfrak{su}(2)$
1616 $BF$-theory in terms of a $\mathfrak{iso}(3)$ Chern-Simons theory [30]. The advantage of the $BF$
1617 formulation is that it is a first order formulation which splits the fields nicely into the
1618 canonical pair $(e, A)$ matching the double structure $\mathbb{R}^3 \rtimes \mathfrak{su}(2)$. This decomposition also
1619 matches the spin-network basis of quantum gravity and lattice field theory. We will come
1620 back to this point at the end of this section.

1621 Similarly, for $\Lambda \neq 0$, one can rephrase the ($\Lambda$-deformed) $BF$-theory (232) in terms of
1622 a Chern-Simons theory for a double-valued connection $\mathcal{A}$. The relevant double of $\mathfrak{su}(2)$
1623 depends on the signature and the value of the cosmological constant. For Euclidean gravity
1624 with a negative cosmological constant – picking $\Lambda = -1$ without loss of generality – this
1625 is $\mathfrak{sl}(2, \mathbb{C}) \simeq \mathfrak{so}(1, 3)$. A puzzle then arises by noting that the double structure is not quite
1626 reflected in the canonical pair $(e, A)$. Indeed, the on-shell flat connection constructed from
1627 $(e, A)$ is given by:

$$\mathcal{A} = A_a \tau^a + e^a b_a, \tag{237}$$

1628 where $b_a$ are generators of the *boosts* in $\mathfrak{so}(1, 3)$ – which fail to form a subalgebra of $\mathfrak{so}(1,3)$.

1629 In [35] it was shown how to perform a canonical transformation of $(e, A)$ so as to make
1630 the PL symmetries of the theory manifest – once one goes on-shell. That is, to replace
1631 $(e, A)$ with a canonical pair having each component manifestly valued respectively in an
1632 isotropic subalgebra of the double. The first question then is what is the correct double
1633 structure in $\mathfrak{sl}(2, \mathbb{C})$. This is given by the Iwasawa decomposition[48]

$$\mathfrak{sl}(2, \mathbb{C}) \simeq \mathfrak{su}(2) \bowtie \mathfrak{an}(2, \mathbb{C}) \simeq \mathfrak{an}(2, \mathbb{C}) \bowtie \mathfrak{su}(2), \tag{238}$$

---

[48]The rotations-boosts decomposition of $\mathfrak{sl}(2, \mathbb{C})$ is instead an instance of a Cartan decomposition.

where $\mathfrak{an}(2,\mathbb{C})$ is given by traceless upper-triangular $2 \times 2$ matrices with real numbers on the diagonal, namely

$$\mathfrak{an}(2,\mathbb{C}) = \left\{ \begin{pmatrix} r & z \\ 0 & -r \end{pmatrix}, \quad z \in \mathbb{C}, r \in \mathbb{R} \right\}. \tag{239}$$

The Iwasawa decomposition requires the choice of an internal direction[49] $n = n_a \tau^a$ in $\mathfrak{su}(2)$. Explicitly [35],[50]

$$\tau_a^* = b_a + \epsilon_{abc} n^b \tau^c \in \mathfrak{an}(2,\mathbb{C}) \subset \mathfrak{sl}(2,\mathbb{C}), \tag{240}$$

with

$$[\tau_a^*, \tau_b^*]_* = n_a \tau_b^* - n_b \tau_a^*. \tag{241}$$

In (239), we implicitly chose the generators $\tau^a$ to be proportional to the Pauli matrices, and $n = n_a \tau^a$ to be given by $n_a = (0,0,1)$.

The presence of the vector $n$ is ultimately responsible for the "kickback" dressing action $\lhd$ of $\mathfrak{an}(2,\mathbb{C})$ on $\mathfrak{su}(2)$ in $\mathfrak{sl}(2,\mathbb{C}) \simeq \mathfrak{su}(2) \bowtie \mathfrak{an}(2,\mathbb{C})$. Since this action is nothing else than $\widetilde{\mathrm{ad}}^*$, we see that it is in fact directly related to the non-Abelian nature of $\mathfrak{g}^* \simeq \mathfrak{an}(2,\mathbb{C})$.

In this new basis of $\mathfrak{sl}(2,\mathbb{C})$, the double connection reads

$$\mathcal{A} = (A_a - \epsilon_{abc} e^b n^c) \tau^a + e^a \tau_a^* =: K_a \tau^a + e^a \tau_a^*, \tag{242}$$

where we introduced the field redefinition:

$$\begin{pmatrix} e^a \\ A_a \end{pmatrix} \mapsto \begin{pmatrix} e^a \\ K_a \end{pmatrix} := \begin{pmatrix} e^a \\ A_a + \epsilon_{abc} n^b e^c \end{pmatrix}. \tag{243}$$

Recall that the equations of motion are given by the flatness of the double connection $F(\mathcal{A}) = d\mathcal{A} + \frac{1}{2}[\mathcal{A}, \mathcal{A}]_{\mathfrak{d}}$. Then, from $\mathfrak{d} = \mathfrak{sl}(2,\mathbb{C}) \simeq \mathfrak{su}(2) \bowtie \mathfrak{an}(2,\mathbb{C})$, it is easy to see that in terms of the new $\mathfrak{su}(2)$ and $\mathfrak{an}(2,\mathbb{C})$-valued variables $(e, K)$, the equations of motion decompose as:

$$de + \frac{1}{2}[e,e]_* + K \rhd e = 0, \tag{244}$$

$$dK + \frac{1}{2}[K,K] + K \lhd e = 0 \tag{245}$$

where $\lhd e = \widetilde{\mathrm{ad}}_e^*$ and $K \rhd = -\mathrm{ad}_K^*$ for $\mathfrak{g} = \mathfrak{su}(2)$ and $\mathfrak{g}^* = \mathfrak{an}(2,\mathbb{C})$ (note that $\widetilde{\mathrm{ad}}$ depends on $n^a$ according to (241)).

Similarly for the gauge symmetries, the original gauge symmetries (233) are mapped onto standard $\mathfrak{sl}(2,\mathbb{C})$ gauge transformations of the double connection $\mathcal{A}$, which can easily be projected down onto the following $\mathfrak{su}(2)$- and $\mathfrak{an}(2,\mathbb{C})$-transformations of $K$ and $e$:

$$\text{rotations}: \quad \begin{cases} \delta_\alpha K = d_K \alpha + e \rhd \alpha \\ \delta_\alpha e = e \lhd \alpha \end{cases} \quad \text{and translations}: \quad \begin{cases} \delta_Y K = K \lhd Y \\ \delta_Y e = dY + [e, Y]_* + K \rhd Y \end{cases} \tag{246}$$

---

[49]More precisely, this is the choice of a maximal Abelian subalgebra within the boost sector, which is here 1-dimensional and unique up to rotations.

[50]This is best read as a definition of $b_a \in \mathfrak{d}$ in terms of $\tau_a^* \in \mathfrak{g}^*$ and $\tau^a \in \mathfrak{g}$.

Since $n^a$ is an auxiliary quantity, we choose it to be field-*in*dependent, that is $\mathbb{d}n^a \equiv 0$. Then, it is easy to check that the above field redefinition is a canonical transformation in the sense that it preserves the "Darboux" form of $\omega$:

$$\omega = \mathbb{d}e^a \wedge \mathbb{d}A_a = \mathbb{d}e^a \wedge \mathbb{d}K_a - \epsilon_{abc}n^b\mathbb{d}e^a \wedge \mathbb{d}e^c = \mathbb{d}e^a \wedge \mathbb{d}K_a, \tag{247}$$

where we used that since $\mathbb{d}e^a$ is a mixed (1,1)-form of total even degree, it wedge-commutes with $\mathbb{d}e^c$. This canonical transformation can be seen to be generated by the change in polarization determined by

$$\Phi = \frac{1}{2}\epsilon_{abc}n^a e^b \wedge e^c, \tag{248}$$

namely:

$$\theta_{\text{Cartan}} := e^a \mathbb{d}A_a \;\; \rightsquigarrow \;\; \theta_{\text{Iwasawa}} := e^a \mathbb{d}K_a = \theta_{\text{Cartan}} + \mathbb{d}\Phi \;\; \implies \;\; \omega = \mathbb{d}\theta_{\text{Cartan}} = \mathbb{d}\theta_{\text{Iwasawa}}. \tag{249}$$

It is shown in [35] that this canonical transformation can be implemented at the level of the action by adding the pure-boundary term $d\Phi$ (here we also assume that $dn^a = 0$):

$$\begin{aligned} S_{\text{Iwasawa}}(e, K) &= S(e, A) + \frac{1}{8\pi G}\int_M d\Phi \\ &= \frac{1}{8\pi G}\int_M \left(e^a \wedge F_a(K) - \frac{1}{2}\epsilon_{abc}e^a \wedge e^b \wedge d_K n^c\right) \end{aligned} \tag{250}$$

where $F(K) = dK + \frac{1}{2}[K, K]$. Note that both $[K, K]$ and $d_K n$ have terms which are linear in $e$ and quadratic in $n$: using that $n^2 \propto \Lambda$ (here set to $-1$), one obtains the cosmological term; see [35] for details.

Finally, let us note that one also has

$$\omega = \frac{1}{2}\langle \mathbb{d}\mathcal{A} \wedge \mathbb{d}\mathcal{A}\rangle, \tag{251}$$

where $\langle \cdot, \cdot \rangle$ is the Ad-invariant non-degenerate inner product of signature (3,3) over $\mathfrak{d} = \mathfrak{sl}(2, \mathbb{C})$ with respect to which $\mathfrak{g} = \mathfrak{su}(2)$ and $\mathfrak{g}^* = \mathfrak{an}(2, \mathbb{C})$ are isotropic subalgebras, i.e. orthogonal to themselves, and such that $\langle \tau_a^*, \tau^b \rangle = \delta_a^b$ (note: this also means $\langle b_a, \tau^b \rangle = \delta_a^b$). This form of the symplectic structure will turn out to be most useful when implementing the constraints in the next section, and hence when deriving a discretization of $\omega$ in its form (247).

## 5.2 Symmetries & The Continuum Covariant Phase Space

Having laid out all the ingredients needed to study 2+1D gravity (250), we now proceed to study its gauge symmetries and associated constraints.

To maintain consistency with the notation of [35], in this section we redefine the symbols $\Delta$ and $\mathbb{\Delta}$, swapping left and right Maurer-Cartan forms. Namely, from now on

$$\Delta G := G^{-1}dG \quad \text{and} \quad \mathbb{\Delta}G := \mathbb{d}GG^{-1}. \tag{252}$$

It is easy to check that the translation and rotation gauge symmetries (246) are generated, up to boundary terms, by the pullback of the equations of motion (244) and (245) to $D^2$. As said earlier, these are equivalent to the $\mathfrak{sl}(2, \mathbb{C})$ flatness of $\mathcal{A}$:

$$d\mathcal{A} + \frac{1}{2}[\mathcal{A}, \mathcal{A}]_{\mathfrak{d}} = 0 \implies \mathcal{A} = \Delta G \tag{253}$$

for some $G \in C^\infty(D^2, \mathcal{D})$, where in the last equation we used that $D^2$ is simply connected.

In fact, the space of flat connections $\mathcal{A}$ and $\mathcal{D}$-valued functions $G$ are not in one-to-one correspondence, unless we consider $G$ to be actually defined in the coset $\mathcal{D} \backslash C^\infty(D^2, \mathcal{D})$, which makes sense since $G(x)$ and $G'(x) = G_0 G(x)$ define the same connection $\mathcal{A}(x) = \Delta G(x) = \Delta G'(x)$. By ignoring this fact here we are surreptitiously introducing an enlargement of the phase space. As we will see shortly this enlargement will be compensated by a new gauge symmetry and thus have no effect – at least until we will subdivide $\partial D^2$ into segments ending at marked points.

Substituting the expression $\mathcal{A} = \Delta G$ in (251), one obtains the following expression for the CPS symplectic structure

$$\Omega \approx \frac{1}{2} \int_{D^2} \langle \mathbb{d}\Delta G \wedge \mathbb{d}\Delta G \rangle = \frac{1}{2} \int_{D^2} \langle d\mathbb{d}G, d\mathbb{d}G \rangle = \frac{1}{2} \int_{\partial D^2} \langle \mathbb{d}G, d\mathbb{d}G \rangle, \tag{254}$$

where we used the Ad-invariance of the $\mathfrak{d}$-pairing, $\mathfrak{d} = \mathfrak{sl}(2, \mathbb{C})$, as well as the analog of (129) adjusted to the left/right change in notation (252) of the various delta's.

Thus, on-shell, $\Omega$ is supported on the boundary of $D^2$ and its (bulk- or, equivalently, constraint-reduced) field space is isomorphic to $\mathcal{F}_\partial := C^\infty(S^1, \mathcal{D})$ [38, 68].

Using the double decomposition $\mathcal{D} = \mathcal{G}^* \bowtie \mathcal{G}$ of $SL(2, \mathbb{C})$,

$$G =: \ell h, \quad (\ell, h) : D^2 \mapsto AN(2, \mathbb{C}) \bowtie SU(2) \simeq SL(2, \mathbb{C}), \tag{255}$$

one can rewrite[51] the CPS symplectic structure $\Omega$ as [35]

$$\Omega \approx -\mathbb{d} \int_{\partial D^2} \langle \Delta \ell \wedge \mathbb{\Delta} h \rangle, \tag{256}$$

which we recognize matches the first order KS model symplectic structure (221) [36].[52]

There are two obvious symplectomorphisms of $\Omega$ written as in (254): left multiplication by a fixed $G_0 \in \mathcal{D}$ and right multiplication by some (generically point dependent) $G_1(x) \in C^\infty(S^1, \mathcal{D})$,

$$G(x) \mapsto \begin{cases} G_0 G(x) \\ G(x) G_1(x) \end{cases} \tag{257}$$

Using the decomposition of $\mathcal{D}$ into elements of $AN(2, \mathbb{C})$ and $SU(2)$ to write $G = \ell h$ and $G_0 = \ell_0 h_0$ etc., we recognize such left/right multiplications as none other than the four Drinfel'd double actions whose infinitesimal version is provided in (67a)-(67d) (right dressings have here a point-dependent parameter).

---

[51]Writing $\mathcal{A} = \Delta G$, and $G = \ell h$, one sees that the decomposition $\mathcal{A} = K + e$ becomes

$$K = \Delta h + \mathrm{Ad}_h \Delta \ell|_{\mathfrak{su}(2)} \quad \text{and} \quad e = \mathrm{Ad}_h \Delta \ell|_{\mathfrak{an}(2,\mathbb{C})}.$$

where the projections on $\mathfrak{su}(2)$ and $\mathfrak{an}(2, \mathbb{C})$ are present given that $\mathfrak{an}(2, \mathbb{C}) \subset SL(2, \mathbb{C})$ is not stable under the conjugation by elements of $SU(2) \subset SL(2, \mathbb{C})$. In particular, one sees that $h$ and $\ell$ are not quite the holonomies (path-ordered exponentials) of $K$ and $e$.

[52]For an exact match up to an irrelevant global sign, we should map $(\ell, h) \mapsto (\tilde{h}, \tilde{\ell}) = (h^{-1}, \ell^{-1})$. Indeed, the symbols $\Delta$ and $\mathbb{\Delta}$ used in this section follow opposite left-right conventions compared to the previous section, and despite $(\ell, h)$ and $(\tilde{h}, \tilde{\ell})$ transforming differently under left/right dressing actions, $(\ell^{-1}, h^{-1})$ and $(\tilde{h}, \tilde{\ell})$ transform the same way.

The point-dependent right actions by $G_1(x)$ are nothing but gauge transformations $\mathcal{A} \mapsto$ $\mathrm{Ad}_{G_1^{-1}(x)}\mathcal{A} + G_1^{-1}(x)dG_1(x)$. It is easy to check that the corresponding point-dependent right rotations (67a) and translations (67b) are Noetherian transformations:

$$\mathbb{i}_{\mathbb{r}(Y_1(x),\alpha_1(x))}\Omega \approx \mathbb{d} \int_{\partial D^2} \langle Y_1(x) + \alpha_1(x), \mathcal{A} \rangle = \mathbb{d} \int_{\partial D^2} \langle Y_1(x), K \rangle + \langle \alpha_1(x), e \rangle, \qquad (258)$$

where $G_1(x) = (\ell_1(x), h_1(x)) \sim (1,1) + (Y_1(x), \alpha_1(x))$ and $K$ and $e$ are given in Footnote 51.

Conversely, since $\Omega$ is ultimately equal to $\frac{1}{2}\int_{D^2}\langle \mathbb{d}\mathcal{A} \wedge \mathbb{d}\mathcal{A} \rangle$, and $\mathcal{A} = \Delta G$ is invariant under left multiplication of $G$ by a constant $G_0$, the left rotations (67c) and translations (67d) have trivial generators, meaning that:

$$\mathbb{i}_{\mathbb{l}(Y,\alpha)}\Omega \approx 0. \qquad (259)$$

As we said earlier, this reflects the fact that there is an ambiguity in the choice of the field $G(x)$ given a flat $\mathcal{A}$ which amounts to space-independent left-translations, $G(x) \mapsto G_0 G(x)$. This ambiguity can be used to make $G(x)$ the identity at a reference point $x_c \in D^2$. This amounts to defining $G_{cx} \equiv G_c(x) := \mathrm{Pexp}\int_{x_c \to x}\mathcal{A}$.

We conclude this discussion by coming back to the fact that, on-shell, $\Omega$ (256) is the symplectic structure of the first order model of Klimčík and Ševera (221) evaluated on a *closed* string [36]. Recalling that the KS closed string supports *no* non-trivial Poisson-Lie symmetries, §4.4, we are now interested in exploring the 2+1D gravity analog of the *open* KS string.

In this context, the KS open string becomes meaningful if on $S^1 = \partial D^2$ there are marked points. This is the case e.g. if $D^2$ is taken to be one polygon in a cellular decomposition of the Cauchy surface of the 2+1D manifold $M$ [35].

Therefore, in preparation for the next section it is instructive to compute the zero on the right hand side of equation (259) from a slightly different perspective. Starting from the right-most expression of $\Omega$ in (254), which is supported on the boundary, we compute:

$$\mathbb{i}_{\mathbb{l}(Y,\alpha)}\Omega \approx \frac{1}{2}\int_{\partial D^2}\langle Y + \alpha, d\Delta G \rangle = \frac{1}{2}\int_{\partial D^2} d\langle Y + \alpha, \Delta G \rangle = 0 \qquad (260)$$

where we used that $Y$ and $\alpha$ are constant and that $\partial^2(D^2) \equiv 0$. What this computation suggests is that, were we to subdivide $\partial D^2$ into segments, the zero on the right-hand side of the above equation could be obtained as the result of a telescopic sum whose non-zero terms are associated to the segments of the subdivision. In other words, the Drinfel'd left rotations and translations can be consistently assigned non-trivial action on the phase space associated to each such segment. In the next section we will make sense of this idea.

However, one might wonder how constant left multiplications of $G(x)$ by $G_0$ can play a role when they are mere redundancy in the parametrization of $\mathcal{A}$ in terms of $G$. The answer relies in the fact that upon discretization one is led to consider piecewise constant transformations. That is, splitting $S^1 \simeq \partial D^2$ into segments joining at marked points, it is only when acting *with* the same element on all segments, that one expects an identically vanishing charge. But this cannot be established when acting on $G(x)$ one segment at a time. The action on $G(x)$ over $S^1$ by a piecewise constant $G_0$ (i.e. by a $G_0$ that is constant on each segment, but not across segments) produces a connection that is singular at the transition between segments. These singularities will play a role in the next section.

### 5.3 Discretization & Poisson-Lie Symmetries

Motivated by our discussions in the previous section, instead of focusing on a single $D^2$ region with boundaries, we now consider a 2D surface $S$ split into polygonal 2-cells $c_i \simeq D^2$ (where the $\simeq$ is a $C^0$ homeomorphism). The boundary $\partial c_i$ of each 2-cell splits into 1-cells (segments) corresponding to the sides of the polygonal 2-cell, so that the boundaries of these 1-cells are the marked points on $\partial c_i$ mentioned at the end of the previous section.

Each 1-cell corresponds to a pair of adjacent 2-cells,

$$\partial c = \bigcup_{c'} [cc'] \quad \text{where} \quad [cc'] := \partial c \cap \partial c'. \tag{261}$$

Thus, recalling (247) and (256),

$$\Omega_S := \int_S \mathrm{d}e^a \wedge \mathrm{d}K_a = \sum_i \Omega_{c_i} \quad \text{where} \quad \Omega_c := -\mathrm{d} \sum_{c'} \int_{[cc']} \langle \Delta \ell_c \wedge \Delta h_c \rangle. \tag{262}$$

The label $\bullet_c$ in $(\ell_c, h_c)$ serves as a reminder that these quantities are defined in terms of a choice of function $G_c(x) = \ell_c(x) h_c(x)$ *within* every cell.

The above is thus readily reorganized in terms of contributions associated to the 1-skeleton of the cellular decomposition:

$$\Omega_S = \sum_{c,c'} \Omega_{[cc']} \quad \text{where} \quad \Omega_{[cc']} := \Omega_c - \Omega_{c'} = -\mathrm{d} \int_{[cc']} \Big( \langle \Delta \ell_c \wedge \Delta h_c \rangle - \langle \Delta \ell_{c'} \wedge \Delta h_{c'} \rangle \Big). \tag{263}$$

Having isolated and independently described the phase space in each cell, we must provide extra "gluing constraints" that allow one to put the different cells together, namely:

$$\mathcal{A}_c(x) = \mathcal{A}_{c'}(x) \iff \partial_x \Big( G_c(x) G_{c'}^{-1}(x) \Big) = 0 \quad \forall x \in [cc']. \tag{264}$$

These gluing constraints can therefore be expressed in terms of the existence of (*constant*) elements $G_{cc'} \equiv G_{c'c}^{-1} \in \mathcal{D} = SL(2, \mathbb{C})$, such that $G_c(x) G_{c'}^{-1}(x) = G_{cx} G_{xc'} = G_{cc'}$ for all $\forall x \in [cc']$.

We are imposing gluing constraints only at the edges (1-cells) of the cellular complex, not at its vertices (0-cells). Indeed, singularities, which are interpretable as point-particle defects [54,67,69–72], might a priori be present at the vertices. Imposition of their absence thus constitutes our discrete equations of motion (in the absence of sources). As we will see this is related to the discussion at the end of the previous section and to the presence of Poisson-Lie symmetries classifying those defects.

This picture is clarified by thinking of the $G_c(x)$ as holonomies of $\mathcal{A}$ from a reference point $x_c \in c$ to any other $x \in c$, namely $G_c(x) = \mathrm{Pexp} \int_{x_c \to x} \mathcal{A}$. Then it is clear that the gluing constraints (264) state that all holonomies from $x_c$ to $x_{c'}$ through $[cc']$ are equal, which is the statement of flatness in $c \cup c'$, i.e. of flatness across $[cc'] = \partial c \cap \partial c'$. Under this condition, the holonomy around a vertex $v$ sitting at the intersection of three cells $c$, $c'$, $c''$, is then given by $G_v := G_{cc'} G_{c'c''} G_{c''c}$ and it equals 1 if and only if the curvature of $\mathcal{A}$ vanishes at $v$. Therefore, if we were to impose flatness everywhere, the elements $G_{cc'}$ would be like "transition functions" over the 2d discretized manifold covered by the atlas $\{c\}$. However, since we are here *not* a priori imposing flatness at the vertices, the only

relation between the gluing functions $G_{cc'}$'s is $G_{cc'}G_{c'c} = 1$, and the vertex monodromies will be among[53] the residual degrees of freedom one obtains after gluing.

After imposing the gluing constraints, one can show [35, Theorem 1] that the phase space assigned to each link $[cc']$ is given by a copy of the Heisenberg double $\mathcal{D}_H$ equipped with its STS symplectic form (64):[54]

$$\Omega_{[cc']} = \frac{1}{2}\Big(\langle \triangle H_{cc'}^v \wedge \triangle L_{vv'}^c \rangle + \langle \triangle (H_{cc'}^{v'})^{-1} \wedge \triangle (L_{vv'}^{c'})^{-1} \rangle \Big), \tag{265}$$

where, for $v, v'$ being 0-cells (vertices) of the cellular decomposition such that $\partial[cc'] = v - v'$ and $G_c(x) =: \ell_c(x)h_c(x)$. We introduced $L_{vv'}^c, L_{vv'}^{c'}$ and $H_{cc'}^v, H_{cc'}^{v'}$:

$$\begin{cases} L_{vv'}^c := \ell_c(v)^{-1}\ell_c(v') \\ L_{vv'}^{c'} := \ell_{c'}(v)^{-1}\ell_{c'}(v') \end{cases} \quad \text{and} \quad \begin{cases} H_{cc'}^v := h_c(v)h_{c'}^{-1}(v) \\ H_{cc'}^{v'} := h_c(v')h_{c'}^{-1}(v') \end{cases} \tag{266}$$

These variables are the classical version of Kitaev's triangular operators [73, 74]. For example, $H_{cc'}^v$ can *loosely*[55] be thought of as a $SU(2)$ holonomy along a path from $x_c$ to $x_{c'}$ through $v$; and similarly $L_{vv'}^c$ can be though of as a $AN(2, \mathbb{C})$ holonomy along a path from $v$ to $v'$ through $x_c$. With this picture in mind one sees that the "symplectic duality" between $L_{vv'}^c$ and $H_{cc'}^v$, is reflected onto the duality between a cellular decomposition and its Poincaré dual, for if the $H_{cc'}^v$ are thought of as supported on the 1-skeleton of the cellular decomposition of $S$, the $L_{vv'}^c$ are supported on the 1-skeleton of its Poincaré dual.

To more easily recognize (265) as the STS symplectic form of equation (64), note that the *ribbon structure* equation

$$L_{vv'}^c H_{cc'}^{v'} = H_{cc'}^v L_{vv'}^{c'}, \tag{267}$$

which readily follows from the gluing constraint $G_c(v)G_{c'}^{-1}(v) = G_c(v')G_{c'}^{-1}(v')$ for $v$ and $v'$ the endpoints of $[cc']$ (264), serves as the discrete analog of the continuum relation $G = \ell h = \tilde{h}\tilde{\ell}$.

Equation (265) shows that, upon a parameterization of the degrees of freedom in terms of holonomies in the bulk of each cell, and after imposing the gluing constraints,[56] one is left with a finite dimensional phase space constituted by the product of a copy of the Heisenberg double for each link of the 1-skeleton of the cellular decomposition:

$$\Omega_S = \sum_{[cc']} \Omega_{[cc']}, \qquad \Omega_{[cc']} = -\Omega_{\text{STS}}\big|_{(\ell,h)=(L_{vv'}^c, H_{cc'}^{v'})}. \tag{268}$$

Taking the semi-Abelian limit of the Heisenberg double, $AN(2, \mathbb{C}) \to \mathbb{R}^3$ and thus $SL(2, \mathbb{C}) \to ISO(3) \simeq SU(2) \ltimes \mathbb{R}^3 \simeq T^*SU(2)$, one readily sees that this generalizes the $\Lambda = 0$ result in 2+1D Euclidean gravity, where the discretization procedure famously yields a copy of $T^*\mathcal{G}$, $\mathcal{G} = SU(2)$, for each link [75].

---

[53]If the discretized surface $S$ is not simply connected, other topological degrees of freedom will emerge as monodromies around the non-contractible cycles of $S$.

[54]From (252) we have $\triangle(\bullet)^{-1} = -(\bullet)^{-1}\mathbb{d}(\bullet)$.

[55]Cf. footnote 51.

[56]At rigor, "imposing the gluing constraints" requires also the (coisotropic) reduction of such constraints. It is this step that reduces the infinite number of degrees of freedom present in (262) to the finite number of degrees of freedom present in (268). It is also responsible for turning a $\mathbb{d}$-exact symplectic structure (262) into one that might not be $\mathbb{d}$-exact (268) (in the fully non-Abelian case). We recognize this reduction step has not been studied carefully (e.g., what is the flow generated by the gluing constraint?) but leave it for future work.

Typically, in the $\Lambda = 0$ case, one subsequently imposes residual $SU(2)$ invariance at the 2-cells (faces) and flatness at the 0-cells (vertices) of the discretization of $S$. The former corresponds to a torsionless condition on the connection, i.e. to a discrete version of the dual flatness equation (244), also called the Gauss constraint. In the discrete these take the form:

$$\sum_{[vv'] \in \partial c} X^c_{vv'} = 0 \quad \text{and} \quad \prod_{[cc'] \ : \ v \in \partial[cc']} H^v_{cc'} = 1. \tag{269}$$

where the second is in fact an ordered product, yielding e.g. $H^v_{cc'} H^v_{c'c''} H^v_{c''c'''} H^v_{c'''c} = 1$ etc.

These discrete Gauss (or, $\mathcal{G}^*$-flatness) and $\mathcal{G}$-flatness conditions have precise analogs in the general Poisson-Lie case, with the exact same interpretation, namely:

$$\prod_{[vv'] \in \partial c} L^c_{vv'} = 1 \quad \text{and} \quad \prod_{[cc'] \ : \ v \in \partial[cc']} H^v_{cc'} = 1. \tag{270}$$

where the semiflat limit is recovered by setting $L^c_{vv'} = 1 + X^c_{vv'} + \dots$ as usual (see Figure 1). To solidify the various Drinfel'd double actions as Poisson-Lie symmetries of the

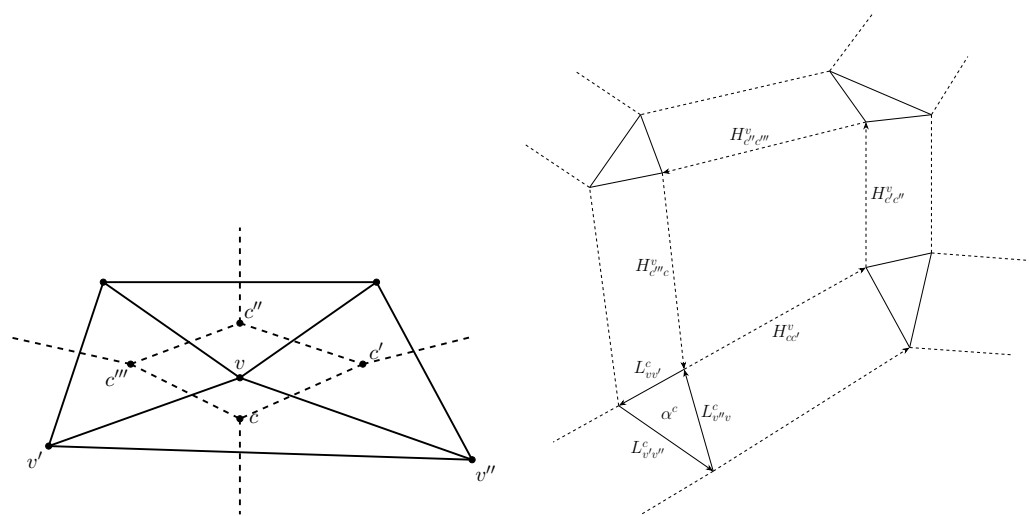

Figure 1: We illustrate how the ribbon variables are associated to a triangulation and its dual complex. In particular, the expression of the constraint makes clear the action of the symmetry transformation on the phase space variables.

discrete model, according to our Definition 2.5, they must also leave invariant (on-shell) the discrete equations of motion (270). Note that a Drinfel'd dressing action has to act diagonally on various elements of the discrete phase space (268). E.g. a left $\mathfrak{su}(2)$ action is attached to a 2-cell of the cellular decomposition and thus acts on all and only the $L^c_{vv'}$ and $H^v_{cc'}$ adjacent to it. However, for consistency, the symmetry parameter has to be parallel transported from one cell to the next. Using left rotations (67c) as an example, one has e.g. [35]:

$$\delta^L_\alpha(\ell_1 \ell_2) = (\delta^L_\alpha \ell_1)\ell_2 + \ell_1(\delta^L_{\alpha \lhd \ell_1} \ell_2) = \alpha(\ell_1 \ell_2) - \ell_1 \ell_2(\alpha \lhd (\ell_1 \ell_2)). \tag{271}$$

Thus, on-shell invariance of the discrete Gauss constraint in (270) under left rotations follows from:

$$\delta^L_{\alpha^c}(L^c_{vv'} L^c_{v'v''} L^c_{v''v}) = \alpha^c(L^c_{vv'} L^c_{v'v''} L^c_{v''v}) - L^c_{vv'} L^c_{v'v''} L^c_{v''v}(\alpha^c \lhd (L^c_{vv'} L^c_{v'v''} L^c_{v''v}))$$
$$\approx \alpha^c - (\alpha^c \lhd 1) \approx \alpha^c - \alpha^c \approx 0. \tag{272}$$

The on-shell invariance of the discrete flatness constraint in (270) is slightly more subtle. As a result of the ribbon structure equation (267) the action of $\alpha^c$ on a closed loop of holonomies $H^v_{cc'} H^v_{c'c''} H^v_{c''c'''} H^v_{c'''c}$ acts from opposite directions on $H^v_{cc'}$ and $H^v_{c'''c}$, and does not act on the holonomies $H^v_{c'c''}$ and $H^v_{c''c'''}$ which are not passing through $c$. Thus we have,

$$\delta_{\alpha^c}(H^v_{cc'} H^v_{c'c''} H^v_{c''c'''} H^v_{c'''c}) = (\delta^L_{\alpha^c} H^v_{cc'}) H^v_{c'c''} H^v_{c''c'''} H^v_{c'''c} + (H^v_{cc'} H^v_{c'c''} H^v_{c''c'''} \delta^R_{\alpha^c} H^v_{c'''c})$$
$$= \alpha^c H^v_{cc'} H^v_{c'c''} H^v_{c''c'''} H^v_{c'''c} - H^v_{cc'} H^v_{c'c''} H^v_{c''c'''} H^v_{c'''c} \alpha^c \approx 0, \quad (273)$$

where we have used a left action on $H^v_{cc'}$ and a right action on $H^v_{c'''c}$. Similar calculations proceed for checking the on-shell invariance of the discrete equations of motion under left and right translations generated by some $Y^v \in \mathfrak{an}(2, \mathbb{C})$, now attached to a *vertex* of the cellular decomposition.

The Poisson-Lie momentum maps associated to these Drinfel'd double actions are then the discrete analogs of the standard results (68), (69), up to a global conventional minus sign as was captured in (268). That is, the various Drinfel'd double actions respectively admit the PL flow equations [35]:

$$\mathring{\mathbb{i}}_{\mathbb{r}(\alpha^{c'})} \Omega_{[cc']} = \langle \alpha^{c'}, \mathbb{\Delta}(L^{c'}_{vv'})^{-1} \rangle, \qquad \mathring{\mathbb{i}}_{\mathbb{r}(Y^{v'})} \Omega_{[cc']} = -\langle Y^{v'}, \mathbb{\Delta}(H^{v'}_{cc'})^{-1} \rangle \qquad (274)$$

$$\mathring{\mathbb{i}}_{\mathbb{l}(\alpha^c)} \Omega_{[cc']} = \langle \alpha^c, \mathbb{\Delta} L^c_{vv'} \rangle, \qquad \mathring{\mathbb{i}}_{\mathbb{l}(Y^v)} \Omega_{[cc']} = -\langle Y^v, \mathbb{\Delta} H^v_{cc'} \rangle. \qquad (275)$$

Summarizing, the rather non-trivial fact which we have arrived at is that by going from the continuum to the discrete, i.e. by introducing a cellular decomposition of the 2-surface $S$ and subsequently imposing the constraints in the bulk of each cell, we traded a combination of trivial (259) and Noetherian (gauge) (258) symmetries for a set of *residual non-trivial Poisson-Lie symmetries*. These symmetries are constituted by copies of the Drinfel'd double $\mathcal{D}_D \simeq AN(2, \mathbb{C}) \bowtie SU(2)$ acting on copies of the corresponding Heisenberg double $\mathcal{D}_H$ associated to the 1-cells of the discretization. This was in turn produced by combining, for each 1-cell $[cc'] = \partial c \cap \partial c'$, two copies of the Klimčík-Ševera phase space, each corresponding to the contribution to $\Omega_{[cc']}$ coming from either one of the two 2-cells $c$ and $c'$.

# 6 Conclusion & Outlook

In this work we investigated how (global) Poisson-Lie symmetries, governed by the action of a Poisson-Lie group $(\mathcal{G}, \pi_{\mathcal{G}})$ on a symplectic phase space, manifest themselves in various low dimensional field theories from a Lagrangian perspective using the covariant phase space framework.

The hallmark of a Poisson Lie action is that its (generalized) momentum map, or "charge", is valued in a *group*, $\mathcal{G}^*$. This generalizes the notion of momentum maps for Hamiltonian actions, which are instead valued in the dual of the Lie algebra, $\mathfrak{g}^*$, a linear space.

As a consequence of this fact, Poisson-Lie actions fail to be symplectomorphisms (([33], Theorem 2.9). Note that this failure happens in a controlled manner, since the terms encoding the lack of symplectomorphicity take a very specific shape, namely that of (33) or (49):

$$\mathbb{L}_{\mathbb{Q}(\alpha)} \Omega \approx \mp \frac{1}{2} \langle [\mathbb{Q}, \mathbb{Q}]_*, \alpha \rangle \quad \text{iff} \quad \mathbb{L}_{\mathbb{Q}(\alpha)} \pi \approx (\mathbb{Q} \wedge \mathbb{Q}) \delta(\alpha), \qquad (276)$$

where $\mathbb{Q} = \mathring{\mathbb{i}}_{\mathbb{Q}(\cdot)} \Omega$, while the Lie bracket $[\cdot, \cdot]_* : \mathfrak{g}^* \wedge \mathfrak{g}^* \to \mathfrak{g}^*$ and cocycle $\delta : \mathfrak{g} \to \mathfrak{g} \wedge \mathfrak{g}$ are dual objects containing the same information that fully encodes both the Poisson structure

| Dim. | Model | Poisson-Lie action | Poisson-Lie charge |
|------|-------|--------------------|--------------------|
| 0+1D | Generalized Spinning Top | local, off-shell | off-shell |
|      | Deformed Generalized Spinning Top | local, off-shell | off-shell |
| 1+1D | Second Order KS String (Open) | non-local, on-shell | on-shell |
|      | Second Order KS String (Closed) | non-local, on-shell | identity (constraint) |
|      | First Order KS String (Open) (*) | local, off-shell | off-shell |
|      | First Order KS String (Closed) (*) | local, off-shell | identity (constraint) |
| 2+1D | Palatini-Cartan Gravity with C.C. | local, on-shell(**) | on-shell (**) |

(*) lacks manifest spacetime covariance
(**) requires marked points

Table 1: The different types of action and charges for the different studied models.

on $\mathcal{G}$ and the Lie group structure on $\mathcal{G}^* = \exp \mathfrak{g}^*$; both $[\cdot, \cdot]_*$ and $\delta$ vanish in the Abelian limit of $\mathcal{G}^*$. Notice that $[\cdot, \cdot]_*$ and $\delta$ must satisfy particular consistency conditions with the Lie algebra $(\mathfrak{g}, [\cdot, \cdot])$ to provide a consistent Poisson-Lie action.

From a Lagrangian perspective, where $\Omega$ is the covariant phase space symplectic structure on the space of solutions to the equations of motion $\mathcal{F}_{\text{EL}}$, the above generalization immediately implies that Poisson-Lie actions with a non-Abelian $\mathcal{G}^*$ go *beyond* the standard Noetherian framework of Lagrangian symmetries (20), since the latter naturally yield Hamiltonian actions and $\mathfrak{g}^*$-valued charges.

Not only does one not expect a Poisson-Lie symmetry to leave the Lagrangian invariant, but from equation (276) one can also argue that a Poisson-Lie symmetry will in general also fail to be a local action—which is a precondition on any Lagrangian symmetry. Indeed in 1+1 or higher dimensions, since $\Omega$ is the integral of a local expression in the fields and their variations, if $\mathbb{Q}$ were a local action then we would have on the left-hand side of (276) the integral of a local expression, but a product of *two* integrals of a local expression on the right-hand side. This tension can be resolved by means of some non-local features of $\mathbb{Q}$ or if, somehow, those integrals localize at points. Both situations arise in the examples analyzed in this article, as we will soon recall.

In summary, Poisson-Lie symmetries go beyond the Noetherian framework in many respects.

As discussed in §2.3, the Noether construction of the current, which relies on the invariance of the Lagrangian off-shell up to boundary terms, is a very powerful tool to ensure that we can construct a charge which does not depend on the chosen slicing, i.e. defined in a covariant way. Not having this prescription at hand considerably complicates the covariant construction of a conserved charge. We thus proposed in Definition 2.5 a new framework to understand symmetries in the covariant phase space which is capable of encompassing not only Lagrangian but also non-trivial Poisson-Lie symmetries as well. Key to the proposal is a delicate balance on the degree of non-locality of the charges. As mentioned at the end of §2.3, it would be interesting in the future to relate the conservation of the conserved charges as defined in Definition 2.5 with Dirac's hypersurface deformation algebra.

We tested this new framework through in depth analyses of Poisson-Lie symmetric low dimensional field theories. A summary of the different cases is given in the Table 1. This included the 0+1D generalized spinning top and deformed spinning top, the 1+1D Klimčík and Ševera (KS) open and closed strings, and 2+1D gravity with cosmological constant. The KS strings were analyzed both in their first order and second order formulations,

although most of our focus was laid on the second order formulation because it is the only manifestly spacetime covariant formulation of the model and also the least studied in the literature.

These models exhibited both proposed resolutions to the aforementioned issue of non-locality. This was most clearly illustrated in our analysis of the second order and first order KS models, respectively through a *non-local* Poisson-Lie symmetry we called "twisted right rotations" (cf. §4.3, in particular Definition 4.3 and Theorem 4.8), and a localization of the conserved Poisson-Lie charges onto the endpoints of the string (cf. §4.5).

We would like to conclude by highlighting potential future areas of exploration: Poisson-Lie symmetries in higher dimensional field theories, and the connection between Poisson-Lie symmetric field theories and field theories on non-commutative spacetimes.

The most obvious next step is to find more covariant field theories with Poisson-Lie symmetry, especially in higher dimensions than those studied so far. We are particularly interested in 4D gravity, where early signs suggest that ideas from quantum group representation theory might be important in the quantum version of the theory [76–79]. This could help build a strong and general framework for further understanding Poisson-Lie symmetries, and also clarify how quantum group symmetries originate in classical field theory. However, similar challenges are expected to arise, especially those related to non-locality. Solving these issues in higher-dimensional theories may require concepts from higher gauge theory, such as surface-ordered integrals, 2-connections, and 2-gauge symmetries [65].

Upon quantization, Poisson-Lie symmetries give rise to quantum group symmetries. On the other hand, spacetimes with quantum group symmetries are non-commutative spaces [23,80,81]. We are therefore intrigued by the prospect of an interesting interplay between between Poisson-Lie symmetric field theories and the formulation of field theories on non-commutative (quantum) spacetimes (see for example [82–84] for a very limited list of references on this vast topic). With this in mind, we note that Noether symmetries have already been studied for some specific examples of non-commutative geometry [85, 86]. Thus, it would be interesting to explore if and how this non-commutative field theory framework can be characterized in terms of our covariant phase space approach to Poisson-Lie symmetries. This would require carefully distinguishing and relating non-commutative spacetimes with non-commutative field spaces. At first glance, these are expected to match only when the target space of the field theory is spacetime itself, as is the case in particle and string mechanics.

# Acknowledgements

Research at Perimeter Institute is supported in part by the Government of Canada through the Department of Innovation, Science and Economic Development and by the Province of Ontario through the Ministry of Colleges and Universities.

**Funding information** Authors are required to provide funding information, including relevant agencies and grant numbers with linked author's initials. Correctly-provided data will be linked to funders listed in the Fundref registry.

# A  Constructing $E_{ab}^{\mu\nu}$

In this appendix we show how a construction of Klimčík and Ševera [34] can be generalized to produce a tensor $E_{ab}^{\mu\nu}$ that satisfies the fundamental KS conditions (132) and (133), which we report here for the reader's convenience:

$$\text{there exists a } \widetilde{E}_{\mu\nu}^{bc} \text{ such that } E_{ab}^{\mu\nu}\widetilde{E}_{\nu\rho}^{bc} = \delta_\rho^\mu\delta_a^c \tag{A.1}$$

$$\mathrm{d}E_{ab}^{\mu\nu} = (\mathbb{\Delta}^c\tilde{h})\,\tilde{c}^{a'b'}{}_c E_{aa''}^{\mu\mu'}E_{bb''}^{\nu\nu'}(\mathrm{Ad}_{\tilde{h}})^{a''}{}_{a'}(\mathrm{Ad}_{\tilde{h}})^{b''}{}_{b'}\epsilon_{\mu'\nu'}. \tag{A.2}$$

Once this construction is established, we use it to give a first order version of the KS-model in the form of a Poisson-Lie $\sigma$-model featuring a symmetry breaking term.

Henceforth, we will assume that the target space $\mathcal{G}$ has been coordinatized with arbitrary coordinates $z^\alpha$. Contravariant tensors on $\mathcal{G}$ are then denoted by $\bullet^{\alpha\beta\cdots}$ indices. These are not to be confused with $\bullet^{ab\cdots}$ indices, which denote elements of $\mathfrak{g}\otimes\mathfrak{g}\otimes\cdots$. However, by means of a (right) trivialization, $T\mathcal{G}\simeq\mathcal{G}\times\mathfrak{g}$, contravariant *k-tensors* and $\mathfrak{g}^{\otimes k}$-valued *functions* can be appropriately mapped onto each other. This will play a role below.

Following Klimčík and Ševera, consider on $\mathcal{G}$ a symmetric, invertible, and right-invariant *tensor* $M^{\alpha\beta}$, and a multiplicative Poisson *tensor* $\pi^{\alpha\beta}$,

$$\mathbb{L}_{\mathbb{r}(\xi)}M^{\alpha\beta} = 0 \quad \text{and} \quad \mathbb{L}_{\mathbb{r}(\xi)}\pi^{\alpha\beta} = -(\mathbb{r}^\alpha\otimes\mathbb{r}^\beta)(\delta(\xi)), \tag{A.3}$$

where

$$\delta:\mathfrak{g}\to\mathfrak{g}\wedge\mathfrak{g}, \quad \xi = \xi^a\tau_a \mapsto \delta\xi = \xi^c\tilde{c}_c{}^{ab}\tau_a\otimes\tau_b, \tag{A.4}$$

for $\tilde{c}_c{}^{ab} = \tilde{c}_c{}^{[ab]}$ a set of structure constants for the Lie algebra $(\mathfrak{g}^*,[\cdot,\cdot]_*)$. By Theorem 2.13, the condition that the Poisson bivector $\pi$ on $\mathcal{G}$ is multiplicative, i.e. that $(\mathcal{G},\pi)$ is a Poisson-Lie group (Definition 2.6), is equivalent to asking that $\mathfrak{d} = \mathfrak{g}\bowtie\mathfrak{g}^*$ is a classical double.

Using the right invariant vector fields $\{\mathbb{r}(\tau_a)\}_{a=1}^{\dim(\mathfrak{g})}$ as a basis of vector fields on $\mathcal{G}$, understood as a $C^\infty(\mathcal{G})$-module, we can write (with a slight abuse of notation):

$$M^{\alpha\beta} = \sum_{a,b} M^{ab}\mathbb{r}^\alpha(\tau_a)\otimes\mathbb{r}^\beta(\tau_b) \quad \text{and} \quad \pi^{\alpha\beta} = \sum_{a,b}\pi^{ab}\mathbb{r}^\alpha(\tau_a)\otimes\mathbb{r}^\beta(\tau_b), \tag{A.5}$$

where $M^{ab} \doteq (\mathbb{\Delta}^a\tilde{h}^{-1}\otimes\mathbb{\Delta}^b\tilde{h}^{-1})(R)$ and $\pi^{ab} \doteq (\mathbb{\Delta}^a\tilde{h}^{-1}\otimes\mathbb{\Delta}^b\tilde{h}^{-1})(\pi)$ are a collection of $(\mathfrak{g}\otimes\mathfrak{g})$-valued *scalar functions* on $\mathcal{G}$, respectively symmetric and skew under the exchange of the indices $a$ and $b$. Since $\mathbb{L}_{\mathbb{r}(\xi)}\mathbb{\Delta}\tilde{h}^{-1} = 0$, one has that the above conditions can be rewritten as

$$\mathrm{d}M^{ab} = 0 \quad \text{and} \quad \mathrm{d}\pi^{ab} = -\mathbb{\Delta}^c\tilde{h}\tilde{c}_c{}^{a'b'}(\mathrm{Ad}_{\tilde{h}})_{a'}^a(\mathrm{Ad}_{\tilde{h}})_{b'}^b. \tag{A.6}$$

From the $(\mathfrak{g}\otimes\mathfrak{g})$-valued constant $M^{ab}$ and scalar $\pi^{ab}$, one introduces

$$\widetilde{E}^{ab} \doteq M^{ab} + \pi^{ab} \quad \text{and} \quad E_{ab} \doteq (\widetilde{E}^{ab})^{-1}, \tag{A.7}$$

which exists for $M^{ab}$ "large enough", i.e. if $\pi M^{-1}$ has eigenvalues of norm strictly less than 1. In turn, decomposing $E_{ab}$ into its symmetric and skew parts, one also defines

$$G_{ab} \doteq E_{(ab)} \quad \text{and} \quad B_{ab} \doteq E_{[ab]}. \tag{A.8}$$

The first can be interpreted as a metric on $\mathcal{G}$ and the second as a 2-magnetic potential, also known as a Kalb-Ramond potential (see Appendix B).

Finally, using the above ingredients together with a non-degenerate (Lorentzian) worldsheet metric $\gamma_{\mu\nu}$ and its compatible Levi-Civita tensor $\epsilon_{\mu\nu}$, one defines:

$$E_{ab}^{\mu\nu}(\tilde{h}, x) \doteq G_{ab}(\tilde{h})\gamma^{\mu\nu}(x) + B_{ab}(\tilde{h})\epsilon^{\mu\nu}(x). \tag{A.9}$$

The following theorem is a simple generalization of results of Klimčík and Ševera:

**Theorem A.1** (Klimčík and Ševera)**.** Let $M$ and $\pi$ be as in (A.3) or, equivalently, (A.6). Then, if $\pi M^{-1}$ has eigenvalues of norm smaller than 1, $E_{ab}^{\mu\nu}(\tilde{h}, x)$ constructed as above is well-defined and satisfies the KS conditions (A.1,A.2), with inverse

$$\widetilde{E}_{\mu\nu}^{ab} = M^{ab}\gamma_{\mu\nu} + \pi^{ab}\epsilon_{\mu\nu}. \tag{A.10}$$

$\diamond$

*Proof.* First we decompose (A.2) onto its symmetric and skew parts:

$$\mathbb{d}E_{ab}^{\mu\nu} = (\mathbb{A}^c\tilde{h})\, \tilde{c}_v^{\,a'b'}\Big((G_{a''a}G_{b''b} - B_{a''a}B_{b''b})\epsilon^{\mu\nu} + (G_{a''a}B_{b''b} - B_{a''a}G_{b''b})\gamma^{\mu\nu}\Big)(\mathrm{Ad}_{\tilde{h}})_{a'}^{a''}(\mathrm{Ad}_{\tilde{h}})_{b'}^{b''}. \tag{A.11}$$

as well as

$$\mathbb{d}E_{ab}^{\mu\nu} = (\mathbb{d}B_{ab})\epsilon^{\mu\nu} + (\mathbb{d}G_{ab})\gamma^{\mu\nu}, \tag{A.12}$$

Comparing, one obtains the following rewriting of (A.2):

$$\begin{cases} \mathbb{d}B_{ab} &= (\mathbb{A}^c\tilde{h})\, \tilde{c}_c^{\,a'b'}(G_{a''a}G_{b''b} - B_{a''a}B_{b''b})(\mathrm{Ad}_{\tilde{h}})_{a'}^{a''}(\mathrm{Ad}_{\tilde{h}})_{b'}^{b''} \\ \mathbb{d}G_{ab} &= (\mathbb{A}^c\tilde{h})\, \tilde{c}_c^{\,a'b'}(G_{a''a}B_{b''b} - B_{a''a}G_{b''b})(\mathrm{Ad}_{\tilde{h}})_{a'}^{a''}(\mathrm{Ad}_{\tilde{h}})_{b'}^{b''}. \end{cases} \tag{A.13}$$

It is not hard to see that these are the symmetric and skew components of the following expression:

$$\mathbb{d}E_{ab} = (\mathbb{A}^c\tilde{h})\, \tilde{c}_c^{\,a'b'} E_{aa''}E_{b''b}(\mathrm{Ad}_{\tilde{h}})_{a'}^{a''}(\mathrm{Ad}_{\tilde{h}})_{b'}^{b''}, \tag{A.14}$$

which is essentially the KS condition as originally expressed by Klimčík and Ševera (see [32] and the end of this appendix for an explicit comparison).

Contracting this expression from the left and from the right with $\widetilde{E}^{ab} = (E_{ab})^{-1}$, which as we will soon prove exists if $\pi G^{-1}$ has eigenvalues of norm smaller than 1, one can rewrite (A.14), and therefore (A.2), as

$$\mathbb{d}\widetilde{E}^{ab} = -(\mathbb{A}^c\tilde{h})\, \tilde{c}_c^{\,a'b'}(\mathrm{Ad}_{\tilde{h}})_{a'}^{a}(\mathrm{Ad}_{\tilde{h}})_{b'}^{b}. \tag{A.15}$$

Finally, we get to the desired equivalence between (A.2) and (A.3) (written in the form (A.6)) by decomposing this equation once again on its symmetric and skew components, to obtain respectively:

$$\mathbb{d}M^{ab} = 0 \quad \text{and} \quad \mathbb{d}\pi^{ab} = -(\mathbb{A}^c\tilde{h})\, \tilde{c}_c^{\,a'b'}(\mathrm{Ad}_{\tilde{h}})_{a'}^{a}(\mathrm{Ad}_{\tilde{h}})_{b'}^{b}. \tag{A.16}$$

We have thus shown that the equivalence between (A.2) and (A.3) provided that $\widetilde{E}_{\mu\nu}^{ab}$ is invertible. We need now to show that $\widetilde{E}^{ab}$ is invertible if $\pi G^{-1}$ has eigenvalues of norm

smaller than 1. We do so by showing that, given this hypothesis, we can define a $G_{ab}$ and $B_{ab}$ with the right properties.

Let $G_{ab}$ and $B_{ab}$ be defined by[57]

$$MG \doteq (1 - \pi M^{-1}\pi M^{-1})^{-1} \quad \text{and} \quad MB \doteq -\pi M^{-1}(1 - \pi M^{-1}\pi M^{-1})^{-1}, \qquad (A.17)$$

where the right hand side can be defined through a power series which converges thanks to the assumption on the norm of the eigenvalues of $\pi M^{-1}$. From this power series it is easy to see that $G$ (resp. $B$) is symmetric (resp. skew) since each term in the power series defining it is so. To check that $\widetilde{E}^{ab} = M^{ab} + \pi^{ab}$ is invertible, we show that $E_{ab} = G_{ab} + B_{ab}$ is indeed its inverse:

$$
\begin{aligned}
(M + \pi)(G + B) &= (1 + \pi M^{-1})(MG + MB) \\
&= (1 + \pi M^{-1})(1 - \pi M^{-1})(1 - \pi M^{-1}\pi M^{-1})^{-1} = 1. \qquad (A.18)
\end{aligned}
$$

Finally, to conclude the proof of the theorem, we need to show that

$$\widetilde{E}^{ab}_{\mu\nu} \doteq M^{ab}\gamma_{\mu\nu} + \pi^{ab}\epsilon_{\mu\nu} \qquad (A.19)$$

is the inverse of $E^{\mu\nu}_{ab} \doteq G_{ab}\gamma^{\mu\nu} + B_{ab}\epsilon^{\mu\nu}$, that is

$$
\begin{aligned}
\widetilde{E}^{ab}_{\mu\nu} E^{\nu\rho}_{bc} &= (M^{ab}\gamma_{\mu\nu} + \pi^{ab}\epsilon_{\mu\nu})(G_{bc}\gamma^{\nu\rho} + B_{bc}\epsilon^{\nu\rho}) \\
&= (MG + \pi B)^a{}_c \delta^\rho_\mu + (\pi G + MB)^a{}_c \epsilon_\mu{}^\rho \qquad (A.20)
\end{aligned}
$$

However, using (A.17), one has

$$
\begin{aligned}
MG + \pi B &= MG + \pi M^{-1}MB \\
&= (1 - \pi M^{-1}\pi M^{-1})^{-1} + \pi M^{-1}(1 - \pi M^{-1})(1 - \pi M^{-1}\pi M^{-1})^{-1} = 1. \quad (A.21)
\end{aligned}
$$

Similarly,

$$\pi G + MB = \pi M^{-1}MG + MB \qquad (A.22)$$
$$= \pi M^{-1}(1 - \pi M^{-1}\pi M^{-1})^{-1} - \pi M^{-1}(1 - \pi M^{-1}\pi M^{-1})^{-1} = 0. \qquad (A.23)$$

Whereby, one concludes as desired that

$$\widetilde{E}^{ab}_{\mu\nu} E^{\nu\rho}_{bc} = (MG + \pi B)^a{}_c \delta^\rho_\mu + (\pi G + MB)^a{}_c \epsilon_\mu{}^\rho = \delta^a_c \delta^\rho_\mu. \qquad (A.24)$$

$\square$

Whence one readily deduces the following corollary whereby the KS model can be written as the sum of the (would-be topological) Poisson $\sigma$-model [87] over the Poisson-Lie group $(\mathcal{G}, \pi)$ and a Proca-like term that breaks its gauge symmetry and topological nature:

**Proposition A.2.** The KS sigma model (130) is obtained by integrating out $A$ from the following non-topological $M$-deformation of the Poisson-Lie $\sigma$-model on $C^\infty(\Sigma, \mathcal{G}) \times \Omega^1(\Sigma, \mathfrak{g}^*)$ $\ni (\tilde{h}, A)$:

$$\mathcal{L}_{\mathrm{dPL}}(\tilde{h}, A) = A_a \wedge \Delta^a \tilde{h} - \frac{1}{2}\pi^{ab}(\tilde{h})A_a \wedge A_b - \frac{1}{2}M^{ab}A_a \wedge \star_\gamma A_b. \qquad (A.25)$$

where $\star_\gamma$ is the Hodge star operator associated to the worldsheet metric $\gamma_{\mu\nu}$. $\diamond$

---

[57]This formula is analogous to the following decomposition of $1/(1+x)$:

$$\frac{1}{1+x} = \frac{1}{1-x^2} - \frac{x}{1-x^2}.$$

To the best of our knowledge this Lagrangian has not appeared in the literature before. It features some notable limits:

1. If $M^{ab} = 0$, then one obtains the topological Poisson $\sigma$-model over the Poisson-Lie group $(\mathcal{G}, \pi)$; this model is equivalent to the gauged Wess-Zumino-Witten model [88–91]; in this limit one cannot integrate out the $A$ field from $\mathcal{L}_{\mathrm{dPL}}$ since $\pi$ is degenerate and thus $\tilde{E}^{ab}_{\mu\nu} = \pi^{ab}\epsilon_{\mu\nu}$ fails to be invertible.

2. If $(\mathcal{G}, \pi)$ is linear, i.e. $(\mathcal{G}, \pi)$ is given by $(\mathfrak{g}, \delta)$, then upon integration by parts, the model reduces to a 2d non-topological deformation of the BF model, i.e. $\mathcal{L}_{\mathrm{dBF}} = B^a \wedge F_a(A) - \frac{1}{2}M^{ab}A_a \wedge \star_\gamma A_b$ [64, Eq. 6.15]. This model is not topological since the term $-\frac{1}{2}M^{ab}A_a \wedge \star A_b$ breaks the $\mathfrak{g}^*$-gauge symmetry $\delta_X(A, B) = (d_A X, \widetilde{\mathrm{ad}}^*_X B)$. The 2d (non-Abelian) $\mathfrak{g}^*$ Proca Lagrangian of mass tensor $M$ can be found by adding to the deformed BF Lagrangian the extra term $\star_\gamma \kappa_{ab} B^a B^b$, for an $\mathrm{ad}$-invariant inner product $\kappa_{ab}$ over $\mathfrak{g}$, and integrating out the $B$-field [87].

3. If $\mathcal{G}^* \simeq \mathfrak{g}^*$ is Abelian and $M^{ab}$ is positive definite and $\mathrm{Ad}$-invariant, then $\pi^{ab} = 0$ and, upon integration of the fields $A_a$, the above model reduces to the principal chiral $\sigma$-model with target $\mathcal{G}$ and "trace" given by $G_{ab} = (M^{ab})^{-1}$, i.e. $\mathcal{L}_{\mathrm{chiral}} = \frac{1}{2}G_{ab}\gamma^{\mu\nu}\Delta^a_\mu \tilde{h}\Delta^b_\nu \tilde{h}$.

4. If $M^{ab} = 0$ and $\pi^{ab}$ were symplectic—which is *never* the case if $(\mathcal{G}, \pi)$ is a Poisson-Lie group as in all the cases considered so far—then the Lagrangian (A.25) would give back the A-model Lagrangian considered in §3.2, equation (110).

Finally, to make the relationship with the KS model as presented in most of the literature explicit, we proceed as follows. First, let $z^\alpha$ be arbitrary coordinates on $\mathcal{G}$ so that, with slight abuse of notation:

$$\Delta^a_\mu \tilde{h} = \left(\frac{\partial \tilde{h}}{\partial z^\alpha}\tilde{h}^{-1}\right)^a \partial_\mu z^\alpha \doteq (\Delta^a_\alpha \tilde{h})\partial_\mu z^\alpha, \tag{A.26}$$

and the (KS) Lagrangian (130) can be written as

$$\mathcal{L} = \frac{1}{2}E^{\mu\nu}_{ab}(\Delta^a_\mu \tilde{h})(\Delta^b_\nu \tilde{h}) = \frac{1}{2}E^{\mu\nu}_{ab}(\Delta^a_\alpha \tilde{h})(\Delta^b_\beta \tilde{h})\partial_\mu z^\alpha \partial_\nu z^\beta \doteq \frac{1}{2}E^{\mu\nu}_{\alpha\beta}\partial_\mu z^\alpha \partial_\nu z^\beta. \tag{A.27}$$

We first specialize to $\gamma_{\mu\nu} = \eta_{\mu\nu}$, the 2d Minkowski metric, and express it in coordinates where $\eta_{\mu\nu} = \mathrm{diag}(-1, 1)$; the corresponding Levi-Civita tensor $\epsilon_{\mu\nu}$ is thus given by $1 = \epsilon_{01} = -\epsilon^{01}$; and then we choose lightcone coordinates $x^\pm = x^1 \pm x^0$ on the worldsheet; whence the corresponding lightcone derivatives are $\partial_\pm = \frac{1}{2}(\partial_1 \pm \partial_0)$. Putting everything together, the KS Lagrangian (130) finally reads

$$\begin{aligned}
\mathcal{L}(z) &= \frac{1}{2}E^{\mu\nu}_{\alpha\beta}\partial_\mu z^\alpha \partial_\nu z^\beta \\
&= \frac{1}{2}E_{(\alpha\beta)}\eta^{\mu\nu}\partial_\mu z^\alpha \partial_\nu z^\beta + \frac{1}{2}E_{[\alpha\beta]}\epsilon^{\mu\nu}\partial_\mu z^\alpha \partial_\nu z^\beta \\
&= \frac{1}{2}E_{\alpha\beta}(\partial_1 z^\alpha \partial_1 z^\beta - \partial_0 z^\alpha \partial_0 z^\beta + \partial_1 z^\alpha \partial_0 z^\beta - \partial_0 z^\alpha \partial_1 z^\beta) \\
&= 2E_{\alpha\beta}\partial_- z^\alpha \partial_+ z^\beta, \tag{A.28a}
\end{aligned}$$

which is the form most commonly used in the literature, possibly up to a global normalization factor.

## B  Kalb-Ramond 2-gauge invariance of the (KS) String

We start by computing the equations of motion of the (KS) string. Using arbitrary coordinates $z^\alpha$ on $\mathcal{G}$, as at the end of Appendix A, the (KS) string Lagrangian (130) can be written as in (A.27), namely

$$\mathcal{L} = \frac{1}{2} E^{\mu\nu}_{\alpha\beta} z^\alpha_{,\mu} z^\beta_{,\nu}, \tag{B.1}$$

where, for this appendix, we introduced the compact notation $z^\alpha_{,\mu} \doteq \partial_\mu z^\alpha$.

With respect to the worldsheet coordinates $z^\alpha$, the equations of motion of the (KS) string take the form

$$\mathcal{E}_\gamma = \frac{1}{2}(\partial_\gamma E^{\mu\nu}_{\alpha\beta} - \partial_\alpha E^{\mu\nu}_{\gamma\beta} - \partial_\beta E^{\mu\nu}_{\alpha\gamma}) z^\alpha_{,\mu} z^\beta_{,\nu} - E^{\mu\nu}_{\gamma\alpha} z^\alpha_{,\mu\nu}. \tag{B.2}$$

Decomposing $E$ into its symmetric and skew parts as per (A.9) in the previous appendix, one finds:

$$\mathcal{E}_\gamma = -G_{\gamma\gamma'}\left(z^{\gamma'}_{,\mu\nu} - \Gamma^{\gamma'}_{\alpha\beta} z^\alpha_{,\mu} z^\beta_{,\nu}\right)\gamma^{\mu\nu} + \frac{1}{2} F_{\gamma\alpha\beta} z^\alpha_{,\mu} z^\beta_{,\nu} \epsilon^{\mu\nu}, \tag{B.3}$$

where $\Gamma$ is the target space Christoffel symbol for the metric $G_{ab}$, and $F$ is the the "magnetic 3-form" Kalb-Ramond flux,

$$\Gamma^{\gamma'}_{\alpha\beta} \doteq \frac{1}{2} G^{\gamma'\gamma}(\partial_\gamma G_{\alpha\beta} - \partial_\alpha G_{\gamma\beta} - \partial_\beta G_{\alpha\gamma}), \tag{B.4}$$

$$F_{\gamma\alpha\beta} \doteq (\partial_\gamma B_{\alpha\beta} + \partial_\alpha B_{\beta\gamma} + \partial_\beta B_{\gamma\alpha}) = 3\partial_{[\gamma} B_{\alpha\beta]}. \tag{B.5}$$

As well known, equation (B.3) is the stringy generalization of the geodesic equation for a point particle moving in a target space coupled to a magnetic field $F$ (Lorentz force). Note that these equations are tensorial with respect to both the worldsheet and the target space, and invariant with respect to the "2-gauge" transformation of the background Kalb-Ramond potential

$$B_{\alpha\beta} \mapsto B_{\alpha\beta} + 2\partial_{[\alpha}\Xi_{\beta]}. \tag{B.6}$$

Under such a transformation, the presymplectic form (pSF) current $\Omega^\mu$ changes as follows. Denoting by $x^\mu$ a set of coordinates on the worldsheet $\Sigma$, the presymplectic potential (pSP) current is

$$\boldsymbol{\theta} \doteq \star_\gamma(\gamma_{\mu\nu}\theta^\mu dx^\nu) = \epsilon_{\mu\nu}\theta^\mu dx^\nu, \tag{B.7}$$

where $\star_\gamma$ is the Hodge star operator associated to the worldsheet metric $\gamma_{\mu\nu}$. Whence, explicitly,

$$\boldsymbol{\theta} = -\mathbb{d}z^\alpha(G_{\alpha\beta} z^\beta_{,\mu}\epsilon^\mu{}_\nu + B_{\alpha\beta} z^\beta_{,\nu})dx^\nu. \tag{B.8}$$

Then, if $B_{\alpha\beta} \mapsto B_{\alpha\beta} + 2\partial_{[\alpha}\Xi_{\beta]}$,

$$\boldsymbol{\theta} \mapsto \boldsymbol{\theta} + d(\Xi_\alpha \mathbb{d}z^\alpha) + \mathbb{d}(\Xi_\alpha dz^\alpha), \tag{B.9}$$

where we used that $dz^\alpha = z^\alpha_{,\mu} dx^\mu$. Thus, the pSC is changed by a boundary term:

$$\boldsymbol{\Omega} \mapsto \boldsymbol{\Omega} - d(\mathbb{d}\Xi_\alpha \wedge \mathbb{d}z^\alpha). \tag{B.10}$$

The closed KS string is therefore left completely unaffected by these "gauge" transformations of the worldsheet tensor $B_{\alpha\beta}$, since neither its equations of motion not its symplectic structure are affected. Presently, the significance of these Kalb-Ramond 2-gauge transformations for the KS string, e.g. in terms of $M^{\alpha\beta}$ and $\pi^{\alpha\beta}$, remains unclear to us.

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
