# Peer review of "Beyond Noether: A Covariant Study of Poisson-Lie Symmetries in Low Dimensional Field Theory"

_SciPost Physics_

## Round 1 · Referee Report · Anonymous (Referee 1) · 2025-10-29

Strengths

1- It reformulates Poisson-Lie symmetries in a rigorous mathematical language 2- It opens the possibility for a community of mathematical physicists to work on the subject of Poisson-Lie symmetries.

Weaknesses

1- Because of the very technical language, it is not accessible to people who are not used to a heavy mathematical formulation. 2- For the moment it is only a first step, because it only rephrases known concepts in a new language without adding new important results.

Report

The article proposes a formulation of Poisson-Lie symmetries in a covariant and rigorously mathematical language. It studies the application of this formulation in mechanical systems and in field theories in 2 and 3 dimensions. The article meets the criteria of the journal because it provides a novel and synergetic link between different research areas. In fact, the reformulation of Poisson-Lie symmetries in this language may make the topic accessible and appealing to certain communities of mathematical physics, who are very mathematically oriented.

Recommendation

Publish (meets expectations and criteria for this Journal)

---

## Round 1 · Referee Report · Anonymous (Referee 2) · 2025-11-2

Report

In the present work authors develop a spacetime-covariant framework for Poisson–Lie (PL) symmetries and illustrate it for several field-theoretic setups, which goes beyond Noether formalism and the standard variational Lagrangian-based treatment of global symmetries. One of the goals is to characterize PL symmetries directly within the covariant phase space formalism, where symmetries act as flows on the solution space. In contrast to Noether procedure, Poisson–Lie symmetries, arise in the classical regime of the quantum group symmetries, in general possess non-Abelian group-valued momentum maps, non–symplectomorphic actions, as well as can develop non-locality in charges and symmetry actions.

A new definition of PL symmetries on CPS replaced linear Noether exactness condition with Maurer-Cartan type, led to modification of EOM in terms of deformed zero-curvature condition and produced group-valued charges and a non-Abelian cocycle condition. This provided a consistent notion of PL symmetries as deformation of the Noetherian counterpart. Importantly, authors clearly demonstrate distinct properties of the first and second order formalism in the context of CPS approach and highlight independence of Cauchy embedding. A systematic analysis of PL flow equation, modified locality, compatibility with conservation was provided for all three frameworks: generalised spinning top (incl. curved spaces), open/closed KS string (with NATD) and 3-dim gravity (BF realisation), where interconnections of three models have been demonstrated at the level of charge localisation, i.e. non-trivial PL charges required codimension-d localization (points-endpoints-vertices accordingly).

The emergence of new nontrivial PL symmetries in the CPS formalism can be very useful in a variety of (integrable) systems. Although some of the results are expectable (e.g. closed KS string), in general, identification of new PL symmetries can be particularly illuminating in description of model properties, like hidden symmetries in field-theoretic and lattice setups. Despite authors providing explicit steps with structural definitions, it would be of particular interest to find a certain unified PL procedure for various classes of models (group symmetries, dimensions) with a criterion for existence of nontrivial PL symmetries.

Another related point would be to investigate actual higher dimensional PL integrable structure. Such constructions would also appear important in higher D TQFT systems (e.g. 4- and higher D Chern-Simons theory, especially with order defects), where, as also noted, momentum maps would require ordered hypersurface integrals and higher symmetries. It would be helpful in the classical integrable structure of new (supersymmetric) sigma models obtained from such higher-dim TQFT. Relatedly, current PL approach would be particularly useful for classical regimes of higher simplices (e.g. tetrahedron, as in exceptional Lie M-brane systems), their solution spaces and conjectured topological invariants (as well as in relation to WRT). By means of the present construction, it would be important to further explore specific hybrid systems with classical/quantum phase space structure (CM type and new classical limits).

The article contains clear propositions, proofs and is self-consistent, which I would recommend for publication. Minor insignificant errata:

p. 6, l. 185: “On the space $ M \times \mathcal{F} $, one can defined the bi-complex” -> define

p. 32, l. 978: “transformation of the (off-shell) worldine $ \xi(t) $” -> worldline

P. 32, l. 989: “the dynamics $ \mathcal{H} = c_1 Tr(h)−c_2 $ for e.g., $ G = SU(2) $” -> “$ \mathcal{H} = c_1 Tr(h)−c_2 $ e.g. ”

p. 63, l. 1917: “interplay between between Poisson-Lie symmetric field theories” -> “interplay between Poisson-Lie”

p. 37, l. 1137: “where on-shell, neither $q$ nor $\omega$ dependss on the choice of $ C \rightarrow \Sigma $.” -> depends

Requested changes

p. 6, l. 185: “On the space $ M \times \mathcal{F} $, one can defined the bi-complex” -> define

p. 32, l. 978: “transformation of the (off-shell) worldine $ \xi(t) $” -> worldline

P. 32, l. 989: “the dynamics $ \mathcal{H} = c_1 Tr(h)−c_2 $ for e.g., $ G = SU(2) $” -> “$ \mathcal{H} = c_1 Tr(h)−c_2 $ e.g. ”

p. 63, l. 1917: “interplay between between Poisson-Lie symmetric field theories” -> “interplay between Poisson-Lie”

p. 37, l. 1137: “where on-shell, neither $q$ nor $\omega$ dependss on the choice of $ C \rightarrow \Sigma $.” -> depends

Recommendation

Publish (meets expectations and criteria for this Journal)

---

## Round 2 · List of Changes

Change 1: Replaced the "Funding Information" template at p. 63, l. 1931, with the appropriate funding information of the authors.

Change 2: Corrected the following minor grammatical typos as per request of referee and editor in charge:

p. 6, l. 185: "On the space $M\times \mathcal{F}$, one can defined the bi-complex" -> define

p. 32, l. 978: “transformation of the (off-shell) worldine $\xi(t)$” -> worldline

p. 32, l. 989: "the dynamics $\mathcal{H} = c_{1}{\rm Tr}(h)-c_{2}$ for e.g., $G = SU(2)$" -> "$\mathcal{H} = c_{1}{\rm Tr}(h)-c_{2}$ e.g."

p. 63, l. 1917: “interplay between between Poisson-Lie symmetric field theories” -> “interplay between Poisson-Lie”

p. 37, l. 1137: "where on-shell, neither $q$ nor $\omega$ dependss on the choice of $C\to \Sigma$. -> depends

---

## Editorial Decision

in_refereeing